# Stronger Approximation Guarantees for Non-Monotone $\gamma$-Weakly DR-Submodular Maximization

## Abstract

We study the maximization of nonnegative, non-monotone $\gamma$-weakly diminishing returns (DR) submodular functions over down-closed convex bodies. The weakly DR model relaxes classical diminishing returns by allowing marginal gains to decay up to a multiplicative factor $\gamma \in (0, 1]$, capturing a broad class of objectives that interpolate between monotone and fully non-monotone DR submodularity. Existing methods in this regime achieve guarantees that deteriorate rapidly as $\gamma$ decreases and fail to recover the best known bounds in the fully DR case.

We develop a $\gamma$-aware algorithmic framework that combines a Frank-Wolfe guided measured continuous greedy procedure with a $\gamma$-weighted double-greedy method. Our analysis explicitly accounts for the asymmetric structure induced by weak diminishing returns, yielding $\gamma$-dependent progress certificates that remain valid across the entire weakly DR spectrum. As a result, we obtain an approximation guarantee that strictly improves upon the baseline $\gamma e^{-\gamma}$ for all $\gamma \in (0, 1)$ and recovers the current best constant $0.401$ when $\gamma = 1$. The proposed algorithms are projection free, use only first order information and linear optimization oracles, and run in polynomial time.

## 1 Introduction

Submodular maximization under various constraints is a central problem in optimization and theoretical computer science. Foundational work established much of the modern toolkit and sparked a rich line of research (Conforti & Cornuéjols, 1984; Fisher et al., 2009; Korte & Hausmann, 1978; Nemhauser & Wolsey, 1978; Nemhauser et al., 1978). Informally, a set function is submodular if marginal gains decrease as the chosen set grows (a discrete "diminishing returns" property). One key reason the problem remains important is its wide scope: many classic combinatorial tasks can be cast as maximizing a submodular objective, including *Max Cut*, the *assignment problem*, *facility location*, and *max bisection* (Håstad, 2001; Karp, 2009; Chekuri & Khanna, 2005; Feige & Vondrák, 2006; Cornuejols et al., 1977; Austrin et al., 2016).

On the continuous side, Diminishing Returns (DR) submodular models support Maximum A Posteriori (MAP) inference for determinantal point processes (Bian et al., 2017a; 2019). Informally, a DR-submodular function is a continuous analogue of submodularity in which marginal gains along each coordinate decrease as the current point increases (a "diminishing returns" property in $\mathbb{R}^n$). We give the formal definition in the preliminaries. The problem also arise in online allocation and learning (Chen et al., 2018; Zhang et al., 2019).

Classical results for continuous DR-submodular maximization are based on projection free, first order methods with provable guarantees. Foundational work developed the geometry and algorithms for continuous DR-submodular maximization (Bian et al., 2017a), as well as optimal algorithms for continuous non-monotone DR-submodular objectives (Niazadeh et al., 2020). This paradigm extends to online and stochastic information models (Chen et al., 2018; Zhang et al., 2022), and recent progress sharpens bounds and constraint handling via DR-based analyses (Buchbinder & Feldman, 2024). Within this line, the *weakly* DR framework broadens modeling reach by relaxing diminishing returns to a factor $\gamma \in (0, 1]$: roughly, marginal gains are still decreasing, but only up to a controlled multiplicative slack $\gamma$. The parameter $\gamma$ is a structural property

of the objective. It quantifies how closely the function satisfies full diminishing returns: $\gamma = 1$ recovers the classical DR-submodular setting, while smaller values of $\gamma$ allow weaker one sided marginal decay. This provides unified algorithms and guarantees (Hassani et al., 2017; Pedramfar et al., 2023; 2024a; Pedramfar & Aggarwal, 2024; Nie et al., 2024). These developments motivate our focus on continuous DR and weakly DR objectives over down-closed convex bodies.

A range of first order methods are known for continuous DR and weakly DR maximization over down-closed convex bodies. *Projected gradient* methods are the most basic scheme: they take a step in the direction of the gradient and then project back to the feasible region $P$ (Hassani et al., 2017; Zhang et al., 2022). *Frank-Wolfe (projection free)* methods, also known as conditional gradient methods, avoid projections by instead solving a linear subproblem $\max_{y \in P}\langle y, \nabla F(x)\rangle$ and moving towards this direction; they leverage DR/weakly-DR restricted concavity to certify progress (Jaggi, 2013; Lacoste-Julien & Jaggi, 2015; Bian et al., 2017a; Niazadeh et al., 2020; Pedramfar et al., 2023). *Double-greedy* style methods adapt the discrete bracketing idea to the continuous setting. They keep two solutions, a "lower" and an "upper" one, and repeatedly adjust their coordinates in opposite directions so that the two solutions move closer together (Buchbinder et al., 2015; Niazadeh et al., 2020; Pedramfar et al., 2023). These techniques have also been extended to online, bandit, and stochastic models (Chen et al., 2018; Zhang et al., 2022; Pedramfar et al., 2023).

At a high level, our algorithm consists of two interacting components. The first is a $\gamma$-aware Frank Wolfe guided measured continuous greedy procedure, which constructs a feasible fractional trajectory inside the down-closed convex body using only linear optimization steps. This part is responsible for global exploration: it follows promising first order directions, accumulates gain over the feasible region, and yields a global progress certificate. The second component is a $\gamma$-aware double-greedy argument applied to the same solution structure. Its role is complementary: it provides a local certificate that is robust to non-monotonicity and controls the loss caused by adverse directions that may not be handled effectively by the continuous greedy phase alone.

Our method combines these two ingredients in a unified way. We design a $\gamma$-aware *Frank-Wolfe guided measured continuous Greedy* ($\gamma$-FWG) algorithm with $\gamma$-dependent thresholds and progress certificates, pair it with a $\gamma$-aware *double-greedy* step, and then optimize a convex combination of their guarantees. Thus, the final approximation bound $\Phi_\gamma$ is obtained not from either component in isolation, but from their optimized mixture. This interaction is crucial: the Frank-Wolfe guided component is stronger in regimes where global exploration is most informative, while the double-greedy component is stronger in regimes where non-monotone effects are more pronounced. By combining them, we obtain a guarantee that strictly improves the baseline (Bian et al., 2017a; Pedramfar et al., 2024b) for all $\gamma \in (0,1)$ and matches the DR boundary at $\gamma = 1$ (Buchbinder & Feldman, 2024).

Beyond establishing new approximation guarantees and improving the best known bounds, our approach introduces the following *technical novelties*:

- We design a novel, $\gamma$-aware Frank Wolfe guided measured continuous greedy algorithm for the non-monotone $\gamma$-weakly DR setting. Our method introduces $\gamma$ dependent threshold schedules and progress certificates that balance ascent along Frank-Wolfe directions with measured updates, while preserving feasibility and ensuring a monotone decay of the residual gap.

- Weakly-DR functions behave asymmetrically. In the DR case, one has the 'naive' inequality

$$F(\mathbf{x}) \geq \frac{F(\mathbf{x} \vee \mathbf{y}) + F(\mathbf{x} \wedge \mathbf{y})}{2}. \tag{1}$$

In contrast, in the $\gamma$-weakly DR setting, Lemma 3.1 yields

$$F(\mathbf{x}) \geq \frac{\gamma^2\,F(\mathbf{x} \vee \mathbf{y}) + F(\mathbf{x} \wedge \mathbf{y})}{1 + \gamma^2}. \tag{2}$$

Here only the $F(\mathbf{x} \vee \mathbf{y})$ term is scaled by $\gamma^2$, so the two sides of the inequality are no longer treated symmetrically. Similar one sided $\gamma$-dependence appears in Lemmas 2.1, 2.2, and several auxiliary results in the appendix. This loss of symmetry breaks the standard potential based arguments used

in classical DR analyses. Our approach therefore uses a case based progress analysis that explicitly tracks one sided marginal decay and introduces $\gamma$-aware thresholds that couple Frank-Wolfe steps with measured updates, allowing us to certify progress despite this asymmetry.

- In parallel, we adapt the classical double–greedy potential to a $\gamma$-weighted variant that explicitly balances asymmetric gains and losses, yielding tight progress guarantees across the weakly DR regime.

## 1.1 Our Contribution

In this paper, we consider *non-monotone* $\gamma$-weakly DR-submodular objectives over a down-closed convex body $P \subseteq [0,1]^n$ with $0 < \gamma \leq 1$. Informally, $\gamma$-weak DR means that marginal gains decrease as coordinates increase, but only up to a factor $\gamma \in (0,1]$ capturing objectives that exhibit *partial*, rather than full, diminishing returns. In this regime, the canonical approximation envelope is $\kappa(\gamma) = \gamma e^{-\gamma}$ (which recovers $e^{-1}$ at $\gamma = 1$) (Bian et al., 2017a; Pedramfar et al., 2024b). In (Bian et al., 2017a), this approximation guarantee is achieved with time complexity $\mathcal{O}(1/\varepsilon)$, where $\varepsilon > 0$ is an accuracy parameter, while (Pedramfar et al., 2024b) attains the same guarantee with running time $\mathcal{O}(1/\varepsilon^3)$. In contrast, our algorithm achieves the $\Phi_\gamma$-approximation in time $\text{Poly}(n, \delta^{-1})$. Recently, Buchbinder and Feldman introduced a novel technique that yields a 0.401-approximation for the (fully) DR-submodular case (Buchbinder & Feldman, 2024); their algorithm also runs in time $\text{Poly}(n, \delta^{-1})$. For direct comparison see Table 1.

*This paper aims to close the gap between weakly and full DR.* We develop $\gamma$-aware algorithms and analyses that (i) recover the classical DR constant at $\gamma = 1$ (Buchbinder & Feldman, 2024) and (ii) strictly improve upon the baseline $\kappa(\gamma) = \gamma e^{-\gamma}$ throughout the weakly-DR regime (Bian et al., 2017a; Pedramfar et al., 2024b). Our approach combines a $\gamma$-aware Frank Wolfe guided measured continuous greedy subroutine with a $\gamma$-aware double-greedy, and then optimizes a convex mixture of their certificates. The resulting guarantee $\Phi_\gamma$ is an *objective value* determined by three tunable parameters $(\alpha, r, t_s)$—where $\alpha$ is the mixing weight between certificates and $(r, t_s)$ govern schedule/tuning in the FW-guided and double-greedy components—and we choose these to maximize $\Phi_\gamma$ at each given $\gamma$. A formal statement of our main result appears as Theorem 4.5. To contextualize our guarantees, Figure 1 plots the optimized curve $\Phi_\gamma$, and Table 2 reports representative values and the associated parameter choices $(\alpha, r, t_s)$.

In this direction, our *key contributions* are summarized below each item highlights a distinct component of our algorithmic framework and its guarantee.

- We present a $\gamma$-aware Frank Wolfe guided measured continuous greedy and a $\gamma$-aware double-greedy, each delivering explicit constant factor guarantees for $\gamma$-weakly DR objectives over down-closed convex bodies.

- We derive a parameter optimized convex mixture of the two certificates producing a performance curve $\Phi_\gamma$ that strictly improves the prior baseline $\kappa(\gamma) = \gamma e^{-\gamma}$ for all $\gamma \in (0,1)$ and matches the classical DR constant at $\gamma = 1$.

- Our proofs are modular and avoid curvature assumptions; they recover the DR boundary as a special case and extend smoothly across the weakly-DR spectrum.

- The methods use only first order information and linear optimization over $P$ (Frank Wolfe oracles), making them projection free and suitable for large scale instances.

Beyond its theoretical interest, our framework is further motivated by machine learning applications including cost aware feature selection, sparse GLMs, and dictionary selection; see Appendix G for a detailed discussion. In these settings, weak submodularity type structural parameters arise naturally, and explicit feature costs can induce non-monotone objectives under down-closed budget constraints (Das & Kempe, 2011; Elenberg et al., 2018).

Table 1: Comparison of the most relevant related results for *non-monotone* continuous maximization over down-closed convex bodies. Here $\gamma \in (0,1]$ is a structural parameter of the objective, while $\epsilon$ and $\delta$ are algorithmic accuracy/discretization parameters. The baseline weakly-DR guarantee $\kappa(\gamma) = \gamma e^{-\gamma}$ has a multiplicative tolerance $O(\epsilon)$ and uses $O(1/\epsilon)$ iterations (Bian et al., 2017a). (Buchbinder & Feldman, 2024), obtains 0.401 approximation with additive discretization/smoothness error $O(\delta D^2 L)$. In our paper, we obtain the guarantee of $(\Phi_\gamma - O(\epsilon)) \operatorname{OPT} - O(\delta L D^2)$, and after setting $\epsilon = \Theta(\delta)$, the final result becomes $\Phi_\gamma \operatorname{OPT} - O(\delta L D^2)$. Thus, the table separates the main constant factor guarantee from the lower order multiplicative and additive error terms.

| Setting | Reference | Approximation Guarantee | Complexity |
|---|---|---|---|
| DR-submodular case only ($\gamma = 1$) over down-closed convex bodies | (Buchbinder & Feldman, 2024) | $0.401 \operatorname{OPT} - O(\delta D^2 L)$ | $\operatorname{Poly}(n, \delta^{-1})$ |
| $\gamma$-weakly DR over down-closed convex bodies | (Bian et al., 2017a) | $(\gamma e^{-\gamma} - O(\epsilon)) \operatorname{OPT}$ | $O(1/\epsilon)$ oracle |
| $\gamma$-weakly DR over down-closed convex bodies | **This paper** | $\Phi_\gamma \operatorname{OPT} - O(\delta L D^2)$ | $\operatorname{Poly}(n, \delta^{-1})$ |

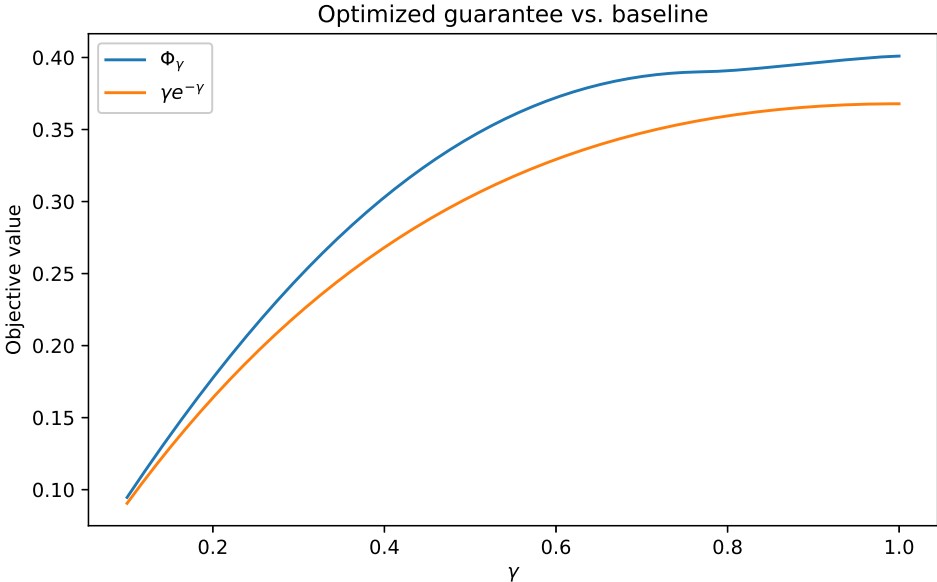

Figure 1: Approximation guarantee versus weakly-DR parameter. The horizontal axis is the weakly-DR parameter $\gamma \in (0,1]$ and the vertical axis is the approximation factor. We plot our optimized guarantee $\Phi_\gamma$ (blue curve) alongside the non-monotone weakly-DR baseline $\kappa(\gamma) = \gamma e^{-\gamma}$ (orange curve). Across the entire regime $\gamma \in (0,1)$, $\Phi_\gamma$ strictly exceeds $\kappa(\gamma)$, and at $\gamma = 1$ (full DR) our curve reaches 0.401, matching the current best bound. Selected parameter choices $(\alpha, r, t_s)$ used to construct $\Phi_\gamma$ are reported in Table 2.

## 1.2 Related Work

For non-monotone, $\gamma$-weakly DR objectives over down-closed convex bodies, unified analyses establish the *baseline envelope* $\kappa(\gamma) = \gamma e^{-\gamma}$ (Pedramfar et al., 2024b). An earlier line derives the same envelope via continuous/"measured" greedy combined with projection free (Frank Wolfe) arguments adapted to one sided weakly-DR gradients (Bian et al., 2017a). A complementary *stationary point baseline* shows that any first order stationary point obtained by projected or mirror ascent achieves $\gamma^2/(1+\gamma^2)$ of OPT (recovering 1/2 at

| $\gamma$ | $\Phi_\gamma$ | $\gamma e^{-\gamma}$ | $\alpha$ | $r$ | $t_s$ |
|------|--------|---------|-------|-------|-------|
| 0.1 | 0.095 | 0.090 | 0.001 | 3.750 | 0.000 |
| 0.2 | 0.178 | 0.164 | 0.000 | 3.083 | 0.000 |
| 0.3 | 0.247 | 0.222 | 0.000 | 2.833 | 0.000 |
| 0.4 | 0.303 | 0.268 | 0.000 | 2.667 | 0.000 |
| 0.5 | 0.345 | 0.303 | 0.000 | 2.583 | 0.000 |
| 0.6 | 0.372 | 0.329 | 0.000 | 2.417 | 0.000 |
| 0.7 | 0.387 | 0.348 | 0.000 | 2.333 | 0.000 |
| 0.8 | 0.391 | 0.359 | 0.054 | 2.250 | 0.075 |
| 0.9 | 0.396 | 0.366 | 0.160 | 2.083 | 0.267 |
| 1.0 | 0.401 | 0.368 | 0.197 | 2.220 | 0.368 |

Table 2: Numerical comparison of our optimized guarantee $\Phi_\gamma$ against non-monotone weakly-DR baseline $\kappa(\gamma) = \gamma e^{-\gamma}$. Each row corresponds to a choice of the weakly-DR parameter $\gamma \in (0, 1]$; the second and third columns report the achieved approximation factor $\Phi_\gamma$ and the baseline, respectively. The final three columns list representative internal parameters $(\alpha, r, t_s)$ used to construct $\Phi_\gamma$: $\alpha$ is the convex mixing weight between the two certificates, while $r$ and $t_s$ are schedule/tuning parameters of the $\gamma$ aware FW-guided measured continuous greedy and $\gamma$-aware double-greedy components (see the algorithm description). All values are rounded to three decimals.

$\gamma = 1$), with stochastic/online refinements using non oblivious surrogates and unbiased gradients (Hassani et al., 2017; Chen et al., 2018; Zhang et al., 2022).

When $\gamma = 1$, the problem reduces to continuous non-monotone DR-submodular maximization over down-closed convex bodies. This line begins with the multilinear/continuous greedy framework (Chekuri et al., 2011) and the *Measured Continuous Greedy* guarantee of $1/e \approx 0.367$ (Feldman et al., 2011), with subsequent improvements culminating in the current best constant $0.401$ (Buchbinder & Feldman, 2024). Hardness results still leave a notable gap for non-monotone objectives under common constraints (with no better than $\approx 0.478$ (Gharan & Vondrák, 2011)), underscoring the importance of tighter algorithms at $\gamma = 1$. Our guarantees match this DR boundary while delivering strict improvements over $\kappa(\gamma)$ for every $\gamma \in (0, 1)$.

## 2 Preliminaries and Notation

In this section, we introduce the basic notation, definitions, and assumptions used throughout the paper. We use boldface letters (e.g., $\mathbf{x}, \mathbf{y}$) to denote vectors in $\mathbb{R}^n$, and write a vector as $\mathbf{x} = (x_1, \cdots, x_n)$. The all ones and all zeros vectors are denoted by $\mathbf{1}$ and $\mathbf{0}$, respectively. We use $\mathbf{e}_i$ to denote the $i$-th standard basis vector in $\mathbb{R}^n$. Let $N$ be the ground set with $|N| = n$ elements. The discrete and continuous hypercubes are defined as

$$\{0, 1\}^n = \{\mathbf{x} \in \mathbb{R}^n : x_i \in \{0, 1\} \; \forall i\},$$

and

$$[0, 1]^n = \{\mathbf{x} \in \mathbb{R}^n : x_i \in [0, 1] \; \forall i\},$$

respectively. For a positive integer $n$, we write $[n] := \{1, 2, \ldots, n\}$.

For $\mathbf{x}, \mathbf{y} \in \mathbb{R}^n$, we use the componentwise order: $\mathbf{x} \le \mathbf{y}$ if and only if $x_i \le y_i$ for all $i$ (and $\mathbf{x} < \mathbf{y}$ if and only if $x_i < y_i$ for all $i$). The *join* and *meet* are defined as

$$\mathbf{x} \vee \mathbf{y} := (\max\{x_i, y_i\})_{i=1}^n, \quad \mathbf{x} \wedge \mathbf{y} := (\min\{x_i, y_i\})_{i=1}^n.$$

We also use the elementwise (Hadamard) product $\mathbf{x} \odot \mathbf{y} \in \mathbb{R}^n$, defined by $(\mathbf{x} \odot \mathbf{y})_i := x_i y_i$ for each $i \in [n]$, and the standard inner product $\langle \mathbf{x}, \mathbf{y} \rangle := \sum_{i=1}^n x_i y_i$. For vectors with entries in $[0, 1]$, the coordinatewise *probabilistic sum* is

$$\mathbf{x} \oplus \mathbf{y} := \mathbf{1} - (\mathbf{1} - \mathbf{x}) \odot (\mathbf{1} - \mathbf{y}). \tag{3}$$

For vectors $\mathbf{x}^{(1)}, \ldots, \mathbf{x}^{(m)} \in [0,1]^n$, we write

$$\bigoplus_{j=1}^m \mathbf{x}^{(j)} = \mathbf{1} - \bigodot_{j=1}^m \big(\mathbf{1} - \mathbf{x}^{(j)}\big) = \mathbf{1} - \big((\mathbf{1} - \mathbf{x}^{(1)}) \odot \cdots \odot (\mathbf{1} - \mathbf{x}^{(m)})\big). \tag{4}$$

The operators $\odot$ and $\oplus$ bind more tightly than vector addition or subtraction, so $\mathbf{x} + \mathbf{y} \odot \mathbf{z}$ means $\mathbf{x} + (\mathbf{y} \odot \mathbf{z})$. When a scalar function is applied to a vector, it is interpreted elementwise; for instance, for $\mathbf{x} \in [0,1]^n$, the vector $e^{\mathbf{x}}$ has entries $\big(e^{\mathbf{x}}\big)_i = e^{x_i}$.

A set $P \subseteq \mathbb{R}^n$ is *convex* if $\lambda \mathbf{x} + (1 - \lambda)\mathbf{y} \in P$ for all $\mathbf{x}, \mathbf{y} \in P$ and $\lambda \in [0,1]$. A *convex body* is a compact, convex set with nonempty interior. A polytope $P \subseteq [0,1]^n$ is *down-closed* if $\mathbf{y} \in P$ implies $\mathbf{x} \in P$ for every $\mathbf{x} \in \mathbb{R}^n$ with $\mathbf{0} \leq \mathbf{x} \leq \mathbf{y}$. We say that $P$ is *solvable* if linear optimization over $P$ can be performed in polynomial time. The (Euclidean) *diameter* of a set $P \subseteq \mathbb{R}^n$ is

$$D := \sup\{ \, \|\mathbf{x} - \mathbf{y}\|_2 : \mathbf{x}, \mathbf{y} \in P \, \}.$$

A nonnegative set function $f : \{0,1\}^n \to \mathbb{R}_{\geq 0}$ is *submodular* if, for all $\mathbf{x}, \mathbf{y} \in \{0,1\}^n$ with $\mathbf{x} \leq \mathbf{y}$ and for all $\mathbf{a} \in \{0,1\}^n$,

$$f(\mathbf{x} \vee \mathbf{a}) - f(\mathbf{x}) \; \geq \; f(\mathbf{y} \vee \mathbf{a}) - f(\mathbf{y}).$$

In the continuous case, a nonnegative function $F : [0,1]^n \to \mathbb{R}_{\geq 0}$ is *diminishing-returns (DR) submodular* if, for all $\mathbf{x}, \mathbf{y} \in [0,1]^n$ with $\mathbf{x} \leq \mathbf{y}$, any coordinate $i \in [n]$, and any $c > 0$ such that $\mathbf{x} + c\,\mathbf{e}_i, \; \mathbf{y} + c\,\mathbf{e}_i \in [0,1]^n$,

$$F(\mathbf{x} + c\,\mathbf{e}_i) - F(\mathbf{x}) \; \geq \; F(\mathbf{y} + c\,\mathbf{e}_i) - F(\mathbf{y}).$$

If $F$ is differentiable, this is equivalent to $\nabla F(\mathbf{x}) \geq \nabla F(\mathbf{y})$ for all $\mathbf{x} \leq \mathbf{y}$ (Bian et al., 2019).

A nonnegative function $F : [0,1]^n \to \mathbb{R}_{\geq 0}$ is *$\gamma$-weakly DR-submodular* if, for all $\mathbf{x}, \mathbf{y} \in [0,1]^n$ with $\mathbf{x} \leq \mathbf{y}$, any $i \in [n]$, and any $c > 0$ with $\mathbf{x} + c\,\mathbf{e}_i, \; \mathbf{y} + c\,\mathbf{e}_i \in [0,1]^n$,

$$F(\mathbf{x} + c\,\mathbf{e}_i) - F(\mathbf{x}) \; \geq \; \gamma\big(F(\mathbf{y} + c\,\mathbf{e}_i) - F(\mathbf{y})\big). \tag{5}$$

When $F$ is differentiable, this is equivalent to $\nabla F(\mathbf{x}) \geq \gamma \nabla F(\mathbf{y})$ for all $\mathbf{x} \leq \mathbf{y}$. This condition holds for some $\gamma > 1$ if and only if $F$ is constant (and a constant $F$ satisfies it for any $\gamma$); it holds for some $\gamma \leq 0$ exactly when $F$ is coordinate wise monotone. Hence, we focus on the nontrivial range $0 < \gamma \leq 1$.

A differentiable function $F : P \to \mathbb{R}$ is *$L$-smooth* if, for all $\mathbf{x}, \mathbf{y} \in P$, it satisfies

$$\|\nabla F(\mathbf{x}) - \nabla F(\mathbf{y})\|_2 \leq L \, \|\mathbf{x} - \mathbf{y}\|_2. \tag{6}$$

Now we discuss some properties of $\gamma$-weakly DR-submodular functions. In the following three lemmas, let $F : [0,1]^n \to \mathbb{R}_{\geq 0}$ be differentiable and $\gamma$-weakly DR-submodular.

**Lemma 2.1.** *For all $\mathbf{x}, \mathbf{y} \in [0,1]^n$ and $\lambda \in [0,1]$, the following hold:*

1. *If $\mathbf{x} \leq \mathbf{y}$, then*

$$F\big(\lambda \mathbf{x} + (1 - \lambda)\mathbf{y}\big) \; \geq \; \frac{\lambda \, F(\mathbf{x}) + \gamma^2 (1 - \lambda) \, F(\mathbf{y})}{\lambda + \gamma^2 (1 - \lambda)}. \tag{7}$$

   *Equivalently,*

$$F\big((1 - \lambda)\mathbf{x} + \lambda \mathbf{y}\big) \; \geq \; \frac{(1 - \lambda) \, F(\mathbf{x}) + \gamma^2 \lambda \, F(\mathbf{y})}{(1 - \lambda) + \gamma^2 \lambda}. \tag{8}$$

2. *If $\mathbf{x} + \mathbf{y} \in [0,1]^n$, then*

$$F(\mathbf{x} + \lambda \mathbf{y}) - F(\mathbf{x}) \; \geq \; \frac{\gamma^2 \lambda}{1 - \lambda + \gamma^2 \lambda} \, \big(F(\mathbf{x} + \mathbf{y}) - F(\mathbf{x})\big). \tag{9}$$

**Lemma 2.2.** *For every* $\mathbf{x}, \mathbf{y} \in [0,1]^n$, *the following inequalities hold:*

*1. If* $\mathbf{x} + \mathbf{y} \leq \mathbf{1}$, *then*

$$\langle \nabla F(\mathbf{x}), \mathbf{y} \rangle \ \geq \ \gamma \big( F(\mathbf{x} + \mathbf{y}) - F(\mathbf{x}) \big). \tag{10}$$

*2. If* $\mathbf{x} - \mathbf{y} \geq \mathbf{0}$, *then*

$$\langle \nabla F(\mathbf{x}), \mathbf{y} \rangle \ \leq \ \frac{1}{\gamma} \big( F(\mathbf{x}) - F(\mathbf{x} - \mathbf{y}) \big). \tag{11}$$

**Lemma 2.3.** *For any fixed* $\mathbf{y} \in [0,1]^n$, *define*

$$G_{\oplus}(\mathbf{x}) \ := \ F(\mathbf{x} \oplus \mathbf{y}) \quad and \quad G_{\odot}(\mathbf{x}) \ := \ F(\mathbf{x} \odot \mathbf{y}). \tag{12}$$

*Then both* $G_{\oplus}$ *and* $G_{\odot}$ *are nonnegative and* $\gamma$*–weakly DR-submodular.*

Proofs of these lemmas are provided in Appendix B. These statements generalize prior results from (Zhang et al., 2022; Hassani et al., 2017) to the $\gamma$-weakly DR setting and hold for all $\gamma \in (0,1]$; in particular, they coincide exactly with the classical DR statements when $\gamma = 1$.

## 3 Supporting Results

In this section, we generalize two standard algorithms to the $\gamma$–weakly DR setting and present their analyses. First, we prove a $\gamma$-weighted Frank–Wolfe certificate over solvable convex bodies (Theorem 3.2). Second, we develop a $\gamma$-aware Double–Greedy algorithm and establish an unbalanced lower bound that interpolates smoothly in $\gamma$ (Theorem 3.3).

We generalize the classical DR framework to the $\gamma$–weakly DR setting and obtain guarantees that *continuously interpolate* between weakly and full DR. Our first ingredient (Lemma 3.1) shows that the local-optimality certificate $\langle \mathbf{y} - \mathbf{x}, \nabla F(\mathbf{x}) \rangle \leq 0$ can be translated, under $\gamma$–weakly DR-submodularity, into a $\gamma$–*weighted* value comparison between $F(\mathbf{x})$ and the join/meet values $F(\mathbf{x} \vee \mathbf{y})$ and $F(\mathbf{x} \wedge \mathbf{y})$, recovering the classical $\frac{1}{2}\big(F(\mathbf{x} \vee \mathbf{y}) + F(\mathbf{x} \wedge \mathbf{y})\big)$ bound at $\gamma = 1$ (cf. (Chekuri et al., 2011); see also (Bian et al., 2017a;b; Chen et al., 2018)). Our second ingredient (Theorem 3.2) "globalizes" the comparison over a solvable convex body $P$: a Frank–Wolfe–type routine yields a *uniform* first-order certificate against every $\mathbf{y} \in P$ without requiring curvature information, a device standard in recent continuous submodular solvers (e.g., (Buchbinder & Feldman, 2024)) and aligned with unified weakly-DR analyses (e.g., (Pedramfar et al., 2023)). Combining this certificate with the weakly-DR property gives a value bound that degrades smoothly with $\gamma$ and exactly matches the DR case at $\gamma = 1$; this is formalized in Lemma 3.1 and Theorem 3.2, and proof of these results are given in Appendix C

**Lemma 3.1.** *Let* $F : [0,1]^n \to \mathbb{R}_{\geq 0}$ *be* $\gamma$*–weakly DR–submodular. If* $\mathbf{x}$ *is a local optimum with respect to a vector* $\mathbf{y}$, *i.e.,*

$$\langle \mathbf{y} - \mathbf{x}, \nabla F(\mathbf{x}) \rangle \leq 0, \tag{13}$$

*then*

$$F(\mathbf{x}) \ \geq \ \frac{\gamma^2 \, F(\mathbf{x} \vee \mathbf{y}) + F(\mathbf{x} \wedge \mathbf{y})}{1 + \gamma^2}. \tag{14}$$

**Theorem 3.2.** *Let* $F : [0,1]^n \to \mathbb{R}_{\geq 0}$ *be a nonnegative, L-smooth function that is* $\gamma$*–weakly DR–submodular. Let* $P \subseteq [0,1]^n$ *be a solvable convex body of diameter* $D$, *and let* $\delta \in (0,1)$. *There is a polynomial time algorithm that outputs* $\mathbf{x} \in P$ *such that, for every* $\mathbf{y} \in P$,

$$F(\mathbf{x}) \geq \frac{\gamma^2 F(\mathbf{x} \vee \mathbf{y}) + F(\mathbf{x} \wedge \mathbf{y})}{1 + \gamma^2} - \frac{\delta \, \gamma}{1 + \gamma^2} \left[ \max_{\mathbf{z} \in P} F(\mathbf{z}) + \frac{LD^2}{2} \right]. \tag{15}$$

In the continuous DR submodular domain, Double–Greedy–type procedures were extended to *DR–submodular* objectives on $[0,1]^n$ (the $\gamma{=}1$ case) by several works (Bian et al., 2019; Chen et al., 2018; Niazadeh et al., 2020), but these analyses still stated a uniform $1/2$ bound and predated the unbalanced

refinement. We generalize this line to the $\gamma$–*weakly DR–submodular* regime $(0 < \gamma \leq 1)$, leveraging the weakly-DR structure introduced for continuous submodular maximization (Bian et al., 2017b;a) and the recent unified weakly-DR perspective (Pedramfar et al., 2023).

Our algorithm retains the Double–Greedy structure but augments it with $\gamma$-*aware* smoothing/thresholding, ensuring that it handles any given $\gamma \in (0, 1]$ robustly; the resulting guarantee interpolates continuously in $\gamma$ and collapses to the classical DR bound at $\gamma = 1$. The corresponding lower bound guarantee is stated in Theorem 3.3, and a detailed proof is provided in Appendix D.

**Theorem 3.3.** *Let $F : [0, 1]^n \to \mathbb{R}_{\geq 0}$ be nonnegative and $\gamma$–weakly DR-submodular for some $\gamma \in (0, 1]$, and fix a parameter $\varepsilon \in (0, 1)$. There exists a polynomial time algorithm that outputs $\mathbf{x} \in [0, 1]^n$ such that*

$$F(\mathbf{x}) \geq \max_{r \geq 0} \frac{\left(2\gamma^{3/2} - 4\varepsilon\gamma^{9/2}\right) r \, F(\mathbf{o}) + F(\mathbf{0}) + r^2 F(\mathbf{1})}{r^2 + 2\gamma^{3/2}r + 1}. \tag{16}$$

When $\gamma = 1$ and $r = 1$ this recovers the canonical $1/2$ approximation, while for many instances one can choose $r \neq 1$ to obtain a strictly better guarantee, in direct analogy to the unbalanced bounds for set functions (Mualem & Feldman, 2022; Qi, 2022).

The unbalanced Double–Greedy guarantee extends directly to axis aligned boxes. Fix an upper bound $\mathbf{x} \in [0, 1]^n$ and consider maximizing $F$ over the box $[\mathbf{0}, \mathbf{x}]$. Define $G : [0, 1]^n \to \mathbb{R}_{\geq 0}$ by $G(\mathbf{a}) := F(\mathbf{x} \odot \mathbf{a})$. By Lemma 2.3, $G$ remains nonnegative and $\gamma$–weakly DR-submodular, so Theorem 3.3 applies to $G$. Translating the output back via $\mathbf{y} := \mathbf{x} \odot \mathbf{a}' \leq \mathbf{x}$ yields the following corollary. We use following corollary as Box Maximization in Algorithm 2.

**Corollary 3.4** (Box maximization). *Let $F : [0, 1]^n \to \mathbb{R}_{\geq 0}$ be nonnegative and $\gamma$–weakly DR-submodular for some $\gamma \in (0, 1]$, let $\mathbf{x} \in [0, 1]^n$, and fix $\varepsilon \in (0, 1)$. There exists a polynomial time algorithm that outputs a vector $\mathbf{y} \in [0, 1]^n$ with $\mathbf{y} \leq \mathbf{x}$ such that, for every fixed $\mathbf{o} \in [0, 1]^n$,*

$$F(\mathbf{y}) \geq \max_{r \geq 0} \frac{\left(2\gamma^{3/2} - 4\varepsilon\,\gamma^{9/2}\right) r \, F(\mathbf{x} \odot \mathbf{o}) + F(\mathbf{0}) + r^2 F(\mathbf{x})}{r^2 + 2\gamma^{3/2}r + 1}. \tag{17}$$

*Proof.* Fix $\mathbf{x} \in [0, 1]^n$ and define the restricted objective

$$G(\mathbf{a}) := F(\mathbf{x} \odot \mathbf{a}) \qquad \text{for all } \mathbf{a} \in [0, 1]^n. \tag{18}$$

By Lemma 2.3, $G$ is nonnegative and $\gamma$–weakly DR-submodular. Applying Theorem 3.3 to $G$ (with the same $\varepsilon \in (0, 1)$) yields some $\mathbf{a}' \in [0, 1]^n$ such that

$$G(\mathbf{a}') \geq \max_{r \geq 0} \frac{\left(2\gamma^{3/2} - 4\varepsilon\,\gamma^{9/2}\right) r \, G(\mathbf{o}) + G(\mathbf{0}) + r^2 \, G(\mathbf{1})}{r^2 + 2\gamma^{3/2}r + 1}. \tag{19}$$

From equation 18, we have

$$G(\mathbf{o}) = F(\mathbf{x} \odot \mathbf{o}), \qquad G(\mathbf{0}) = F(\mathbf{0}), \qquad G(\mathbf{1}) = F(\mathbf{x}).$$

Substituting these identities into equation 19 gives

$$G(\mathbf{a}') \geq \max_{r \geq 0} \frac{\left(2\gamma^{3/2} - 4\varepsilon\,\gamma^{9/2}\right) r \, F(\mathbf{x} \odot \mathbf{o}) + F(\mathbf{0}) + r^2 \, F(\mathbf{x})}{r^2 + 2\gamma^{3/2}r + 1}. \tag{20}$$

Define $\mathbf{y} := \mathbf{x} \odot \mathbf{a}'$. Then $\mathbf{y} \leq \mathbf{x}$ coordinate wise and, by equation 18,

$$F(\mathbf{y}) = F(\mathbf{x} \odot \mathbf{a}') = G(\mathbf{a}'). \tag{21}$$

Combining equation 20 and equation 21 yields the claimed bound equation 17. $\qquad \square$

# 4 Main Algorithm and Results

In this section, we present our main result together with the algorithm that achieves it. Our approach is recursive and hinges on a core subroutine invoked at every level of recursion: the $\gamma$–*Frank Wolfe Guided Measured Continuous Greedy* ($\gamma$-**FWG**). We describe $\gamma$-**FWG** and establish its guarantees in Section 4.1. Building on this component, Section 4.2 introduces the full recursive algorithm, and Section 4.3 proves our main theorem.

## 4.1 $\gamma$-FWG Algorithm

We develop a measured continuous greedy method, steered by Frank Wolfe directions and explicitly tuned by $\gamma$, to operate in the $\gamma$–weakly DR setting. The algorithm is explicitly $\gamma$-parameterized, so it works for any $\gamma \in (0,1]$ and reduces to the classical DR case when $\gamma = 1$ (Buchbinder & Feldman, 2024). For clarity, in the description of Algorithm 1 we assume that $\delta^{-1}$ is an integer and that $\delta \leq \varepsilon$ (which lets us set $m = \delta^{-1}$). If these conditions do not hold, we reduce $\delta$ to $1/\lceil 1/\min\{\delta,\varepsilon\}\rceil$ without affecting the analysis. Also we define $\beta := \frac{\gamma^2 \delta}{1-\delta+\gamma^2\delta}$. Since the algorithm does not know the values $F(\mathbf{o})$, $F(\mathbf{z} \odot \mathbf{o})$, and $F(\mathbf{z} \oplus \mathbf{o})$, we rely on the following guessing lemma; its proof uses standard guessing arguments and proof of this lemma is given in Appendix E.

**Lemma 4.1.** *Let $F : [0,1]^n \to \mathbb{R}_{\geq 0}$ be nonnegative and $\gamma$-weakly DR-submodular for some $0 < \gamma \leq 1$, and let $P \subseteq [0,1]^n$ be down-closed. There exists a constant size (depending only on $\varepsilon$ and $\gamma$) set of triples $\mathcal{G} \subseteq \mathbb{R}_{\geq 0}^3$ such that $\mathcal{G}$ contains a triple $(g, g_\odot, g_\oplus)$ with*

$$(1 - \varepsilon) F(\mathbf{o}) \leq g \leq F(\mathbf{o}), \tag{22a}$$

$$F(\mathbf{z} \odot \mathbf{o}) - \varepsilon g \leq g_\odot \leq F(\mathbf{z} \odot \mathbf{o}), \tag{22b}$$

$$F(\mathbf{z} \oplus \mathbf{o}) - \varepsilon g \leq g_\oplus \leq F(\mathbf{z} \oplus \mathbf{o}). \tag{22c}$$

Therefore, by trying all triples in $\mathcal{G}$, we can act as if Algorithm 1 is given valid surrogates $g$, $g_\odot$, and $g_\oplus$ that meet these bounds. For convenience, we first define the threshold functions. For $i \in \{0, 1, \ldots, \delta^{-1} - 1\}$, define

$$v_1(i) := \left[(1 - \beta)^i + \frac{1-(1-\beta)^i - 2\varepsilon}{\gamma}\right] g - \frac{1}{\gamma} g_\odot - \frac{1-(1-\beta)^i}{\gamma} g_\oplus \tag{23}$$

$$v_2(i) := (1 - \beta)^i \left[\left(\frac{(1-\beta)^{-i_s}}{\gamma} - \left(1 + \frac{3}{\gamma}\right)\varepsilon + 1 - \frac{1}{\gamma}\right) g - \left(\frac{(1-\beta)^{-i_s}}{\gamma} - \frac{1}{\gamma} - \beta\,(i - i_s)\right) g_\oplus\right] \tag{24}$$

For a fixed $\gamma$, the parameters $t_s$ and $\varepsilon$ are constants, and hence the running time of Algorithm 1 is $\text{Poly}(n, \delta^{-1})$. Algorithm 1 therefore guarantees the following performance on its output. The proof of this theorem is provided in Appendix E.

**Theorem 4.2.** $\gamma$-FWG *takes as input a nonnegative, $L$-smooth, $\gamma$–weakly DR-submodular function $F : [0,1]^n \to \mathbb{R}_{\geq 0}$, a meta solvable down-closed convex body $P \subseteq [0,1]^n$ of diameter $D$, a vector $\mathbf{z} \in P$, and parameters $t_s \in (0,1)$, $\varepsilon \in (0, 1/2)$, and $\delta \in (0,1)$. Given this input, $\gamma$-FWG outputs a vector $\mathbf{y} \in P$ and vectors $\mathbf{x}(1), \ldots, \mathbf{x}(m) \in P$ for some $m = O(\delta^{-1} + \varepsilon^{-1})$, such that at least one of the following holds.*

*1.*

$$F(\mathbf{y}) \geq A_\gamma(t_s) F(\mathbf{o}) + B_\gamma(t_s) F(\mathbf{z} \odot \mathbf{o}) + C_\gamma(t_s) F(\mathbf{z} \oplus \mathbf{o}) - \delta L D^2. \tag{25}$$

*2. There exists $i \in [m]$ such that*

$$F\big(\mathbf{x}(i) \oplus \mathbf{o}\big) \leq F(\mathbf{z} \oplus \mathbf{o}) - \varepsilon F(\mathbf{o}), \tag{26}$$

*and the point $\mathbf{x}(i)$ satisfies the $\gamma$–weakly DR local–value bound*

$$F\big(\mathbf{x}(i)\big) \geq \frac{\gamma^2 F\big(\mathbf{x}(i) \vee \mathbf{o}\big) + F\big(\mathbf{x}(i) \wedge \mathbf{o}\big)}{1 + \gamma^2} - \frac{\delta\,\gamma}{1 + \gamma^2}\left(\max_{\mathbf{y}' \in Q(i)} F(\mathbf{y}') + \tfrac{1}{2} L D^2\right). \tag{27}$$

*i.e., $\mathbf{x}(i)$ is an approximate local maximum with respect to $\mathbf{o}$ under the $\gamma$–weakly DR guarantee and the Frank–Wolfe certificate over $Q(i)$.*

---

**Algorithm 1** $\gamma$-FWG$(F, P, \mathbf{z}, \gamma, t_s, \varepsilon, \delta)$

---

1: **Input:** nonnegative $L$-smooth $\gamma$-weakly DR-submodular $F : [0,1]^n \to \mathbb{R}_{\geq 0}$; meta solvable down-closed $P \subseteq [0,1]^n$; $\mathbf{z} \in P$; $\gamma \in (0,1]$; parameters $t_s \in (0,1)$, $\varepsilon \in (0, 1/2)$, $\delta \in (0,1)$.

2: $i_s \leftarrow \lceil t_s/\delta \rceil$

3: **for** $i = 0$ to $\delta^{-1} - 1$ **do**

4: $\quad v(i) := \begin{cases} v_1(i), & \text{if } i \leq i_s, \\ v_2(i), & \text{if } i \geq i_s. \end{cases}$

5: $\quad \mathbf{z}(i) \leftarrow \begin{cases} \mathbf{z}, & \text{if } i < i_s, \\ \mathbf{0}, & \text{if } i \geq i_s \end{cases}$

6: **end for**

7: $\mathbf{y}(0) \leftarrow \mathbf{0}$

8: **for** $i = 1$ to $\delta^{-1}$ **do**

9: $\quad \mathbf{w}(i) \leftarrow \big(\mathbf{1} - \mathbf{y}(i-1) - \mathbf{z}(i-1)\big) \odot \nabla F\big(\mathbf{y}(i-1)\big)$

10: $\quad Q(i) \leftarrow \big\{ \mathbf{x} \in P \mid \langle \mathbf{w}(i), \mathbf{x} \rangle \geq \gamma\big(v(i-1) - F(\mathbf{y}(i-1))\big) \big\}$

11: $\quad$ Use Theorem 3.2 to compute an approximate local maximum $\mathbf{x}(i)$ of $Q(i)$ (if $Q(i) = \varnothing$, set $\mathbf{x}(i)$ to an arbitrary vector in $P$)

12: $\quad \mathbf{y}(i) \leftarrow \mathbf{y}(i-1) + \delta\big(\mathbf{1} - \mathbf{y}(i-1) - \mathbf{z}(i-1)\big) \odot \mathbf{x}(i)$

13: **end for**

14: **return** $\mathbf{y}(\delta^{-1})$ and the sequence $\mathbf{x}(1), \mathbf{x}(2), \ldots, \mathbf{x}(\delta^{-1})$

---

*Here the $\gamma$–dependent coefficients are*

$$A_\gamma(t_s) := -\frac{e^{\gamma t_s - \gamma}}{1 - \gamma} + \frac{e^{-\gamma^2}}{\gamma(1-\gamma)}\Big(e^{\gamma^2 t_s} - (1 - \gamma)\Big) \; - \; O(\varepsilon) \tag{28a}$$

$$B_\gamma(t_s) := \frac{e^{-\gamma} - e^{\gamma t_s - \gamma}}{\gamma} \tag{28b}$$

$$\begin{aligned}
C_\gamma(t_s) := {} & \frac{e^{\gamma^2 t_s} - 1}{\gamma(1-\gamma)}\Big(e^{-\gamma(1-t_s) - \gamma^2 t_s} - e^{-\gamma^2}\Big) \\
& + \frac{e^{-\gamma(1-t_s)}}{\gamma}\left[\big(e^{-\gamma t_s} - 1\big) + \frac{e^{-\gamma^2 t_s} - e^{-\gamma t_s}}{1 - \gamma}\right] \\
& + e^{-\gamma(1-t_s) - \gamma^2 t_s}\left[\frac{\gamma^2}{1-\gamma}(1 - t_s)\, e^{\gamma(1-\gamma)(1-t_s)}\right. \\
& \qquad \left. + \frac{\gamma}{(1-\gamma)^2}\Big(1 - e^{\gamma(1-\gamma)(1-t_s)}\Big)\right].
\end{aligned} \tag{28c}$$

## 4.2 Main Algorithm

In this section we describe our main algorithm and analyze the recursive framework that establishes our main result (Theorem 4.5). The core building block is our new procedure $\gamma$-*weakly Frank-Wolfe Guided Measured Continuous Greedy* ($\gamma$-FWG), introduced in Section 4.1. We classify an execution of $\gamma$-FWG as *successful* if the first outcome holds (i.e., when $F(\mathbf{y})$ attains the "large value" case). Otherwise, the execution is deemed *unsuccessful*. With this terminology in place, we now describe the main recursive driver, *Algorithm 2*, which we use to prove Theorem 4.5. In addition to the parameters appearing in Theorem 4.5, Algorithm 2 takes two auxiliary inputs: $\varepsilon \in (0, 1/2)$ and $t_s \in (0,1)$. These are forwarded unchanged to every call to $\gamma$-FWG.

Algorithm 2 runs for $L = 1 + \left\lceil \frac{1+\gamma}{\varepsilon\gamma} \right\rceil$ recursion levels, indexed by $i$. At level 1, it finds an approximate local maximizer $\mathbf{z}(0) \in P$ and, from this seed, runs BOX MAXIMIZATION and $\gamma$-FWG, producing $\mathbf{z}'$, $\mathbf{y}$, and a batch $\{\mathbf{x}(1), \ldots, \mathbf{x}(m)\}$. For each subsequent level $i = 2, \ldots, L$, every candidate $\mathbf{x}(\cdot)$ emitted at level $i - 1$ becomes a new seed $\mathbf{z}$; the same two subroutines are applied to each seed, yielding fresh outputs $\mathbf{z}'$, $\mathbf{y}$, and

$\{\mathbf{x}(1), \ldots, \mathbf{x}(m)\}$. After all levels complete, the algorithm returns the vector with the largest objective value among all $\mathbf{z}'$, $\mathbf{y}$, and $\mathbf{x}(\cdot)$ produced at any level.

---

**Algorithm 2** Main Algorithm$(F, P, \gamma, t_s, \varepsilon, \delta)$

---

1: Let $\mathbf{z}(0)$ be an local maximum in $P$ obtained via the Theorem 3.3.
2: Execute MAIN-RECURSIVE$(F, P, \gamma, t_s, \mathbf{z}(0), \varepsilon, \delta, 1)$.
3: **function** MAIN-RECURSIVE$(F, P, \gamma, t_s, \mathbf{z}, \varepsilon, \delta, i)$
4:      $\mathbf{z}' = $ Box-Maximization$(\mathbf{z})$
5:      Let $(\mathbf{y}, \mathbf{x}(1), \ldots, \mathbf{x}(m)) = \gamma$-FWG$(F, P, \mathbf{z}, \gamma, t_s, \varepsilon)$ (Algorithm 1)
6:      **if** $i < L$ **then**
7:          **for** $j = 1$ **to** $m$ **do**
8:              $\mathbf{y}(j) = $ MAIN-RECURSIVE$(F, P, \gamma, \mathbf{x}(j), t_s, \varepsilon, \delta, i+1)$
9:          **end for**
10:     **end if**
11:     **return** the vector maximizing $F$ among $\mathbf{z}'$, $\mathbf{y}$ and the vectors in $\{\mathbf{y}(j) \mid j \in [m]\}$.
12: **end function**

---

Observe that, for fixed $\gamma$, the number of recursive calls executed by Algorithm 1 is $O(m^L) = (\delta^{-1} + \varepsilon^{-1})^{O(1/\varepsilon)}$. For any constant $\varepsilon$, this quantity is polynomial in $\delta^{-1}$. Moreover, each individual recursive call runs in time polynomial in $\delta^{-1}$ and $n$. Therefore, the overall running time of Algorithm 1 is polynomial in $\delta^{-1}$ and $n$.

We say that a recursive call of Algorithm 2 is *successful* if its internal run of $\gamma$-FWG is successful. Section 4.3 shows that Algorithm 2 performs sufficiently many recursive invocations to ensure that at least one call is successful and, moreover, that it obtains a vector $\mathbf{z}$ which is an approximate local maximizer with respect to $\mathbf{o}$. From such a call, the analysis further proves that either the accompanying vector $\mathbf{z}'$ or the vector $\mathbf{y}$ satisfies the performance guarantee stated in Theorem 4.5.

### 4.3 The Main Result

In the recursion tree of Algorithm 2, we focus on one designated path of *heir* calls. Along this path, the "fallback" guarantee of $\gamma$-FWG is passed forward at each level. A recursive call $\mathcal{C}$ is an *heir* if either

1. $\mathcal{C}$ is the unique level-1 call (i.e., the first invocation in the recursion), or

2. $\mathcal{C}$ was invoked by another heir call $\mathcal{C}_p$ that is *unsuccessful*, and its input seed $\mathbf{z}$ equals $\mathbf{x}(j^\star)$ for some index $j^\star$ that satisfies the second outcome of Theorem 4.2 for the $\gamma$-FWG run inside $\mathcal{C}_p$.

Intuitively, if an heir call $\mathcal{C}_p$ is *unsuccessful* (its run of $\gamma$-FWG does not return a large value $\mathbf{y}$), then Theorem 4.2 guarantees an index $j^\star \in [m]$ for which $\mathbf{x}(j^\star)$ satisfies a first order $\gamma$–weakly DR certificate. We then set the seed of the next heir on the designated path to $\mathbf{z} \leftarrow \mathbf{x}(j^\star)$. The following observation records the certificate's invariant, which we subsequently combine to obtain the final guarantee.

**Observation 1.** Fix $\gamma \in (0, 1]$. Every heir recursive call in Algorithm 2 receives a seed $\mathbf{z} \in P$ satisfying

$$F(\mathbf{z}) \geq \frac{\gamma^2 F(\mathbf{z} \vee \mathbf{o}) + F(\mathbf{z} \wedge \mathbf{o})}{1 + \gamma^2} - O(\varepsilon) F(\mathbf{o}) - O(\delta L D^2). \tag{29}$$

*Proof.* We argue by cases on the recursion level that produces $\mathbf{z}$.

**Case 1: $\mathbf{z}$ comes from a later level.** Here $\mathbf{z} = \mathbf{x}(i)$ for some index $i$ returned by the previous $\gamma$-FWG call, where the "successful" bound equation 25 did not apply. Hence equation 27 holds:

$$F(\mathbf{x}(i)) \geq \frac{\gamma^2 F(\mathbf{x}(i) \vee \mathbf{o}) + F(\mathbf{x}(i) \wedge \mathbf{o})}{1 + \gamma^2} - \frac{\delta \gamma}{1 + \gamma^2} \left( \max_{\mathbf{y}' \in Q(i)} F(\mathbf{y}') + \tfrac{1}{2} L D^2 \right). \tag{30}$$

Since $Q(i) \subseteq P$, we have $\max_{\mathbf{y}' \in Q(i)} F(\mathbf{y}') \leq \max_{\mathbf{y}' \in P} F(\mathbf{y}') \leq F(\mathbf{o})$, and substituting this into equation 30 with $\mathbf{z} = \mathbf{x}(i)$ yields

$$F(\mathbf{z}) \; \geq \; \frac{\gamma^2 F(\mathbf{z} \vee \mathbf{o}) + F(\mathbf{z} \wedge \mathbf{o})}{1 + \gamma^2} - \frac{\delta \, \gamma}{1 + \gamma^2} \left( F(\mathbf{o}) + \tfrac{1}{2} LD^2 \right), \tag{31}$$

which matches the invariant equation 29 up to the stated $O(\delta) \, F(\mathbf{o})$ and $O(\delta \, LD^2)$ terms.

**Case 2: $\mathbf{z}$** is produced at the first recursion level. Here $\mathbf{z}$ is the output of the weakly-DR local maximization routine from Theorem 3.2 with accuracy parameter $\eta \; := \; \min\{\varepsilon, \delta\}$. Applying Theorem 3.2 with $\mathbf{y} = \mathbf{o}$ and using $\max_{\mathbf{y}' \in P} F(\mathbf{y}') \leq F(\mathbf{o})$ gives

$$F(\mathbf{z}) \; \geq \; \frac{\gamma^2 F(\mathbf{z} \vee \mathbf{o}) + F(\mathbf{z} \wedge \mathbf{o})}{1 + \gamma^2} \; - \; \frac{\eta \, \gamma}{1 + \gamma^2} \left( F(\mathbf{o}) + \tfrac{1}{2} LD^2 \right). \tag{32}$$

Since $\eta \leq \varepsilon$ and $\eta \leq \delta$ by definition of $\eta$, the error term in equation 32 is again of the form $O(\varepsilon) \, F(\mathbf{o}) + O(\delta \, LD^2)$, yielding equation 29.

Both cases establish equation 29, it completes the proof. $\qquad \square$

Before proving that a successful heir exists (Corollary 4.3), we note a simple measure that drops at each level of recursion.

**Observation 2.** Assume every heir recursive call of Algorithm 2 is unsuccessful (in the sense of Theorem 4.2). Then, for every recursion level $i \geq 1$, there exists an heir call at level $i$ that receives a seed $\mathbf{z}$ with

$$F(\mathbf{z} \oplus \mathbf{o}) \; \leq \; F\big(\mathbf{z}(0) \oplus \mathbf{o}\big) \; - \; \varepsilon \, (i - 1) \, F(\mathbf{o}). \tag{33}$$

*Proof.* We argue by induction on the level $i$.

*Base case $(i = 1)$.* The unique level-1 heir call receives $\mathbf{z} = \mathbf{z}(0)$. Hence

$$F(\mathbf{z} \oplus \mathbf{o}) \; = \; F\big(\mathbf{z}(0) \oplus \mathbf{o}\big) \; \leq \; F\big(\mathbf{z}(0) \oplus \mathbf{o}\big) - \varepsilon \cdot 0 \cdot F(\mathbf{o}), \tag{34}$$

which is exactly equation 33 with $i = 1$.

*Inductive step.* Assume the statement holds for level $i - 1 \geq 1$; i.e., there is an heir call on level $i - 1$ with seed $\mathbf{z}$ such that

$$F(\mathbf{z} \oplus \mathbf{o}) \; \leq \; F\big(\mathbf{z}(0) \oplus \mathbf{o}\big) \; - \; \varepsilon \, (i - 2) \, F(\mathbf{o}). \tag{35}$$

By assumption, this heir call is unsuccessful. Therefore, the "gap" outcome equation 26 of Theorem 4.2 applies to its internal run of $\gamma$-FWG, and hence there exists $j^\star \in [m]$ such that

$$F\big(\mathbf{x}(j^\star) \oplus \mathbf{o}\big) \; \leq \; F(\mathbf{z} \oplus \mathbf{o}) \; - \; \varepsilon \, F(\mathbf{o}). \tag{36}$$

Combining equation 35 and equation 36 gives

$$F\big(\mathbf{x}(j^\star) \oplus \mathbf{o}\big) \; \leq \; F\big(\mathbf{z}(0) \oplus \mathbf{o}\big) \; - \; \varepsilon \, (i - 1) \, F(\mathbf{o}). \tag{37}$$

By the definition of heirs, the child call at level $i$ seeded with $\mathbf{z} \leftarrow \mathbf{x}(j^\star)$ is itself an heir call and satisfies equation 37, which is precisely equation 33 for level $i$. $\qquad \square$

The weakly-DR specifics (the $\gamma$-aware local value bound and Frank-Wolfe certificate) only affect the *quality* guarantee for the seed $\mathbf{x}(j^\star)$, not the *descent amount* on $F(\cdot \oplus o)$. Thus the $\varepsilon$ per level decrease remains identical to the DR case (Buchbinder & Feldman, 2024), while $\gamma$ enters later in the value lower bounds used to conclude the analysis.

**Corollary 4.3.** *Some recursive call of Algorithm 2 is a successful heir.*

*Proof.* Assume, toward a contradiction, that no recursive call is a successful heir. By Observation 2, at level

$$i := 1 + \left\lceil \frac{\gamma+1}{\gamma\varepsilon} \right\rceil \tag{38}$$

there exists an heir call with seed $\mathbf{z}$ such that

$$F(\mathbf{z} \oplus \mathbf{o}) \overset{(a)}{\leq} F(\mathbf{z}(0) \oplus \mathbf{o}) - \varepsilon(i-1)F(\mathbf{o})$$

$$\overset{(b)}{\leq} F(\mathbf{z}(0) \oplus \mathbf{o}) - \left(1 + \frac{1}{\gamma}\right)F(\mathbf{o}), \tag{39}$$

where (a) follows from Observation 2 applied at level $i$, and (b) uses $i - 1 \geq (\gamma+1)/(\gamma\varepsilon)$ from equation 38.

Since this call is (by assumption) also unsuccessful, the gap alternative equation 26 of Theorem 4.2 applies, yielding some $j$ with

$$F(\mathbf{x}(j) \oplus \mathbf{o}) \leq F(\mathbf{z} \oplus \mathbf{o}) - \varepsilon F(\mathbf{o}). \tag{40}$$

Combining equation 39 and equation 40 gives

$$F(\mathbf{x}(j) \oplus \mathbf{o}) \leq F(\mathbf{z}(0) \oplus \mathbf{o}) - \left(1 + \frac{1}{\gamma} + \varepsilon\right)F(\mathbf{o}). \tag{41}$$

By nonnegativity of $F$, we have $F(\mathbf{x}(j) \oplus \mathbf{o}) \geq 0$, so equation 41 implies

$$F(\mathbf{z}(0) \oplus \mathbf{o}) - F(\mathbf{o}) \geq \left(\frac{1}{\gamma} + \varepsilon\right)F(\mathbf{o}) > \frac{1}{\gamma}F(\mathbf{o}). \tag{42}$$

On the other hand, by $\gamma$–weakly DR property we have

$$F(\mathbf{z}(0) \oplus \mathbf{o}) - F(\mathbf{o}) \overset{(c)}{\leq} \frac{1}{\gamma}\Big(F\big(\mathbf{z}(0) \odot (\mathbf{1} - \mathbf{o})\big) - F(\mathbf{0})\Big)$$

$$\overset{(d)}{\leq} \frac{1}{\gamma} F\big(\mathbf{z}(0) \odot (\mathbf{1} - \mathbf{o})\big) \tag{43}$$

where (c) follows from the $\gamma$–weakly DR definition, and (d) uses $F(\mathbf{0}) \geq 0$ (nonnegativity). Combining equation 42 and equation 43 yields

$$F(\mathbf{o}) < F\big(\mathbf{z}(0) \odot (\mathbf{1} - \mathbf{o})\big). \tag{44}$$

Since $P$ is down-closed and $\mathbf{z}(0) \in P$, we have $\mathbf{z}(0) \odot (\mathbf{1} - \mathbf{o}) \leq \mathbf{z}(0)$ coordinate-wise, hence $\mathbf{z}(0) \odot (\mathbf{1} - \mathbf{o}) \in P$. The strict improvement in equation 44 contradicts the optimality of $\mathbf{o}$ over $P$. Therefore, our assumption was false, and some recursive call must be a successful heir. $\qquad\square$

Consider any successful heir call within Algorithm 2. Denote its input seed by $\mathbf{z}^\star$, and let $\mathbf{y}^\star$ be the high value solution returned by $\gamma$-FWG, while $\mathbf{z}'^\star$ is the child seed produced during the same call. Invoking Theorem 4.2 (1) yields

$$F(\mathbf{y}^\star) \geq A_\gamma(t_s)F(\mathbf{o}) + B_\gamma(t_s)F(\mathbf{z}^\star \odot \mathbf{o}) + C_\gamma(t_s)F(\mathbf{z}^\star \oplus \mathbf{o}) - \delta LD^2. \tag{45}$$

For the ensuing analysis of a successful heir, we also need a companion lower bound for $F(\mathbf{z}'^\star)$; this is provided by the next lemma.

**Lemma 4.4.** *Let $F : [0,1]^n \to \mathbb{R}_{\geq 0}$ be nonnegative, $L$-smooth, and $\gamma$-weakly DR-submodular for some $\gamma \in (0,1]$. Let $\mathbf{z}^* \in [0,1]^n$ be the incumbent vector provided to the heir recursive call, and let $\mathbf{z}'^* \leq \mathbf{z}^*$ be the output of Corollary 3.4 (run on the box $[0, \mathbf{z}^*]$ with error $\varepsilon$). Then*

$$F(\mathbf{z}'^*) \geq \max_{r \geq 0} \frac{\left(2\gamma^{3/2}r + \frac{\gamma}{1+\gamma^2}r^2\right)F(\mathbf{z}^* \odot \mathbf{o}) + \frac{\gamma^2}{1+\gamma^2}r^2 F(\mathbf{z}^* \oplus \mathbf{o})}{r^2 + 2\gamma^{3/2}r + 1} - O(\varepsilon)F(\mathbf{o}) - O(\delta LD^2). \tag{46}$$

*Proof.* Applying Corollary 3.4 to the box $[0, \mathbf{z}^*]$ (with error $\varepsilon$) gives

$$F(\mathbf{z}'^*) \geq \max_{r \geq 0} \frac{(2\gamma^{3/2} - 4\varepsilon\,\gamma^{9/2})rF(\mathbf{z}^* \odot \mathbf{o}) + F(\mathbf{0}) + r^2\,F(\mathbf{z}^*)}{r^2 + 2\gamma^{3/2}r + 1}. \tag{47}$$

Since $F(\mathbf{0}) \geq 0$, dropping the nonnegative $F(\mathbf{0})$ can only decrease the right hand side, therefore we get (a);

$$\begin{aligned}
F(\mathbf{z}'^*) &\overset{(a)}{\geq} \max_{r \geq 0} \frac{(2\gamma^{3/2} - 4\varepsilon\,\gamma^{9/2})\,r\,F(\mathbf{z}^* \odot \mathbf{o}) + r^2\,F(\mathbf{z}^*)}{r^2 + 2\gamma^{3/2}r + 1} \\
&\overset{(b)}{\geq} \max_{r \geq 0} \frac{2\gamma^{3/2}\,r\,F(\mathbf{z}^* \odot \mathbf{o}) + r^2\,F(\mathbf{z}^*)}{r^2 + 2\gamma^{3/2}r + 1} - O(\varepsilon)\,F(\mathbf{o}),
\end{aligned} \tag{48}$$

where (b) uses $F(\mathbf{z}^* \odot \mathbf{o}) \leq F(\mathbf{o})$ and the bound $\frac{r}{r^2 + 2\gamma^{3/2}r + 1} \leq 1$, so the negative perturbation term $-4\varepsilon\,\gamma^{9/2}\,\frac{r}{r^2 + 2\gamma^{3/2}r + 1}\,F(\mathbf{z}^* \odot \mathbf{o})$ is at worst $-O(\varepsilon)\,F(\mathbf{o})$ after maximizing over $r \geq 0$.

Next, since $\mathbf{z}^*$ is the seed of an heir recursive call, Observation 1 yields

$$F(\mathbf{z}^*) \geq \frac{\gamma^2 F(\mathbf{z}^* \vee \mathbf{o}) + F(\mathbf{z}^* \wedge \mathbf{o})}{1 + \gamma^2} - O(\varepsilon)F(\mathbf{o}) - O(\delta L D^2). \tag{49}$$

Because $F \geq 0$ and $\gamma \leq 1$, we also have $F(\mathbf{z}^* \wedge \mathbf{o}) \geq \gamma\,F(\mathbf{z}^* \wedge \mathbf{o})$; combining this with the weakly-DR "swap" inequality

$$\gamma\,F(\mathbf{x} \vee \mathbf{y}) + F(\mathbf{x} \wedge \mathbf{y}) \geq \gamma\,F(\mathbf{x} \oplus \mathbf{y}) + F(\mathbf{x} \odot \mathbf{y}) \quad (\forall \mathbf{x}, \mathbf{y} \in [0,1]^n), \tag{50}$$

we get the refined lower bound

$$F(\mathbf{z}^*) \geq \frac{\gamma^2 F(\mathbf{z}^* \oplus \mathbf{o}) + \gamma\,F(\mathbf{z}^* \odot \mathbf{o})}{1 + \gamma^2} - O(\varepsilon)F(\mathbf{o}) - O(\delta L D^2). \tag{51}$$

Substituting equation 51 into equation 48, and noting that $\frac{r^2}{r^2 + 2\gamma^{3/2}r + 1} \leq 1$, preserves the additive error $-O(\varepsilon)\,F(\mathbf{o}) - O(\delta L D^2)$ and yields coefficients

$$\frac{2\gamma^{3/2}r}{r^2 + 2\gamma^{3/2}r + 1} + \frac{\gamma}{1 + \gamma^2} \cdot \frac{r^2}{r^2 + 2\gamma^{3/2}r + 1} \quad \text{for } F(\mathbf{z}^* \odot \mathbf{o}),$$

$$\frac{\gamma^2}{1 + \gamma^2} \cdot \frac{r^2}{r^2 + 2\gamma^{3/2}r + 1} \quad \text{for } F(\mathbf{z}^* \oplus \mathbf{o}),$$

which is exactly the claimed form. $\qquad\square$

We have established two certified lower bounds for any *successful* heir call: one for the Frank Wolfe guided output $\mathbf{y}^\star$ (Theorem 4.2(1)) and one for the box-restricted child $\mathbf{z}'^\star$ (Lemma 4.4). Since the algorithm returns the better of these two values, *any* convex combination of the two bounds remains a valid lower bound on the algorithm's output. Let $\alpha \in [0,1]$ be the mixing parameter. Putting the two bounds into a common form and combining them gives, for any $r \geq 0$ and $t_s \in (0,1)$,

$$\begin{aligned}
F(\text{ALG}) \geq{}& (1-\alpha)\,A_\gamma(t_s)\,F(\mathbf{o}) + \left[(1-\alpha)\,B_\gamma(t_s) + \alpha\,D_\gamma(r)\right]F(\mathbf{z}^\star \odot \mathbf{o}) \\
&+ \left[(1-\alpha)\,C_\gamma(t_s) + \alpha\,E_\gamma(r)\right]F(\mathbf{z}^\star \oplus \mathbf{o}) - O(\varepsilon)\,F(\mathbf{o}) - O(\delta\,L D^2),
\end{aligned} \tag{52}$$

where

$$D_\gamma(r) := \frac{2\gamma^{3/2}\,r + \frac{\gamma}{1+\gamma^2}\,r^2}{r^2 + 2\gamma^{3/2}r + 1}, \qquad E_\gamma(r) := \frac{\frac{\gamma^2}{1+\gamma^2}\,r^2}{r^2 + 2\gamma^{3/2}r + 1}. \tag{53}$$

To extract a *pure* multiplicative factor in front of $F(\mathbf{o})$, we choose parameters so that the coefficients multiplying $F(\mathbf{z}^\star \odot \mathbf{o})$ and $F(\mathbf{z}^\star \oplus \mathbf{o})$ are nonnegative, allowing these terms to be dropped by nonnegativity of $F$. Define the feasible set

$$\mathcal{F}_\gamma := \Big\{ (\alpha, r, t_s) \in [0,1] \times [0,\infty) \times (0,1) \ : $$

$$(1-\alpha)B_\gamma(t_s) + \alpha D_\gamma(r) \geq 0, (1-\alpha)C_\gamma(t_s) + \alpha E_\gamma(r) \geq 0 \Big\}. \tag{54}$$

For any $(\alpha, r, t_s) \in \mathcal{F}_\gamma$, inequality equation 52 yields

$$F(\mathrm{ALG}) \ \geq \ \Big[ (1-\alpha)\, A_\gamma(t_s) \ - \ O(\varepsilon) \Big] F(\mathbf{o}) \ - \ O(\delta\, LD^2). \tag{55}$$

This motivates optimizing the leading factor:

$$\Phi_\gamma \ := \ \max_{(\alpha, r, t_s) \in \mathcal{F}_\gamma} \ (1-\alpha)\, A_\gamma(t_s). \tag{56}$$

Our approximation guarantee $\Phi_\gamma$ is defined as the optimal value of the maximization problem in equation 56. This is an optimization over only three scalar parameters $(\alpha, r, t_s)$ with simple linear feasibility constraints in equation 54, so for any fixed $\gamma$ the problem is easy to solve numerically. In particular, even though $\Phi_\gamma$ does not appear to admit a closed form expression as a function of $\gamma$, it can be computed to any desired accuracy in polynomial time (in the inverse of the discretization step) by a direct grid search.

For each fixed $\gamma$, we solve equation 56 by performing an explicit grid search over $(r, t_s)$ on a bounded domain $r \in [0, r_{\max}]$, $t_s \in (0,1)$, and then optimizing over $\alpha$ in closed form using the linear constraints in equation 54. For every grid point $(r, t_s)$, we determine the interval of feasible $\alpha \in [0,1]$ for which both inequalities in equation 54 hold, and then choose the endpoint of this interval that maximizes $(1-\alpha)A_\gamma(t_s)$. This yields a candidate triple $(\alpha, r, t_s)$ and a corresponding candidate value of $\Phi_\gamma$, and we keep the best one over all grid points. The running time is polynomial in the inverse grid step. This is precisely the procedure implemented in our Python code to generate Fig. 1 and the parameter table (we used $r_{\max} = 10$, since larger values of $r_{\max}$ did not change the outcome).

*Boundary case $\gamma = 1$.* Several coefficients (e.g., $A_\gamma, B_\gamma, C_\gamma$) contain factors of $(1-\gamma)$ in the denominator; consequently, the expressions inside equation 52 may exhibit an apparent $0/0$ form as $\gamma \to 1$. We interpret all such terms by taking their continuous limits, and evaluate via L'Hôpital's rule where needed. Substituting these limits into equation 52 yields the DR (i.e., $\gamma = 1$) specialization of our mixture bound, and optimizing equation 56 at $\gamma = 1$ reproduces the current best DR guarantee (0.401) of Buchbinder and Feldman (Buchbinder & Feldman, 2024).

**What the optimization achieves.** The optimized guarantee $\Phi_\gamma$ (a) exactly matches the current best DR constant at $\gamma = 1$, and (b) strictly improves on the non-monotone weakly-DR baseline $\kappa(\gamma) = \gamma e^{-\gamma}$ for all $\gamma \in (0,1)$. Intuitively, the mixture balances Frank Wolfe guided progress with box restricted improvements through $(\alpha, r, t_s)$, certifying the best factor per $\gamma$.

**Theorem 4.5.** *Fix $\gamma \in (0,1]$ and $\delta \in (0,1)$. Let $F : [0,1]^n \to \mathbb{R}_{\geq 0}$ be a nonnegative, L-smooth, $\gamma$-weakly DR-submodular function, and let $P \subseteq [0,1]^n$ be a down-closed, meta-solvable convex body of diameter $D$. There exists a polynomial time algorithm that returns a point $\mathbf{x} \in P$ such that*

$$F(\mathbf{x}) \ \geq \ \Phi_\gamma \cdot \max_{\mathbf{y} \in P} F(\mathbf{y}) \ - \ O\big(\delta\, D^2 L\big), \tag{57}$$

*where $\Phi_\gamma$ is the optimal value of equation 56.*

*Proof.* Let $\mathbf{o} \in \arg\max_{\mathbf{y} \in P} F(\mathbf{y})$. By Corollary 4.3, Algorithm 2 has at least one *successful* heir call. For such a call with seed $\mathbf{z}^\star$, Theorem 4.2 (1) and Lemma 4.4 provide two certified lower bounds on the returned candidates $\mathbf{y}^\star$ and $\mathbf{z}'^\star$. Forming any convex combination of these two bounds yields equation 52, with $D_\gamma, E_\gamma$ given in equation 53.

Choose $(\alpha, r, t_s) \in \mathcal{F}_\gamma$ (cf. equation 54) so that the coefficients of $F(\mathbf{z}^\star \odot \mathbf{o})$ and $F(\mathbf{z}^\star \oplus \mathbf{o})$ in equation 52 are nonnegative. Dropping these nonnegative contributions gives equation 55:

$$F(\text{ALG}) \ \geq \ \left[(1 - \alpha)A_\gamma(t_s) - O(\varepsilon)\right] F(\mathbf{o}) \ - \ O(\delta L D^2). \tag{58}$$

Maximizing over feasible $(\alpha, r, t_s)$ yields $\Phi_\gamma$ from equation 56, so, for an optimal choice,

$$F(\text{ALG}) \ \geq \ \left[\Phi_\gamma - O(\varepsilon)\right] F(\mathbf{o}) \ - \ O(\delta L D^2). \tag{59}$$

Finally, set $\varepsilon = \Theta(\delta)$ to absorb the $-O(\varepsilon)F(\mathbf{o})$ term into the $-O(\delta L D^2)$ smoothing error. $\qquad\square$

## 5    Conclusion

This paper develops a unified, projection free framework for maximizing continuous, non-monotone $\gamma$-weakly DR-submodular functions over down-closed convex bodies. Our method couples a $\gamma$-aware Frank Wolfe guided measured continuous greedy with a $\gamma$-aware double-greedy, and optimizes a convex mixture of their certificates through three tunable parameters $(\alpha, r, t_s)$. Across the entire weakly-DR spectrum, the resulting guarantee $\Phi_\gamma$ *strictly* improves the canonical non-monotone baseline $\kappa(\gamma) = \gamma e^{-\gamma}$, and at the DR boundary ($\gamma = 1$) it *matches* the current best constant 0.401. We show improvements over prior work in Figure 1 and Table 2.

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

# A    Notations and Definitions

## Notation Summary

- $N = [n] := \{1, 2, \ldots, n\}$ denotes the ground set of coordinates, with $|N| = n$.

- Boldface letters such as $\mathbf{x}, \mathbf{y}, \mathbf{z}, \mathbf{o}, \mathbf{a} \in \mathbb{R}^n$ denote vectors. We write $\mathbf{0}$ and $\mathbf{1}$ for the all zeros and all ones vectors, and $\mathbf{e}_i$ for the $i$th standard basis vector.

- For vectors $\mathbf{x}, \mathbf{y} \in \mathbb{R}^n$, the componentwise order is

$$\mathbf{x} \le \mathbf{y} \iff x_i \le y_i \text{ for all } i \in [n].$$

- The coordinatewise join and meet are

$$\mathbf{x} \vee \mathbf{y} := (\max\{x_i, y_i\})_{i=1}^n, \qquad \mathbf{x} \wedge \mathbf{y} := (\min\{x_i, y_i\})_{i=1}^n.$$

- The Hadamard (elementwise) product is

$$(\mathbf{x} \odot \mathbf{y})_i := x_i y_i \qquad \text{for each } i \in [n].$$

- The coordinatewise probabilistic sum is

$$\mathbf{x} \oplus \mathbf{y} := \mathbf{1} - (\mathbf{1} - \mathbf{x}) \odot (\mathbf{1} - \mathbf{y}).$$

More generally, for $\mathbf{x}^{(1)}, \ldots, \mathbf{x}^{(m)} \in [0, 1]^n$,

$$\bigoplus_{j=1}^m \mathbf{x}^{(j)} := \mathbf{1} - \bigodot_{j=1}^m (\mathbf{1} - \mathbf{x}^{(j)}).$$

- The inner product and Euclidean norm are denoted by

$$\langle \mathbf{x}, \mathbf{y} \rangle := \sum_{i=1}^{n} x_i y_i, \qquad \|\mathbf{x}\|_2 := \left( \sum_{i=1}^{n} x_i^2 \right)^{1/2}.$$

- The weak DR parameter is $\gamma \in (0, 1]$. The case $\gamma = 1$ reduces to the classical DR-submodular setting.

- The diameter of $P$ is

$$D := \sup\{\|\mathbf{x} - \mathbf{y}\|_2 : \mathbf{x}, \mathbf{y} \in P\}.$$

- $\alpha$ is a mixing parameter. It is used to form a convex combination of the two guarantees obtained from the $\gamma$-FWG algorithm and the double-greedy algorithm.

- $r$ is the double-greedy balance parameter.

- $t_s$ is the switching/scheduling parameter used in the $\gamma$-FWG algorithm.

- $\varepsilon$ is an accuracy parameter used in the main algorithm and its proof.

- $\delta$ is the accuracy/discretization parameter.

- The baseline approximation curve is $\kappa(\gamma) := \gamma e^{-\gamma}$.

- The optimized approximation guarantee obtained by our method is denoted by $\Phi_\gamma$.

## Definitions Summary

**Definition A.1** (Submodular set function)**.** A nonnegative set function $f : \{0, 1\}^n \to \mathbb{R}_{\geq 0}$ is called *submodular* if for all $\mathbf{x}, \mathbf{y} \in \{0, 1\}^n$ with $\mathbf{x} \leq \mathbf{y}$ and all $\mathbf{a} \in \{0, 1\}^n$,

$$f(\mathbf{x} \vee \mathbf{a}) - f(\mathbf{x}) \ \geq \ f(\mathbf{y} \vee \mathbf{a}) - f(\mathbf{y}).$$

**Definition A.2** (DR-submodular function)**.** A nonnegative function $F : [0, 1]^n \to \mathbb{R}_{\geq 0}$ is *diminishing returns (DR) submodular* if for all $\mathbf{x}, \mathbf{y} \in [0, 1]^n$ with $\mathbf{x} \leq \mathbf{y}$, every coordinate $i \in [n]$, and every $c > 0$ such that $\mathbf{x} + c\mathbf{e}_i, \mathbf{y} + c\mathbf{e}_i \in [0, 1]^n$,

$$F(\mathbf{x} + c\mathbf{e}_i) - F(\mathbf{x}) \ \geq \ F(\mathbf{y} + c\mathbf{e}_i) - F(\mathbf{y}).$$

**Definition A.3** ($\gamma$-weakly DR-submodular function)**.** A nonnegative function $F : [0, 1]^n \to \mathbb{R}_{\geq 0}$ is *$\gamma$-weakly DR-submodular* if for all $\mathbf{x}, \mathbf{y} \in [0, 1]^n$ with $\mathbf{x} \leq \mathbf{y}$, every coordinate $i \in [n]$, and every $c > 0$ such that $\mathbf{x} + c\mathbf{e}_i, \mathbf{y} + c\mathbf{e}_i \in [0, 1]^n$,

$$F(\mathbf{x} + c\mathbf{e}_i) - F(\mathbf{x}) \ \geq \ \gamma \Big( F(\mathbf{y} + c\mathbf{e}_i) - F(\mathbf{y}) \Big),$$

where $\gamma \in (0, 1]$. This relaxes classical DR-submodularity by allowing marginal gains to decay up to a multiplicative factor $\gamma$.

**Definition A.4** (Down-closed convex body)**.** A set $P \subseteq [0, 1]^n$ is called *down-closed* if for every $\mathbf{y} \in P$ and every $\mathbf{x} \in \mathbb{R}^n$ satisfying $\mathbf{0} \leq \mathbf{x} \leq \mathbf{y}$, we also have $\mathbf{x} \in P$. If in addition $P$ is compact, convex, and has nonempty interior, then $P$ is a down-closed convex body.

**Definition A.5** (*L*-smoothness)**.** A differentiable function $F : P \to \mathbb{R}$ is called *L-smooth* if for all $\mathbf{x}, \mathbf{y} \in P$,

$$\|\nabla F(\mathbf{x}) - \nabla F(\mathbf{y})\|_2 \leq L \|\mathbf{x} - \mathbf{y}\|_2.$$

Equivalently, the gradient of $F$ is Lipschitz continuous with constant $L$.

# B   Proofs of Section 2 Lemmas

In this appendix, we provide detailed proofs of the fundamental properties of $\gamma$-weakly DR-submodular functions and extend classical results for DR-submodular functions (Buchbinder & Feldman, 2024) to the generalized $\gamma$-weakly setting. For clarity, we restate lemmas before presenting their proofs.

**Lemma 2.1.** Let $F : [0,1]^n \to \mathbb{R}_{\geq 0}$ be differentiable and $\gamma$-weakly DR-submodular. Then for all $\mathbf{x}, \mathbf{y} \in [0,1]^n$ and $\lambda \in [0,1]$ the following hold:

(1) If $\mathbf{x} \leq \mathbf{y}$, then

$$F\big(\lambda \mathbf{x} + (1-\lambda)\mathbf{y}\big) \;\geq\; \frac{\lambda\, F(\mathbf{x}) + \gamma^2(1-\lambda)\, F(\mathbf{y})}{\lambda + \gamma^2(1-\lambda)}. \tag{60}$$

Equivalently,

$$F\big((1-\lambda)\mathbf{x} + \lambda \mathbf{y}\big) \;\geq\; \frac{(1-\lambda)\, F(\mathbf{x}) + \gamma^2\lambda\, F(\mathbf{y})}{(1-\lambda) + \gamma^2\lambda}. \tag{61}$$

(2) If $\mathbf{y} \geq \mathbf{0}$ and $\mathbf{x} + \mathbf{y} \in [0,1]^n$, then

$$F(\mathbf{x} + \lambda \mathbf{y}) - F(\mathbf{x}) \;\geq\; \frac{\gamma^2 \lambda}{1 - \lambda + \gamma^2 \lambda}\,\big(F(\mathbf{x} + \mathbf{y}) - F(\mathbf{x})\big). \tag{62}$$

*Proof.* Fix $\mathbf{x} \in [0,1]^n$ and a direction $\mathbf{v} \geq 0$ such that $\mathbf{x} + t\mathbf{v} \in [0,1]^n$ for all $t \in [0,1]$. Define the univariate function

$$\phi(t) \;:=\; F(\mathbf{x} + t\mathbf{v}), \qquad t \in [0,1]. \tag{63}$$

By the chain rule, $\phi$ is differentiable and

$$\phi'(t) \;=\; \big\langle \nabla F(\mathbf{x} + t\mathbf{v}), \mathbf{v} \big\rangle. \tag{64}$$

Now fix $0 \leq s \leq t \leq 1$. Since $\mathbf{v} \geq 0$, we have $\mathbf{x} + s\mathbf{v} \leq \mathbf{x} + t\mathbf{v}$, and by $\gamma$-weak DR-submodularity this implies

$$\nabla F(\mathbf{x} + s\mathbf{v}) \;\geq\; \gamma\, \nabla F(\mathbf{x} + t\mathbf{v}). \tag{65}$$

Taking inner products of both sides of equation 65 with $\mathbf{v} \geq 0$ and using equation 64 gives

$$\phi'(s) \;=\; \big\langle \nabla F(\mathbf{x} + s\mathbf{v}), \mathbf{v} \big\rangle \;\geq\; \gamma \big\langle \nabla F(\mathbf{x} + t\mathbf{v}), \mathbf{v} \big\rangle \;=\; \gamma\, \phi'(t), \qquad 0 \leq s \leq t \leq 1. \tag{66}$$

Next fix $\lambda \in (0,1)$. For $t \in [\lambda, 1]$, applying equation 66 with $s = \lambda$ gives $\phi'(\lambda) \geq \gamma\, \phi'(t)$, so

$$\phi'(t) \;\leq\; \frac{1}{\gamma}\, \phi'(\lambda).$$

Integrating this upper bound over $t \in [\lambda, 1]$ and using the fundamental theorem of calculus yields

$$\phi(1) - \phi(\lambda) \;=\; \int_\lambda^1 \phi'(t)\, dt \;\leq\; \int_\lambda^1 \frac{1}{\gamma}\, \phi'(\lambda)\, dt \;=\; \frac{1-\lambda}{\gamma}\, \phi'(\lambda). \tag{67}$$

Similarly, for $s \in [0, \lambda]$, applying equation 66 with $t = \lambda$ gives $\phi'(s) \geq \gamma\, \phi'(\lambda)$. Integrating this lower bound over $s \in [0, \lambda]$ we obtain

$$\phi(\lambda) - \phi(0) \;=\; \int_0^\lambda \phi'(s)\, ds \;\geq\; \int_0^\lambda \gamma\, \phi'(\lambda)\, ds \;=\; \gamma\, \lambda\, \phi'(\lambda). \tag{68}$$

From equation 67 we get

$$\phi'(\lambda) \;\geq\; \frac{\gamma}{1-\lambda}\,\big(\phi(1) - \phi(\lambda)\big),$$

and substituting this lower bound into equation 68 gives

$$\phi(\lambda) - \phi(0) \;\geq\; \gamma\,\lambda \cdot \frac{\gamma}{1-\lambda}\left(\phi(1) - \phi(\lambda)\right) \;=\; \frac{\gamma^2\lambda}{1-\lambda}\left(\phi(1) - \phi(\lambda)\right). \tag{69}$$

Multiplying both sides of equation 69 by $1-\lambda$ and expanding, we obtain

$$(1-\lambda)\,\phi(\lambda) - (1-\lambda)\,\phi(0) \;\geq\; \gamma^2\lambda\,\phi(1) - \gamma^2\lambda\,\phi(\lambda).$$

Rearranging the terms involving $\phi(\lambda)$ to the left and the remaining terms to the right yields

$$\left(1 - \lambda + \gamma^2\lambda\right)\phi(\lambda) \;\geq\; (1-\lambda)\,\phi(0) + \gamma^2\lambda\,\phi(1).$$

Dividing by $1 - \lambda + \gamma^2\lambda > 0$ we get

$$\phi(\lambda) \;\geq\; \frac{(1-\lambda)\,\phi(0) + \gamma^2\lambda\,\phi(1)}{(1-\lambda) + \gamma^2\lambda}. \tag{70}$$

To prove part (2), take $\mathbf{v} = \mathbf{y} \geq 0$ and $\phi(t) = F(\mathbf{x} + t\mathbf{y})$ as in equation 63. Since $\mathbf{x} + \mathbf{y} \in [0,1]^n$, the entire segment $\{\mathbf{x} + t\mathbf{y} : t \in [0,1]\}$ lies in $[0,1]^n$, so the above argument applies. In this case,

$$\phi(0) = F(\mathbf{x}), \qquad \phi(1) = F(\mathbf{x} + \mathbf{y}), \qquad \phi(\lambda) = F(\mathbf{x} + \lambda\mathbf{y}).$$

Substituting these expressions into equation 70 gives

$$F(\mathbf{x} + \lambda\mathbf{y}) \;\geq\; \frac{(1-\lambda)F(\mathbf{x}) + \gamma^2\lambda F(\mathbf{x} + \mathbf{y})}{(1-\lambda) + \gamma^2\lambda}. \tag{71}$$

Subtracting $F(\mathbf{x})$ from both sides of equation 71, we obtain

$$F(\mathbf{x} + \lambda\mathbf{y}) - F(\mathbf{x}) \;\geq\; \frac{(1-\lambda)F(\mathbf{x}) + \gamma^2\lambda F(\mathbf{x} + \mathbf{y})}{(1-\lambda) + \gamma^2\lambda} \;-\; F(\mathbf{x}).$$

Writing $F(\mathbf{x})$ as $\frac{(1-\lambda)+\gamma^2\lambda}{(1-\lambda)+\gamma^2\lambda}F(\mathbf{x})$ and simplifying the numerator, we get

$$F(\mathbf{x} + \lambda\mathbf{y}) - F(\mathbf{x}) \;\geq\; \frac{\gamma^2\lambda}{1 - \lambda + \gamma^2\lambda}\left(F(\mathbf{x} + \mathbf{y}) - F(\mathbf{x})\right),$$

which is exactly equation 62. This proves part (2).

For part (1), assume $\mathbf{x} \leq \mathbf{y}$ and define $\mathbf{v} := \mathbf{y} - \mathbf{x} \geq 0$. For $t \in [0,1]$ set

$$\phi(t) \;:=\; F\big(\mathbf{x} + t(\mathbf{y} - \mathbf{x})\big) \;=\; F\big((1-t)\mathbf{x} + t\mathbf{y}\big). \tag{72}$$

Again, the segment between $\mathbf{x}$ and $\mathbf{y}$ lies in $[0,1]^n$, so equation 70 applies. Here

$$\phi(0) = F(\mathbf{x}), \qquad \phi(1) = F(\mathbf{y}), \qquad \phi(\lambda) = F\big((1-\lambda)\mathbf{x} + \lambda\mathbf{y}\big).$$

Substituting into equation 70, we obtain

$$F\big((1-\lambda)\mathbf{x} + \lambda\mathbf{y}\big) \;\geq\; \frac{(1-\lambda)\,F(\mathbf{x}) + \gamma^2\lambda\,F(\mathbf{y})}{(1-\lambda) + \gamma^2\lambda}, \tag{73}$$

which is exactly equation 61. Finally, replacing $\lambda$ by $1 - \lambda$ in equation 73 gives

$$F\big(\lambda\mathbf{x} + (1-\lambda)\mathbf{y}\big) \;\geq\; \frac{\lambda\,F(\mathbf{x}) + \gamma^2(1-\lambda)\,F(\mathbf{y})}{\lambda + \gamma^2(1-\lambda)},$$

which is equation : 60. This proves part (1). $\qquad\square$

**Lemma 2.2.** Let $F : [0,1]^n \to \mathbb{R}_{\geq 0}$ be a differentiable $\gamma$-weakly DR-submodular function. Then, for every $\mathbf{x}, \mathbf{y} \in [0,1]^n$ with $\mathbf{y} \geq \mathbf{0}$, the following inequalities hold:

1. If $\mathbf{x} + \mathbf{y} \leq \mathbf{1}$, then

$$\langle \nabla F(\mathbf{x}), \mathbf{y} \rangle \geq \gamma \big( F(\mathbf{x} + \mathbf{y}) - F(\mathbf{x}) \big). \tag{74}$$

2. If $\mathbf{x} - \mathbf{y} \geq \mathbf{0}$, then

$$\langle \nabla F(\mathbf{x}), \mathbf{y} \rangle \leq \frac{1}{\gamma} \big( F(\mathbf{x}) - F(\mathbf{x} - \mathbf{y}) \big). \tag{75}$$

*Proof.* We first prove part (1). Assume $\mathbf{x} + \mathbf{y} \leq \mathbf{1}$ and define

$$g(t) := F(\mathbf{x} + t\mathbf{y}), \qquad t \in [0,1]. \tag{76}$$

The condition $\mathbf{x} + \mathbf{y} \leq \mathbf{1}$ and $\mathbf{y} \geq \mathbf{0}$ implies $\mathbf{x} + t\mathbf{y} \in [0,1]^n$ for all $t \in [0,1]$. By the chain rule,

$$g'(t) = \langle \nabla F(\mathbf{x} + t\mathbf{y}), \mathbf{y} \rangle. \tag{77}$$

For each $t \in [0,1]$ we have $\mathbf{x} \leq \mathbf{x} + t\mathbf{y}$, so by $\gamma$-weak DR-submodularity,

$$\nabla F(\mathbf{x}) \geq \gamma \nabla F(\mathbf{x} + t\mathbf{y}), \qquad 0 \leq t \leq 1. \tag{78}$$

Taking inner products of both sides of equation 78 with $\mathbf{y} \geq \mathbf{0}$ and using equation 77 gives

$$\langle \nabla F(\mathbf{x}), \mathbf{y} \rangle \geq \gamma \langle \nabla F(\mathbf{x} + t\mathbf{y}), \mathbf{y} \rangle = \gamma g'(t), \qquad 0 \leq t \leq 1. \tag{79}$$

Equivalently,

$$g'(t) \leq \frac{1}{\gamma} \langle \nabla F(\mathbf{x}), \mathbf{y} \rangle, \qquad 0 \leq t \leq 1. \tag{80}$$

Integrating the bound equation 80 over $t \in [0,1]$ and using the fundamental theorem of calculus yields

$$F(\mathbf{x} + \mathbf{y}) - F(\mathbf{x}) = \int_0^1 g'(t)\, dt \leq \int_0^1 \frac{1}{\gamma} \langle \nabla F(\mathbf{x}), \mathbf{y} \rangle\, dt = \frac{1}{\gamma} \langle \nabla F(\mathbf{x}), \mathbf{y} \rangle. \tag{81}$$

Rearranging equation 81 gives

$$\langle \nabla F(\mathbf{x}), \mathbf{y} \rangle \geq \gamma \big( F(\mathbf{x} + \mathbf{y}) - F(\mathbf{x}) \big), \tag{82}$$

which is exactly equation 74.

We now prove part (2). Assume $\mathbf{x} - \mathbf{y} \geq \mathbf{0}$ and define

$$h(t) := F(\mathbf{x} - t\mathbf{y}), \qquad t \in [0,1]. \tag{83}$$

The condition $\mathbf{x} - \mathbf{y} \geq \mathbf{0}$ and $\mathbf{y} \geq \mathbf{0}$ implies $\mathbf{x} - t\mathbf{y} \in [0,1]^n$ for all $t \in [0,1]$. Again by the chain rule,

$$h'(t) = \langle \nabla F(\mathbf{x} - t\mathbf{y}), -\mathbf{y} \rangle = -\langle \nabla F(\mathbf{x} - t\mathbf{y}), \mathbf{y} \rangle. \tag{84}$$

For each $t \in [0,1]$ we have $\mathbf{x} - t\mathbf{y} \leq \mathbf{x}$, so $\gamma$-weak DR-submodularity gives

$$\nabla F(\mathbf{x} - t\mathbf{y}) \geq \gamma \nabla F(\mathbf{x}), \qquad 0 \leq t \leq 1. \tag{85}$$

Taking inner products with $\mathbf{y} \geq \mathbf{0}$ and using equation 84 we obtain

$$\langle \nabla F(\mathbf{x} - t\mathbf{y}), \mathbf{y} \rangle \geq \gamma \langle \nabla F(\mathbf{x}), \mathbf{y} \rangle, \qquad 0 \leq t \leq 1, \tag{86}$$

and hence

$$h'(t) = -\langle \nabla F(\mathbf{x} - t\mathbf{y}), \mathbf{y} \rangle \leq -\gamma \langle \nabla F(\mathbf{x}), \mathbf{y} \rangle, \qquad 0 \leq t \leq 1. \tag{87}$$

Integrating equation 87 over $t \in [0, 1]$ and applying the fundamental theorem of calculus gives

$$F(\mathbf{x} - \mathbf{y}) - F(\mathbf{x}) \;=\; \int_0^1 h'(t)\, dt \;\leq\; \int_0^1 -\gamma \left\langle \nabla F(\mathbf{x}), \mathbf{y} \right\rangle dt \;=\; -\gamma \left\langle \nabla F(\mathbf{x}), \mathbf{y} \right\rangle. \tag{88}$$

Multiplying equation 88 by $-1$ yields

$$F(\mathbf{x}) - F(\mathbf{x} - \mathbf{y}) \;\geq\; \gamma \left\langle \nabla F(\mathbf{x}), \mathbf{y} \right\rangle. \tag{89}$$

Rearranging equation 89, we obtain

$$\left\langle \nabla F(\mathbf{x}), \mathbf{y} \right\rangle \;\leq\; \frac{1}{\gamma} \big( F(\mathbf{x}) - F(\mathbf{x} - \mathbf{y}) \big), \tag{90}$$

which is exactly equation 75. This completes the proof. $\qquad\square$

**Lemma 2.3.** Let $F : [0, 1]^n \to \mathbb{R}_{\geq 0}$ be nonnegative and $\gamma$–weakly DR-submodular for some $\gamma \in (0, 1]$. For any fixed $\mathbf{y} \in [0, 1]^n$, define

$$G_\oplus(\mathbf{x}) \;:=\; F(\mathbf{x} \oplus \mathbf{y}) \tag{91}$$

and

$$G_\odot(\mathbf{x}) \;:=\; F(\mathbf{x} \odot \mathbf{y}). \tag{92}$$

Then both $G_\oplus$ and $G_\odot$ are nonnegative and $\gamma$–weakly DR-submodular, that is, for all $\mathbf{x}^{(1)}, \mathbf{x}^{(2)} \in [0, 1]^n$ with $\mathbf{x}^{(1)} \leq \mathbf{x}^{(2)}$, any coordinate $u \in [n]$, and any step $p \in [0, 1 - \mathbf{x}_u^{(2)}]$ such that the updates stay in $[0, 1]^n$, we have

$$G_\oplus\big(\mathbf{x}^{(1)} + p\,\mathbf{e}_u\big) - G_\oplus(\mathbf{x}^{(1)}) \;\geq\; \gamma \Big( G_\oplus\big(\mathbf{x}^{(2)} + p\,\mathbf{e}_u\big) - G_\oplus(\mathbf{x}^{(2)}) \Big) \tag{93}$$

and

$$G_\odot\big(\mathbf{x}^{(1)} + p\,\mathbf{e}_u\big) - G_\odot(\mathbf{x}^{(1)}) \;\geq\; \gamma \Big( G_\odot\big(\mathbf{x}^{(2)} + p\,\mathbf{e}_u\big) - G_\odot(\mathbf{x}^{(2)}) \Big). \tag{94}$$

*Proof.* Nonnegativity of $G_\oplus$ and $G_\odot$ follows directly from equation 91, equation 92 and the nonnegativity of $F$.

Fix $\mathbf{x}^{(1)}, \mathbf{x}^{(2)} \in [0, 1]^n$ with $\mathbf{x}^{(1)} \leq \mathbf{x}^{(2)}$, a coordinate $u \in [n]$, and a step $p \in [0, 1 - \mathbf{x}_u^{(2)}]$, so that $\mathbf{x}^{(j)} + p\,\mathbf{e}_u \in [0, 1]^n$ for $j = 1, 2$.

We first treat the $\oplus$ case. Set

$$\mathbf{z}^{(j)} \;:=\; \mathbf{x}^{(j)} \oplus \mathbf{y}, \qquad j \in \{1, 2\}. \tag{95}$$

Since $\mathbf{x}^{(1)} \leq \mathbf{x}^{(2)}$ and the map $x \mapsto x \oplus y$ is nondecreasing in $x$, we have

$$\mathbf{z}^{(1)} \;\leq\; \mathbf{z}^{(2)}. \tag{96}$$

Using the coordinate wise identity

$$(\mathbf{x} \oplus \mathbf{y})_i \;=\; x_i + y_i - x_i y_i, \tag{97}$$

we can express the update $(\mathbf{x}^{(j)} + p\,\mathbf{e}_u) \oplus \mathbf{y}$ in terms of $\mathbf{z}^{(j)}$. Indeed, only the $u$-th coordinate of $\mathbf{x}^{(j)}$ changes, so for each $j \in \{1, 2\}$ we have

$$(\mathbf{x}^{(j)} + p\,\mathbf{e}_u) \oplus \mathbf{y} \;=\; \mathbf{z}^{(j)} + \alpha\,\mathbf{e}_u, \tag{98}$$

where

$$\alpha \;:=\; p\,(1 - \mathbf{y}_u). \tag{99}$$

This follows by plugging $x_u^{(j)} + p$ into the expression equation 97 and simplifying.

Next we check that the updated point stays in $[0, 1]^n$ at coordinate $u$. Using equation 97 with $x_u^{(2)}$ we obtain

$$1 - \mathbf{z}_u^{(2)} \;=\; 1 - \big( x_u^{(2)} + y_u - x_u^{(2)} y_u \big) \;=\; (1 - \mathbf{x}_u^{(2)})(1 - \mathbf{y}_u). \tag{100}$$

Since $p \leq 1 - \mathbf{x}_u^{(2)}$ by assumption and $1 - \mathbf{y}_u \geq 0$, we get

$$\alpha = p(1 - \mathbf{y}_u) \leq (1 - \mathbf{x}_u^{(2)})(1 - \mathbf{y}_u) = 1 - \mathbf{z}_u^{(2)}. \tag{101}$$

Thus $\mathbf{z}^{(2)} + \alpha \, \mathbf{e}_u$ remains in $[0,1]^n$.

Now we use the $\gamma$–weak DR-submodularity of $F$. From equation 96 and equation 101 we can apply the definition of $\gamma$–weak DR-submodularity to the pair $(\mathbf{z}^{(1)}, \mathbf{z}^{(2)})$ with step $\alpha$ in coordinate $u$ and obtain

$$F\big(\mathbf{z}^{(1)} + \alpha \, \mathbf{e}_u\big) - F(\mathbf{z}^{(1)}) \geq \gamma\Big(F\big(\mathbf{z}^{(2)} + \alpha \, \mathbf{e}_u\big) - F(\mathbf{z}^{(2)})\Big). \tag{102}$$

Using equation 91 and equation 98, we can rewrite the left hand side and the right hand side of equation 102 in terms of $G_\oplus$ as

$$F\big(\mathbf{z}^{(j)} + \alpha \, \mathbf{e}_u\big) = G_\oplus\big(\mathbf{x}^{(j)} + p \, \mathbf{e}_u\big), \qquad F(\mathbf{z}^{(j)}) = G_\oplus(\mathbf{x}^{(j)}), \quad j = 1, 2. \tag{103}$$

Substituting equation 103 into equation 102 gives

$$G_\oplus\big(\mathbf{x}^{(1)} + p \, \mathbf{e}_u\big) - G_\oplus(\mathbf{x}^{(1)}) \geq \gamma\Big(G_\oplus\big(\mathbf{x}^{(2)} + p \, \mathbf{e}_u\big) - G_\oplus(\mathbf{x}^{(2)})\Big), \tag{104}$$

which is exactly equation 93. This shows that $G_\oplus$ is $\gamma$–weakly DR-submodular.

We now treat the $\odot$ case. Set

$$\mathbf{w}^{(j)} := \mathbf{x}^{(j)} \odot \mathbf{y}, \qquad j \in \{1, 2\}. \tag{105}$$

Since the map $x \mapsto x \odot y$ is also nondecreasing in $x$, we again have

$$\mathbf{w}^{(1)} \leq \mathbf{w}^{(2)}. \tag{106}$$

By definition of $\odot$,

$$(\mathbf{x} \odot \mathbf{y})_i = x_i y_i, \tag{107}$$

so updating $\mathbf{x}^{(j)}$ in coordinate $u$ by $p$ gives

$$(\mathbf{x}^{(j)} + p \, \mathbf{e}_u) \odot \mathbf{y} = \mathbf{w}^{(j)} + \beta \, \mathbf{e}_u, \tag{108}$$

where

$$\beta := p \, \mathbf{y}_u. \tag{109}$$

Again we check that the updated point stays in $[0,1]^n$ at coordinate $u$. From equation 107 we have

$$\mathbf{w}_u^{(2)} = \mathbf{x}_u^{(2)} \mathbf{y}_u, \tag{110}$$

so

$$1 - \mathbf{w}_u^{(2)} = 1 - \mathbf{x}_u^{(2)} \mathbf{y}_u. \tag{111}$$

Using $p \leq 1 - \mathbf{x}_u^{(2)}$ and $\mathbf{y}_u \leq 1$, we obtain

$$\beta = p \, \mathbf{y}_u \leq (1 - \mathbf{x}_u^{(2)}) \mathbf{y}_u \leq 1 - \mathbf{x}_u^{(2)} \mathbf{y}_u = 1 - \mathbf{w}_u^{(2)}. \tag{112}$$

Hence $\mathbf{w}^{(2)} + \beta \, \mathbf{e}_u \in [0,1]^n$.

Now we again use the $\gamma$–weak DR-submodularity of $F$. By equation 106 and equation 112, the definition applied to $(\mathbf{w}^{(1)}, \mathbf{w}^{(2)})$ with step $\beta$ in coordinate $u$ yields

$$F\big(\mathbf{w}^{(1)} + \beta \, \mathbf{e}_u\big) - F(\mathbf{w}^{(1)}) \geq \gamma\Big(F\big(\mathbf{w}^{(2)} + \beta \, \mathbf{e}_u\big) - F(\mathbf{w}^{(2)})\Big). \tag{113}$$

Using equation 92 and equation 108, we can rewrite equation 113 as

$$G_\odot\big(\mathbf{x}^{(1)} + p \, \mathbf{e}_u\big) - G_\odot(\mathbf{x}^{(1)}) \geq \gamma\Big(G_\odot\big(\mathbf{x}^{(2)} + p \, \mathbf{e}_u\big) - G_\odot(\mathbf{x}^{(2)})\Big), \tag{114}$$

which is exactly equation 94. Thus $G_\odot$ is also $\gamma$–weakly DR-submodular, completing the proof. $\square$

# C Frank-Wolfe Algorithm and Proof of Theorem 3.2

This section develops a first–order certificate tailored to the $\gamma$–weakly DR setting and uses it to prove our main result, Theorem 3.2. The argument follows a *local to global* template: (i) a local optimality condition at $\mathbf{x}$ yields a $\gamma$–weighted comparison between $F(\mathbf{x})$ and the join/meet values with any comparator $\mathbf{y}$ (Lemma 3.1); (ii) a Frank-Wolfe variant produces a point $\mathbf{x} \in P$ that satisfies a *uniform* first–order certificate against *every* $\mathbf{y} \in P$ (Lemma C.1); and (iii) combining the two delivers a global value bound that degrades smoothly with $\gamma$ and matches the classical DR guarantee at $\gamma = 1$.

**Algorithmic setup.** We will invoke the following Frank Wolfe type routine from (Buchbinder & Feldman, 2024). For clarity of presentation, we assume $\delta^{-1} \in \mathbb{N}$; if not, we replace $\delta$ by $1/\lceil \delta^{-1} \rceil$ without affecting the asymptotics.

---

**Algorithm 3** Frank-Wolfe Variant$(F, P, \delta)$

---

1: Let $\mathbf{x}(0)$ be an arbitrary vector in $P$.
2: **for** $i = 1$ to $\delta^{-2}$ **do**
3:     Let $\mathbf{z}(i) \in \arg\max_{\mathbf{y} \in P} \langle \mathbf{y}, \nabla F(\mathbf{x}(i-1)) \rangle$.
4:     Let $\mathbf{x}(i) \leftarrow (1 - \delta)\,\mathbf{x}(i-1) + \delta\,\mathbf{z}(i)$.
5: **end for**
6: Let $i^* \in \arg\min_{1 \le i \le \delta^{-2}} \{\langle \mathbf{z}(i) - \mathbf{x}(i-1), \nabla F(\mathbf{x}(i-1)) \rangle\}$.
7: **return** $\mathbf{x}(i^* - 1)$.

---

As observed in (Buchbinder & Feldman, 2024), the update rule

$$\mathbf{x}(i) \ = \ (1 - \delta)\,\mathbf{x}(i-1) + \delta\,\mathbf{z}(i), \qquad \mathbf{z}(i) \in P,$$

keeps all iterates in $P$; in particular, $\mathbf{x}(i) \in P$ for every $0 \le i \le \delta^{-2}$.

The next lemma converts the local optimality condition into a lattice based comparison that interpolates in $\gamma$; it coincides with the classical $\frac{1}{2}\big(F(\mathbf{x} \vee \mathbf{y}) + F(\mathbf{x} \wedge \mathbf{y})\big)$ bound at $\gamma = 1$.

**Lemma 3.1.** *Let $F : [0,1]^n \to \mathbb{R}_{\ge 0}$ be differentiable and $\gamma$–weakly DR-submodular. If $\mathbf{x}$ is a local optimum with respect to $\mathbf{y}$, it means;*

$$\langle \mathbf{y} - \mathbf{x}, \nabla F(\mathbf{x}) \rangle \ \le \ 0, \tag{115}$$

*then*

$$F(\mathbf{x}) \ \ge \ \frac{\gamma^2\, F(\mathbf{x} \vee \mathbf{y}) + F(\mathbf{x} \wedge \mathbf{y})}{1 + \gamma^2}. \tag{116}$$

*Proof.* Starting from the local optimality condition equation 115, we decompose $\mathbf{y} - \mathbf{x}$ as

$$\mathbf{y} - \mathbf{x} \ = \ (\mathbf{y} \vee \mathbf{x} - \mathbf{x}) \ - \ (\mathbf{x} - \mathbf{y} \wedge \mathbf{x}). \tag{117}$$

Substituting equation 117 into equation 115 gives

$$0 \ \ge \ \langle \mathbf{y} - \mathbf{x}, \nabla F(\mathbf{x}) \rangle \ = \ \langle \mathbf{y} \vee \mathbf{x} - \mathbf{x}, \nabla F(\mathbf{x}) \rangle \ - \ \langle \mathbf{x} - \mathbf{y} \wedge \mathbf{x}, \nabla F(\mathbf{x}) \rangle. \tag{118}$$

We now bound each inner product using Lemma 2.2. First, note that $\mathbf{y} \vee \mathbf{x} - \mathbf{x} \ge \mathbf{0}$ and

$$\mathbf{x} + (\mathbf{y} \vee \mathbf{x} - \mathbf{x}) \ = \ \mathbf{y} \vee \mathbf{x} \ \in \ [0,1]^n,$$

so Lemma 2.2(1) applies with the direction $\mathbf{y} \vee \mathbf{x} - \mathbf{x}$. We obtain

$$\langle \nabla F(\mathbf{x}), \mathbf{y} \vee \mathbf{x} - \mathbf{x} \rangle \ \ge \ \gamma\big(F(\mathbf{y} \vee \mathbf{x}) - F(\mathbf{x})\big). \tag{119}$$

Similarly, $\mathbf{x} - \mathbf{y} \wedge \mathbf{x} \ge \mathbf{0}$ and

$$\mathbf{x} - (\mathbf{x} - \mathbf{y} \wedge \mathbf{x}) \ = \ \mathbf{y} \wedge \mathbf{x} \ \in \ [0,1]^n,$$

so Lemma 2.2(2) applies with the direction $\mathbf{x} - \mathbf{y} \wedge \mathbf{x}$. This gives

$$\langle \nabla F(\mathbf{x}), \mathbf{x} - \mathbf{y} \wedge \mathbf{x} \rangle \;\leq\; \frac{1}{\gamma}\big(F(\mathbf{x}) - F(\mathbf{y} \wedge \mathbf{x})\big). \tag{120}$$

Substituting the bounds equation 119 and equation 120 into equation 118 yields

$$0 \;\geq\; \gamma\big(F(\mathbf{y} \vee \mathbf{x}) - F(\mathbf{x})\big) \;-\; \frac{1}{\gamma}\big(F(\mathbf{x}) - F(\mathbf{y} \wedge \mathbf{x})\big). \tag{121}$$

Expanding equation 121, we obtain

$$0 \;\geq\; \gamma F(\mathbf{y} \vee \mathbf{x}) \;-\; \gamma F(\mathbf{x}) \;-\; \frac{1}{\gamma}F(\mathbf{x}) \;+\; \frac{1}{\gamma}F(\mathbf{y} \wedge \mathbf{x}). \tag{122}$$

Rearranging equation 122 by bringing the terms involving $F(\mathbf{x})$ to the right hand side gives

$$\gamma F(\mathbf{y} \vee \mathbf{x}) \;+\; \frac{1}{\gamma}F(\mathbf{y} \wedge \mathbf{x}) \;\leq\; \left(\gamma + \frac{1}{\gamma}\right) F(\mathbf{x}). \tag{123}$$

Dividing both sides of equation 123 by $\gamma + 1/\gamma = (1 + \gamma^2)/\gamma$ yields

$$F(\mathbf{x}) \;\geq\; \frac{\gamma F(\mathbf{y} \vee \mathbf{x}) + \frac{1}{\gamma}F(\mathbf{y} \wedge \mathbf{x})}{\gamma + \frac{1}{\gamma}} \;=\; \frac{\gamma^2 F(\mathbf{y} \vee \mathbf{x}) + F(\mathbf{y} \wedge \mathbf{x})}{1 + \gamma^2}. \tag{124}$$

This is exactly equation 116, after noting that $\mathbf{y} \vee \mathbf{x} = \mathbf{x} \vee \mathbf{y}$ and $\mathbf{y} \wedge \mathbf{x} = \mathbf{x} \wedge \mathbf{y}$. $\qquad\square$

The next lemma is a standard smoothness based guarantee produced by Algorithm 3 (see Theorem 2.4 in (Buchbinder & Feldman, 2024)).

**Lemma C.1** (Theorem 2.4 of (Buchbinder & Feldman, 2024)). *Let $F : [0,1]^n \to \mathbb{R}_{\geq 0}$ be nonnegative and $L$-smooth, let $P \subseteq [0,1]^n$ be a solvable convex body of diameter $D$, and let $\delta \in (0,1)$. There is a polynomial time algorithm that returns $\mathbf{x} \in P$ such that*

$$\langle \mathbf{y} - \mathbf{x}, \nabla F(\mathbf{x}) \rangle \;\leq\; \delta \left[ \max_{\mathbf{z} \in P} F(\mathbf{z}) + \frac{LD^2}{2} \right] \qquad \textit{for all } \mathbf{y} \in P. \tag{125}$$

Combining the uniform certificate equation 125 with the weakly–DR inequalities (Lemma 2.2) and the local to lattice comparison (Lemma 3.1) yields our main bound.

**Theorem 3.2.** *Let $F : [0,1]^n \to \mathbb{R}_{\geq 0}$ be nonnegative and $L$-smooth, and suppose $F$ is $\gamma$–weakly DR-submodular for some $\gamma \in (0,1]$. Let $P \subseteq [0,1]^n$ be a solvable convex body of diameter $D$, and let $\delta \in (0,1)$. Then there is a polynomial time algorithm that outputs $\mathbf{x} \in P$ such that, for every $\mathbf{y} \in P$,*

$$F(\mathbf{x}) \;\geq\; \frac{\gamma^2 F(\mathbf{x} \vee \mathbf{y}) + F(\mathbf{x} \wedge \mathbf{y})}{1 + \gamma^2} \;-\; \frac{\delta\,\gamma}{1 + \gamma^2} \left[ \max_{\mathbf{z} \in P} F(\mathbf{z}) + \frac{LD^2}{2} \right]. \tag{126}$$

*Proof.* Let $\mathbf{x} \in P$ be returned by Lemma C.1; then equation 125 holds for all $\mathbf{y} \in P$. As in the proof of Lemma 3.1, Lemma 2.2 implies that for every $\mathbf{y} \in [0,1]^n$,

$$\langle \mathbf{y} - \mathbf{x}, \nabla F(\mathbf{x}) \rangle \;\geq\; \gamma\,F(\mathbf{x} \vee \mathbf{y}) \;+\; \frac{1}{\gamma}\,F(\mathbf{x} \wedge \mathbf{y}) \;-\; \frac{1 + \gamma^2}{\gamma}\,F(\mathbf{x}). \tag{127}$$

Combining equation 125 and equation 127 for any $\mathbf{y} \in P$ gives

$$\delta \left[ \max_{\mathbf{z} \in P} F(\mathbf{z}) + \frac{LD^2}{2} \right] \;\geq\; \gamma\,F(\mathbf{x} \vee \mathbf{y}) \;+\; \frac{1}{\gamma}\,F(\mathbf{x} \wedge \mathbf{y}) \;-\; \frac{1 + \gamma^2}{\gamma}\,F(\mathbf{x}). \tag{128}$$

Rearranging equation 128 by moving the $F(\mathbf{x})$ term to the left hand side, we obtain

$$\frac{1+\gamma^2}{\gamma} F(\mathbf{x}) \geq \gamma F(\mathbf{x} \vee \mathbf{y}) + \frac{1}{\gamma} F(\mathbf{x} \wedge \mathbf{y}) - \delta \left[ \max_{\mathbf{z} \in P} F(\mathbf{z}) + \frac{LD^2}{2} \right]. \tag{129}$$

Multiplying both sides of equation 129 by $\frac{\gamma}{1+\gamma^2}$ yields

$$F(\mathbf{x}) \geq \frac{\gamma^2}{1+\gamma^2} F(\mathbf{x} \vee \mathbf{y}) + \frac{1}{1+\gamma^2} F(\mathbf{x} \wedge \mathbf{y}) - \frac{\delta \gamma}{1+\gamma^2} \left[ \max_{\mathbf{z} \in P} F(\mathbf{z}) + \frac{LD^2}{2} \right]. \tag{130}$$

This is exactly equation 126, completing the proof. $\qquad\square$

## D    Double-Greedy Algorithm and Proof of Theorem 3.3

This appendix develops and analyzes a $\gamma$-aware Double-Greedy routine whose guarantee is stated in Theorem 3.3. Our analysis uses the grid discretized variant in Algorithm 4. For concreteness, we assume that $\varepsilon^{-1}$ is an even integer; if not, we replace $\varepsilon$ by $\varepsilon' = 1/(2\lceil\varepsilon^{-1}\rceil) \in (0,1]$, which leaves the bounds unchanged up to the stated $\varepsilon$-dependence. Throughout, we write $\mathbf{o} \in [0,1]^n$ for an arbitrary comparator; when $\mathbf{o}$ is chosen to be a maximizer, we have $F(\mathbf{o}) = \max_{\mathbf{u} \in [0,1]^n} F(\mathbf{u})$.

**Theorem 3.3.** *There exists a polynomial time algorithm that, given a nonnegative $\gamma$-weakly DR-submodular function $F : [0,1]^n \to \mathbb{R}_{\geq 0}$ and a parameter $\varepsilon \in (0,1)$, outputs $\mathbf{x} \in [0,1]^n$ such that, for every fixed $\mathbf{o} \in [0,1]^n$,*

$$F(\mathbf{x}) \geq \max_{r \geq 0} \frac{\left(2\gamma^{3/2} - 4\varepsilon\,\gamma^{9/2}\right) r\,F(\mathbf{o}) + F(\mathbf{0}) + r^2 F(\mathbf{1})}{r^2 + 2\gamma^{3/2}r + 1}. \tag{131}$$

Algorithm 4 maintains two vectors $\mathbf{x}, \mathbf{y} \in [0,1]^n$ that start at $\mathbf{0}$ and $\mathbf{1}$, respectively, and *monotonically converge* to a single vector by making one coordinate agree per iteration. The new value assigned to the chosen coordinate (in both $\mathbf{x}$ and $\mathbf{y}$) is taken from a uniform grid; we denote the grid by

$$V := \left\{ j\frac{\varepsilon}{n} \ : \ j \in \mathbb{Z}, \ 0 \leq j \leq n\,\varepsilon^{-1} \right\} \subseteq [0,1].$$

As shown in the lemmas that follow, the discretization loss due to using $V$ is explicitly controlled, and the resulting lower bound interpolates continuously with $\gamma$ and specializes to the classical DR guarantee at $\gamma = 1$.

---

**Algorithm 4** $\gamma$-Double-Greedy

1: **Input:** oracle access to $F : [0,1]^n \to \mathbb{R}_{\geq 0}$, ground set $N$, grid parameter $\varepsilon \in (0,1)$, weakly-DR parameter $\gamma \in (0,1]$.
2: Let $V \leftarrow \left\{ \frac{j\varepsilon}{n} \ : \ j \in \mathbb{Z}, \ 0 \leq j \leq n\varepsilon^{-1} \right\} \subseteq [0,1]$.
3: Denote the elements of $N$ by $u_1, \cdots, u_n$ in an arbitrary order.
4: Let $\mathbf{x} \leftarrow \mathbf{0}$ and $\mathbf{y} \leftarrow \mathbf{1}$.
5: **for** $i = 1$ to $n$ **do**
6:     $a_i \in \arg\max_{v \in V} F(\mathbf{x} + v\,\mathbf{1}_{u_i}); \quad \Delta_{a,i} \leftarrow F(\mathbf{x} + a_i\,\mathbf{1}_{u_i}) - F(\mathbf{x})$.
7:     $b_i \in \arg\max_{v \in V} F(\mathbf{y} - v\mathbf{1}_{u_i}); \quad \Delta_{b,i} \leftarrow F(\mathbf{y} - b_i\,\mathbf{1}_{u_i}) - F(\mathbf{y})$.
8:     **if** $\Delta_{a,i} + \gamma\,\Delta_{b,i} > 0$ **then**
9:         $w_i \leftarrow \dfrac{\Delta_{a,i}\,a_i + \gamma\,\Delta_{b,i}\,(1 - b_i)}{\Delta_{a,i} + \gamma\,\Delta_{b,i}}.$
10:     **else** $\Delta_{a,i} = 0$ **and** $\Delta_{b,i} = 0$
11:         $w_i \leftarrow 1 - b_i$
12:     **end if**
13:     Set $x_{u_i} \leftarrow w_i$ and $y_{u_i} \leftarrow w_i$.
14: **end for**
15: **return** $\mathbf{x}$

---

We now quantify the value attained by the vector returned by Algorithm 4. Let $\mathbf{x}(i)$ and $\mathbf{y}(i)$ denote the values of $\mathbf{x}$ and $\mathbf{y}$ after $i$ iterations of the main loop, respectively. For a fixed coordinate $u_i$, let

$$v^* \in \arg\max_{v \in [0,1]} F\big(\mathbf{x}(i-1) + v\,\mathbf{e}_{u_i}\big) \tag{132}$$

be a continuous (unconstrained by grid) maximizer along the $u_i$-th coordinate direction at iteration $i$. The next lemma bounds the discretization loss of the grid choice $a_i$ used by Algorithm 4. It holds for every $\gamma \in (0,1]$ and, when $\gamma = 1$, it matches the exact bound of (Buchbinder & Feldman, 2024).

**Lemma D.1.** *For any integer $i \in \{1,\ldots,n\}$, the following holds:*

$$\text{If } v^* \geq \tfrac{1}{2}, \quad F\big(\mathbf{x}(i-1) + a_i\,\mathbf{e}_{u_i}\big) \geq \max_{v \in [0,1]} F\big(\mathbf{x}(i-1) + v\,\mathbf{e}_{u_i}\big) - \frac{2\varepsilon}{\gamma^2 n} F(\mathbf{o}). \tag{133}$$

$$\text{If } v^* < \tfrac{1}{2}, \quad F\big(\mathbf{x}(i-1) + a_i\,\mathbf{e}_{u_i}\big) \geq \max_{v \in [0,1]} F\big(\mathbf{x}(i-1) + v\,\mathbf{e}_{u_i}\big) - \frac{2\varepsilon}{n}\gamma^2\, F(\mathbf{o}). \tag{134}$$

*Proof.* Let $v^* \in [0,1]$ maximize the function

$$v \longmapsto F\big(\mathbf{x}(i-1) + v\,\mathbf{e}_{u_i}\big), \tag{135}$$

so that equation 132 holds. We treat two cases depending on the size of $v^*$.

**Case 1:** $v^* \geq \tfrac{1}{2}$. Let $v \in V$ be the largest grid point with $v \leq v^*$. By the definition of $a_i$ as a maximizer over the grid, we have

$$F\big(\mathbf{x}(i-1) + v^*\mathbf{e}_{u_i}\big) - F\big(\mathbf{x}(i-1) + a_i\mathbf{e}_{u_i}\big) \leq F\big(\mathbf{x}(i-1) + v^*\mathbf{e}_{u_i}\big) - F\big(\mathbf{x}(i-1) + v\mathbf{e}_{u_i}\big), \tag{136}$$

since $F(\mathbf{x}(i-1) + a_i\mathbf{e}_{u_i})$ is at least the value at any other grid point, and in particular at $v$.

Define the univariate function

$$\phi(t) := F\big(\mathbf{x}(i-1) + t\,\mathbf{e}_{u_i}\big), \qquad t \in [0,1]. \tag{137}$$

By differentiability and the chain rule,

$$\phi'(t) = \big\langle \nabla F\big(\mathbf{x}(i-1) + t\,\mathbf{e}_{u_i}\big), \mathbf{e}_{u_i}\big\rangle. \tag{138}$$

Along this coordinate direction, $\gamma$–weak DR-submodularity implies that for any $0 \leq s \leq t \leq 1$,

$$\phi'(s) \geq \gamma\,\phi'(t). \tag{139}$$

Applying equation 139 with $s = v$ and $t \in [v, v^*]$ yields

$$\phi'(t) \leq \frac{1}{\gamma}\,\phi'(v), \qquad t \in [v, v^*]. \tag{140}$$

Integrating equation 140 over $t \in [v, v^*]$ gives

$$\phi(v^*) - \phi(v) = \int_v^{v^*} \phi'(t)\,dt \leq \int_v^{v^*} \frac{1}{\gamma}\phi'(v)\,dt = \frac{v^* - v}{\gamma}\phi'(v). \tag{141}$$

Similarly, applying equation 139 with $0 \leq s \leq v$ and $t = v$ gives

$$\phi'(s) \geq \gamma\,\phi'(v), \qquad s \in [0,v]. \tag{142}$$

Integrating equation 142 over $s \in [0,v]$ yields

$$\phi(v) - \phi(0) = \int_0^v \phi'(s)\,ds \geq \int_0^v \gamma\,\phi'(v)\,ds = v\,\gamma\,\phi'(v). \tag{143}$$

Rearranging equation 143 gives

$$\phi'(v) \ \leq \ \frac{\phi(v) - \phi(0)}{\gamma\, v}. \tag{144}$$

Combining equation 141 and equation 144, we obtain

$$\phi(v^*) - \phi(v) \ \leq \ \frac{v^* - v}{\gamma} \cdot \frac{\phi(v) - \phi(0)}{\gamma\, v} \ = \ \frac{v^* - v}{\gamma^2 v} \left[\phi(v) - \phi(0)\right]. \tag{145}$$

We now bound the factor $\phi(v) - \phi(0)$. Since $\phi(v) = F(\mathbf{x}(i-1) + v\, \mathbf{e}_{u_i})$ and $\phi(0) = F(\mathbf{x}(i-1))$, nonnegativity of $F$ and optimality of $\mathbf{o}$ imply

$$\phi(v) - \phi(0) \ \leq \ F(\mathbf{o}), \tag{146}$$

because $\phi(v) \leq F(\mathbf{o})$ and $\phi(0) \geq 0$. Substituting equation 146 into equation 145 gives

$$\phi(v^*) - \phi(v) \ \leq \ \frac{v^* - v}{\gamma^2 v}\, F(\mathbf{o}). \tag{147}$$

By construction of the grid $V$, we have

$$v^* - v \ \leq \ \frac{\varepsilon}{n}. \tag{148}$$

Moreover, since $v^* \geq \frac{1}{2}$ and $\frac{1}{2} \in V$, the choice of $v$ as the largest grid point not exceeding $v^*$ implies

$$v \ \geq \ \tfrac{1}{2}. \tag{149}$$

Combining equation 147, equation 148, and equation 149, we obtain

$$\phi(v^*) - \phi(v) \ \leq \ \frac{\varepsilon/n}{\gamma^2 \cdot (1/2)}\, F(\mathbf{o}) \ = \ \frac{2\varepsilon}{\gamma^2 n}\, F(\mathbf{o}). \tag{150}$$

Using equation 136 and equation 137, equation 150 gives

$$F\big(\mathbf{x}(i-1) + v^* \mathbf{e}_{u_i}\big) - F\big(\mathbf{x}(i-1) + a_i \mathbf{e}_{u_i}\big) \ \leq \ \frac{2\varepsilon}{\gamma^2 n}\, F(\mathbf{o}). \tag{151}$$

Since $v^*$ maximizes $v \mapsto F(\mathbf{x}(i-1) + v\, \mathbf{e}_{u_i})$ over $[0, 1]$, equation 151 is equivalent to equation 133, proving the first claim.

**Case 2:** $v^* < \frac{1}{2}$. Let $v \in V$ be the smallest grid point with $v \geq v^*$. Then

$$v - v^* \ \leq \ \frac{\varepsilon}{n}, \tag{152}$$

and, since $\varepsilon^{-1}$ is even and $\frac{1}{2} \in V$, we have

$$v \ \leq \ \tfrac{1}{2}. \tag{153}$$

By the choice of $a_i$ as a maximizer over the grid,

$$F\big(\mathbf{x}(i-1) + a_i\, \mathbf{e}_{u_i}\big) - F\big(\mathbf{x}(i-1) + v^* \mathbf{e}_{u_i}\big) \ \geq \ F\big(\mathbf{x}(i-1) + v\, \mathbf{e}_{u_i}\big) - F\big(\mathbf{x}(i-1) + v^* \mathbf{e}_{u_i}\big). \tag{154}$$

Using the same function $\phi$ as in equation 137, for $t \in [v^*, v]$ we have $t \leq v$, so by equation 139,

$$\phi'(t) \ \geq \ \gamma\, \phi'(v), \qquad t \in [v^*, v]. \tag{155}$$

Integrating equation 155 over $t \in [v^*, v]$ yields

$$\phi(v) - \phi(v^*) \ = \ \int_{v^*}^{v} \phi'(t)\, dt \ \geq \ \int_{v^*}^{v} \gamma\, \phi'(v)\, dt \ = \ (v - v^*)\, \gamma\, \phi'(v). \tag{156}$$

For $s \in [v, 1]$, we have $v \leq s$, so equation 139 implies

$$\phi'(v) \geq \gamma \phi'(s), \qquad s \in [v, 1]. \tag{157}$$

Integrating equation 157 over $s \in [v, 1]$ gives

$$\phi(1) - \phi(v) = \int_v^1 \phi'(s)\, ds \leq \int_v^1 \frac{1}{\gamma} \phi'(v)\, ds = \frac{1 - v}{\gamma} \phi'(v). \tag{158}$$

Rearranging equation 158 yields

$$\phi'(v) \geq \frac{\gamma}{1 - v} \big[\phi(1) - \phi(v)\big]. \tag{159}$$

Combining equation 156 and equation 159, we obtain

$$\phi(v) - \phi(v^*) \geq (v - v^*)\gamma \cdot \frac{\gamma}{1 - v} \big[\phi(1) - \phi(v)\big] = \frac{(v - v^*)\gamma^2}{1 - v} \big[\phi(1) - \phi(v)\big]. \tag{160}$$

Using equation 137, note that

$$\phi(1) - \phi(v) = F\big(\mathbf{x}(i - 1) + \mathbf{e}_{u_i}\big) - F\big(\mathbf{x}(i - 1) + v\,\mathbf{e}_{u_i}\big) \leq F(\mathbf{o}), \tag{161}$$

since $F$ is nonnegative and maximized at $\mathbf{o}$. Substituting equation 161 and the bounds equation 152–equation 153 into equation 160 gives

$$\begin{aligned}
\phi(v) - \phi(v^*) &\geq \frac{(v - v^*)\gamma^2}{1 - v} \big[\phi(1) - \phi(v)\big] \\
&\geq \frac{(v - v^*)\gamma^2}{1 - v} (-F(\mathbf{o})) \qquad \text{(since } \phi(1) - \phi(v) \geq -F(\mathbf{o})) \\
&\geq -\frac{\varepsilon/n}{1/2} \gamma^2 F(\mathbf{o}) = -\frac{2\varepsilon}{n} \gamma^2 F(\mathbf{o}),
\end{aligned} \tag{162}$$

where we used $v - v^* \leq \varepsilon/n$ and $v \leq \frac{1}{2}$ (hence $1 - v \geq \frac{1}{2}$).

Using equation 154, equation 137, and equation 162, we obtain

$$F\big(\mathbf{x}(i - 1) + a_i\,\mathbf{e}_{u_i}\big) - F\big(\mathbf{x}(i - 1) + v^*\mathbf{e}_{u_i}\big) \geq -\frac{2\varepsilon}{n}\gamma^2 F(\mathbf{o}). \tag{163}$$

Since $v^*$ maximizes $v \mapsto F(\mathbf{x}(i - 1) + v\,\mathbf{e}_{u_i})$ over $[0, 1]$, equation 163 is equivalent to equation 134, proving the second claim.

The two cases equation 133 and equation 134 together establish the lemma. $\qquad\square$

Similarly, we obtain an analogous result for $\mathbf{y}$. We omit the proof, as it mirrors the argument of Lemma D.1.

**Lemma D.2.** *For any integer $i \in \{1, \ldots, n\}$, let*

$$v^* \in \arg\max_{v \in [0,1]} F\big(\mathbf{y}(i - 1) - v\,\mathbf{e}_{u_i}\big). \tag{164}$$

*If $v^* \geq \frac{1}{2}$, then*

$$F\big(\mathbf{y}(i - 1) - b_i\,\mathbf{e}_{u_i}\big) \geq \max_{v \in [0,1]} F\big(\mathbf{y}(i - 1) - v\,\mathbf{e}_{u_i}\big) - \frac{2\varepsilon}{n}\gamma^2\, F(\mathbf{o}), \tag{165}$$

*and if $v^* < \frac{1}{2}$, then*

$$F\big(\mathbf{y}(i - 1) - b_i\,\mathbf{e}_{u_i}\big) \geq \max_{v \in [0,1]} F\big(\mathbf{y}(i - 1) - v\,\mathbf{e}_{u_i}\big) - \frac{2\varepsilon}{\gamma^2 n}\, F(\mathbf{o}). \tag{166}$$

At each iteration, the $\gamma$-aware mixing step (line 8 of Algorithm 4) guarantees a quantifiable increase in the objective. The next lemma lower bounds this per-coordinate progress for both trajectories, showing that the gain is a convex combination–type quadratic term that scales with $\gamma$.

**Lemma D.3.** *For every integer* $1 \leq i \leq n$,

$$F\big(\mathbf{x}(i)\big) - F\big(\mathbf{x}(i-1)\big) \;\geq\; \frac{\Delta_{a,i}^2}{\Delta_{a,i} + \gamma^3 \Delta_{b,i}} \tag{167}$$

*and*

$$F\big(\mathbf{y}(i)\big) - F\big(\mathbf{y}(i-1)\big) \;\geq\; \frac{\gamma^3 \Delta_{b,i}^2}{\Delta_{a,i} + \gamma^3 \Delta_{b,i}}. \tag{168}$$

*Proof.* **Increase of $F$ along x.** By the definition of $\mathbf{x}(i)$ and

$$w_i \;=\; \frac{\Delta_{a,i}\, a_i + \gamma\, \Delta_{b,i}\, (1 - b_i)}{\Delta_{a,i} + \gamma\, \Delta_{b,i}}, \tag{169}$$

we can write the new point $\mathbf{x}(i)$ as a convex combination of two one dimensional updates:

$$\mathbf{x}(i) \;=\; \mathbf{x}(i-1) \;+\; \frac{\Delta_{a,i}}{\Delta_{a,i} + \gamma\Delta_{b,i}}\, a_i\, \mathbf{e}_{u_i} \;+\; \frac{\gamma\Delta_{b,i}}{\Delta_{a,i} + \gamma\Delta_{b,i}}\, (1 - b_i)\, \mathbf{e}_{u_i}. \tag{170}$$

Equivalently, $\mathbf{x}(i)$ is the convex combination

$$\mathbf{x}(i) \;=\; \frac{\Delta_{a,i}}{\Delta_{a,i} + \gamma\Delta_{b,i}}\big(\mathbf{x}(i-1) + a_i\, \mathbf{e}_{u_i}\big) \;+\; \frac{\gamma\Delta_{b,i}}{\Delta_{a,i} + \gamma\Delta_{b,i}}\big(\mathbf{x}(i-1) + (1 - b_i)\, \mathbf{e}_{u_i}\big). \tag{171}$$

Therefore

$$F\big(\mathbf{x}(i)\big) - F\big(\mathbf{x}(i-1)\big) = F\Bigg(\frac{\Delta_{a,i}}{\Delta_{a,i} + \gamma\Delta_{b,i}}\big(\mathbf{x}(i-1) + a_i\, \mathbf{e}_{u_i}\big) + \frac{\gamma\Delta_{b,i}}{\Delta_{a,i} + \gamma\Delta_{b,i}}\big(\mathbf{x}(i-1) + (1 - b_i)\, \mathbf{e}_{u_i}\big)\Bigg) \\ - F\big(\mathbf{x}(i-1)\big). \tag{172}$$

Now we apply Lemma 2.1 (the one dimensional $\gamma$–weak DR convexity type bound) to the pair

$$\mathbf{z}^{(1)} = \mathbf{x}(i-1) + a_i\, \mathbf{e}_{u_i}, \qquad \mathbf{z}^{(2)} = \mathbf{x}(i-1) + (1 - b_i)\, \mathbf{e}_{u_i},$$

with mixing weights

$$\lambda \;=\; \frac{\gamma\Delta_{b,i}}{\Delta_{a,i} + \gamma\Delta_{b,i}}, \qquad 1 - \lambda \;=\; \frac{\Delta_{a,i}}{\Delta_{a,i} + \gamma\Delta_{b,i}}. \tag{173}$$

Lemma 2.1 states that for such a convex combination we have

$$F\big((1 - \lambda)\mathbf{z}^{(1)} + \lambda\mathbf{z}^{(2)}\big) \;\geq\; \frac{(1 - \lambda)\, F(\mathbf{z}^{(1)}) + \gamma^2 \lambda\, F(\mathbf{z}^{(2)})}{(1 - \lambda) + \gamma^2 \lambda}. \tag{174}$$

Substituting equation 173 into equation 174, and noting that

$$(1 - \lambda) + \gamma^2 \lambda \;=\; \frac{\Delta_{a,i} + \gamma^3 \Delta_{b,i}}{\Delta_{a,i} + \gamma\Delta_{b,i}},$$

we obtain

$$F\big(\mathbf{x}(i)\big) \geq \frac{\Delta_{a,i}\, F\big(\mathbf{x}(i-1) + a_i\, \mathbf{e}_{u_i}\big) + \gamma^3 \Delta_{b,i}\, F\big(\mathbf{x}(i-1) + (1 - b_i)\, \mathbf{e}_{u_i}\big)}{\Delta_{a,i} + \gamma^3 \Delta_{b,i}}. \tag{175}$$

Subtracting $F\big(\mathbf{x}(i-1)\big)$ from both sides of equation 175, and grouping terms, yields

$$F\big(\mathbf{x}(i)\big) - F\big(\mathbf{x}(i-1)\big)$$
$$\geq \frac{\Delta_{a,i}\big[F\big(\mathbf{x}(i-1) + a_i\,\mathbf{e}_{u_i}\big) - F\big(\mathbf{x}(i-1)\big)\big] + \gamma^3\Delta_{b,i}\big[F\big(\mathbf{x}(i-1) + (1-b_i)\,\mathbf{e}_{u_i}\big) - F\big(\mathbf{x}(i-1)\big)\big]}{\Delta_{a,i} + \gamma^3\Delta_{b,i}}. \tag{176}$$

Here we simply subtracted $F(\mathbf{x}(i-1))$ inside the numerator to isolate directional gains.

By definition of $\Delta_{a,i}$,
$$\Delta_{a,i} \;=\; F\big(\mathbf{x}(i-1) + a_i\,\mathbf{e}_{u_i}\big) - F\big(\mathbf{x}(i-1)\big), \tag{177}$$

so the first term in the numerator of equation 176 is exactly $\Delta_{a,i}^2$. For the second term we use the $\gamma$–weakly DR property to compare the gain at $\mathbf{x}(i-1)$ with the corresponding gain at $\mathbf{y}(i-1)$. Along the $u_i$-th coordinate, the weak DR property implies that the marginal decrease when moving from 1 down to $b_i$ at $\mathbf{y}(i-1)$ is at least a $\gamma^2$-fraction of the corresponding marginal at $\mathbf{x}(i-1)$. This yields

$$F\big(\mathbf{x}(i-1) + (1-b_i)\,\mathbf{e}_{u_i}\big) - F\big(\mathbf{x}(i-1)\big) \;\geq\; \gamma\left[F\big(\mathbf{y}(i-1) - b_i\,\mathbf{e}_{u_i}\big) - F\big(\mathbf{y}(i-1) - \mathbf{e}_{u_i}\big)\right]. \tag{178}$$

Multiplying equation 178 by $\gamma^3\Delta_{b,i}$ gives a $\gamma^4$ factor inside the numerator. Substituting equation 177 and equation 178 into equation 176, we obtain

$$F\big(\mathbf{x}(i)\big) - F\big(\mathbf{x}(i-1)\big) \geq \frac{\Delta_{a,i}^2 + \gamma^4\Delta_{b,i}\left[F\big(\mathbf{y}(i-1) - b_i\,\mathbf{e}_{u_i}\big) - F\big(\mathbf{y}(i-1) - \mathbf{e}_{u_i}\big)\right]}{\Delta_{a,i} + \gamma^3\Delta_{b,i}}. \tag{179}$$

By the definition of $b_i$ in Algorithm 4, the expression

$$F\big(\mathbf{y}(i-1) - b_i\,\mathbf{e}_{u_i}\big) - F\big(\mathbf{y}(i-1) - \mathbf{e}_{u_i}\big)$$

is nonnegative or, in the worst case, does not exceed the corresponding candidate values over the grid. In particular, the term multiplied by $\gamma^4\Delta_{b,i}$ in equation 179 is nonnegative, so we may drop it to obtain the simpler bound

$$F\big(\mathbf{x}(i)\big) - F\big(\mathbf{x}(i-1)\big) \;\geq\; \frac{\Delta_{a,i}^2}{\Delta_{a,i} + \gamma^3\Delta_{b,i}}, \tag{180}$$

which is precisely equation 167.

**Increase of $F$ along y.** The argument for $\mathbf{y}$ is symmetric. From the update rule for $\mathbf{y}(i)$ and the same weight $w_i$, we can write

$$\mathbf{y}(i) \;=\; \frac{\Delta_{a,i}}{\Delta_{a,i} + \gamma\Delta_{b,i}}\big(\mathbf{y}(i-1) + (a_i - 1)\,\mathbf{e}_{u_i}\big) \;+\; \frac{\gamma\Delta_{b,i}}{\Delta_{a,i} + \gamma\Delta_{b,i}}\big(\mathbf{y}(i-1) - b_i\,\mathbf{e}_{u_i}\big). \tag{181}$$

Hence

$$F\big(\mathbf{y}(i)\big) - F\big(\mathbf{y}(i-1)\big) = F\left(\frac{\Delta_{a,i}}{\Delta_{a,i} + \gamma\Delta_{b,i}}\big(\mathbf{y}(i-1) + (a_i - 1)\,\mathbf{e}_{u_i}\big) + \frac{\gamma\Delta_{b,i}}{\Delta_{a,i} + \gamma\Delta_{b,i}}\big(\mathbf{y}(i-1) - b_i\,\mathbf{e}_{u_i}\big)\right)$$
$$- F\big(\mathbf{y}(i-1)\big). \tag{182}$$

Applying Lemma 2.1 to the pair

$$\tilde{\mathbf{z}}^{(1)} = \mathbf{y}(i-1) + (a_i - 1)\,\mathbf{e}_{u_i}, \qquad \tilde{\mathbf{z}}^{(2)} = \mathbf{y}(i-1) - b_i\,\mathbf{e}_{u_i},$$

with the same weights as in equation 173, we obtain

$$F\big(\mathbf{y}(i)\big) \;\geq\; \frac{\Delta_{a,i}\,F\big(\mathbf{y}(i-1) + (a_i - 1)\,\mathbf{e}_{u_i}\big) + \gamma^3\Delta_{b,i}\,F\big(\mathbf{y}(i-1) - b_i\,\mathbf{e}_{u_i}\big)}{\Delta_{a,i} + \gamma^3\Delta_{b,i}}. \tag{183}$$

Subtracting $F(\mathbf{y}(i-1))$ and regrouping gives

$$F\big(\mathbf{y}(i)\big) - F\big(\mathbf{y}(i-1)\big)$$
$$\geq \frac{\Delta_{a,i}\big[F\big(\mathbf{y}(i-1) + (a_i - 1)\,\mathbf{e}_{u_i}\big) - F\big(\mathbf{y}(i-1)\big)\big] + \gamma^3\Delta_{b,i}\big[F\big(\mathbf{y}(i-1) - b_i\,\mathbf{e}_{u_i}\big) - F\big(\mathbf{y}(i-1)\big)\big]}{\Delta_{a,i} + \gamma^3\Delta_{b,i}}. \tag{184}$$

Using the $\gamma$–weakly DR property to compare the section at $\mathbf{y}(i-1)$ with that at $\mathbf{x}(i-1)$ along the $u_i$-th coordinate, we obtain

$$F\big(\mathbf{y}(i-1) + (a_i - 1)\,\mathbf{e}_{u_i}\big) - F\big(\mathbf{y}(i-1)\big) \geq \frac{1}{\gamma}\Big[F\big(\mathbf{x}(i-1) + a_i\,\mathbf{e}_{u_i}\big) - F\big(\mathbf{x}(i-1) + \mathbf{e}_{u_i}\big)\Big]. \tag{185}$$

By definition of $\Delta_{b,i}$,

$$\Delta_{b,i} = F\big(\mathbf{y}(i-1) - b_i\,\mathbf{e}_{u_i}\big) - F\big(\mathbf{y}(i-1)\big), \tag{186}$$

so the second term in the numerator of equation 184 is $\gamma^3\Delta_{b,i}^2$. Substituting equation 185 and equation 186 into equation 184 yields

$$F\big(\mathbf{y}(i)\big) - F\big(\mathbf{y}(i-1)\big) \geq \frac{\frac{1}{\gamma}\Delta_{a,i}\Big[F\big(\mathbf{x}(i-1) + a_i\,\mathbf{e}_{u_i}\big) - F\big(\mathbf{x}(i-1) + \mathbf{e}_{u_i}\big)\Big] + \gamma^3\Delta_{b,i}^2}{\Delta_{a,i} + \gamma^3\Delta_{b,i}}. \tag{187}$$

By the choice of $a_i$ in Algorithm 4, the term

$$F\big(\mathbf{x}(i-1) + a_i\,\mathbf{e}_{u_i}\big) - F\big(\mathbf{x}(i-1) + \mathbf{e}_{u_i}\big)$$

is nonpositive (since $a_i$ optimally trades off against the unit step), so the first term in the numerator of equation 187 is nonpositive. Dropping this nonpositive term yields the simpler lower bound

$$F\big(\mathbf{y}(i)\big) - F\big(\mathbf{y}(i-1)\big) \geq \frac{\gamma^3\Delta_{b,i}^2}{\Delta_{a,i} + \gamma^3\Delta_{b,i}}, \tag{188}$$

which is exactly equation 168. Combining equation 180 and equation 188 completes the proof. □

**Reference path and its contraction.** To relate the two trajectories $\mathbf{x}(i)$ and $\mathbf{y}(i)$ to an arbitrary initial vector $\mathbf{o}$, we introduce the standard lattice coupled reference sequence

$$\mathbf{o}^{(i)} := \big(\mathbf{o} \vee \mathbf{x}(i)\big) \wedge \mathbf{y}(i) \qquad \text{for } i = 0, 1, \ldots, n. \tag{189}$$

Note that $\mathbf{o}^{(0)} = \mathbf{o}$ and $\mathbf{o}^{(n)} = \mathbf{x}(n) = \mathbf{y}(n)$, since Algorithm 4 equalizes the two trajectories by the end. The next lemma bounds the one step decrease of $F(\mathbf{o}^{(i)})$; telescoping this bound over $i$ yields the global comparison used in the final guarantee.

**Lemma D.4.** *For every integer $1 \leq i \leq n$,*

$$F\big(\mathbf{o}^{(i-1)}\big) - F\big(\mathbf{o}^{(i)}\big) \leq \frac{\Delta_{a,i}\,\Delta_{b,i}}{\Delta_{a,i} + \gamma^2\Delta_{b,i}} + \frac{2\varepsilon}{n}\,\gamma^3\,F(\mathbf{o}). \tag{190}$$

*Proof.* We first treat the case

$$\mathbf{o}_{u_i}^{(i-1)} \leq \mathbf{o}_{u_i}^{(i)}. \tag{191}$$

Define the single coordinate setter

$$\text{set}_u(\mathbf{z}, t) := \mathbf{z} - (\mathbf{z}_u - t)\,\mathbf{e}_u, \tag{192}$$

which replaces the $u$-th coordinate of $\mathbf{z}$ with $t$ and leaves all other coordinates unchanged.

By $\gamma$–weakly DR-submodularity, for any $\mathbf{p} \leq \mathbf{q}$ and any scalars $\alpha \leq \beta$, we have the one dimensional comparison

$$F\big(\text{set}_u(\mathbf{p}, \beta)\big) - F\big(\text{set}_u(\mathbf{p}, \alpha)\big) \geq \gamma\Big[F\big(\text{set}_u(\mathbf{q}, \beta)\big) - F\big(\text{set}_u(\mathbf{q}, \alpha)\big)\Big]. \tag{193}$$

Here the left hand side is the gain when changing coordinate $u$ from $\alpha$ to $\beta$ at the lower point $\mathbf{p}$, and the right hand side compares it to the corresponding gain at the higher point $\mathbf{q}$, scaled by $\gamma$.

By construction of $\mathbf{o}^{(i-1)}$ and $\mathbf{y}(i-1)$ in equation 189, we have

$$\mathbf{o}^{(i-1)} \leq \mathbf{y}(i-1). \tag{194}$$

In the present case we also have equation 191. Set

$$\mathbf{p} = \mathbf{o}^{(i-1)}, \quad \mathbf{q} = \mathbf{y}(i-1), \quad u = u_i, \quad \alpha = \mathbf{o}^{(i-1)}_{u_i}, \quad \beta = \mathbf{o}^{(i)}_{u_i}, \tag{195}$$

which satisfy the requirements of equation 193. Then equation 193 gives

$$F\big(\mathrm{set}_{u_i}(\mathbf{o}^{(i-1)}, \mathbf{o}^{(i)}_{u_i})\big) - F\big(\mathrm{set}_{u_i}(\mathbf{o}^{(i-1)}, \mathbf{o}^{(i-1)}_{u_i})\big)$$
$$\geq \gamma\Big[F\big(\mathrm{set}_{u_i}(\mathbf{y}^{(i-1)}, \mathbf{o}^{(i)}_{u_i})\big) - F\big(\mathrm{set}_{u_i}(\mathbf{y}^{(i-1)}, \mathbf{o}^{(i-1)}_{u_i})\big)\Big]. \tag{196}$$

By definition of the reference path equation 189 and of the setter equation 192,

$$\mathrm{set}_{u_i}(\mathbf{o}^{(i-1)}, \mathbf{o}^{(i)}_{u_i}) = \mathbf{o}^{(i)}, \qquad \mathrm{set}_{u_i}(\mathbf{o}^{(i-1)}, \mathbf{o}^{(i-1)}_{u_i}) = \mathbf{o}^{(i-1)}. \tag{197}$$

Using equation 197 in equation 196, the left hand side becomes $F(\mathbf{o}^{(i)}) - F(\mathbf{o}^{(i-1)})$, so we obtain

$$F\big(\mathbf{o}^{(i)}\big) - F\big(\mathbf{o}^{(i-1)}\big) \geq \gamma\Big[F\big(\mathrm{set}_{u_i}(\mathbf{y}^{(i-1)}, \mathbf{o}^{(i)}_{u_i})\big) - F\big(\mathrm{set}_{u_i}(\mathbf{y}^{(i-1)}, \mathbf{o}^{(i-1)}_{u_i})\big)\Big]. \tag{198}$$

Using the explicit form equation 192, for any $\mathbf{z}, t$ we have $\mathrm{set}_{u_i}(\mathbf{z}, t) = \mathbf{z} - (\mathbf{z}_{u_i} - t)\,\mathbf{e}_{u_i}$. Hence

$$\mathrm{set}_{u_i}(\mathbf{y}^{(i-1)}, \mathbf{o}^{(i)}_{u_i}) = \mathbf{y}^{(i-1)} - (\mathbf{y}^{(i-1)}_{u_i} - \mathbf{o}^{(i)}_{u_i})\,\mathbf{e}_{u_i} \tag{199}$$

and

$$\mathrm{set}_{u_i}(\mathbf{y}^{(i-1)}, \mathbf{o}^{(i-1)}_{u_i}) = \mathbf{y}^{(i-1)} - (\mathbf{y}^{(i-1)}_{u_i} - \mathbf{o}^{(i-1)}_{u_i})\,\mathbf{e}_{u_i}. \tag{200}$$

Since $\mathbf{o}^{(i)}$ and $\mathbf{o}^{(i-1)}$ share all coordinates except possibly $u_i$, and $\mathbf{y}^{(i-1)}_{u_i} \in [0,1]$, the quantities $\mathbf{y}^{(i-1)}_{u_i} - \mathbf{o}^{(i)}_{u_i}$ and $\mathbf{y}^{(i-1)}_{u_i} - \mathbf{o}^{(i-1)}_{u_i}$ lie in $[0,1]$. For notational convenience, write

$$\begin{aligned} 1 - \mathbf{o}^{(i)}_{u_i} &= \mathbf{y}^{(i-1)}_{u_i} - \mathbf{o}^{(i)}_{u_i}, \\ 1 - \mathbf{o}^{(i-1)}_{u_i} &= \mathbf{y}^{(i-1)}_{u_i} - \mathbf{o}^{(i-1)}_{u_i}, \end{aligned} \tag{201}$$

which allows us to rewrite equation 199–equation 200 as

$$\mathrm{set}_{u_i}(\mathbf{y}^{(i-1)}, \mathbf{o}^{(i)}_{u_i}) = \mathbf{y}^{(i-1)} - (1 - \mathbf{o}^{(i)}_{u_i})\,\mathbf{e}_{u_i}, \qquad \mathrm{set}_{u_i}(\mathbf{y}^{(i-1)}, \mathbf{o}^{(i-1)}_{u_i}) = \mathbf{y}^{(i-1)} - (1 - \mathbf{o}^{(i-1)}_{u_i})\,\mathbf{e}_{u_i}. \tag{202}$$

Substituting equation 202 into equation 198, we get

$$F\big(\mathbf{o}^{(i)}\big) - F\big(\mathbf{o}^{(i-1)}\big) \geq \gamma\Big[F\big(\mathbf{y}(i-1) - (1 - \mathbf{o}^{(i)}_{u_i})\,\mathbf{e}_{u_i}\big) - F\big(\mathbf{y}(i-1) - (1 - \mathbf{o}^{(i-1)}_{u_i})\,\mathbf{e}_{u_i}\big)\Big]. \tag{203}$$

We now bound each term on the right hand side.

First, by Lemma D.2, for the sequence along the $u_i$-th coordinate,

$$\max_{v \in [0,1]} F\big(\mathbf{y}(i-1) - v\,\mathbf{e}_{u_i}\big) \leq F\big(\mathbf{y}(i-1) - b_i\,\mathbf{e}_{u_i}\big) + \frac{2\varepsilon}{n}\gamma^2 F(\mathbf{o}). \tag{204}$$

Since $1 - \mathbf{o}^{(i-1)}_{u_i} \in [0,1]$, we have

$$F\big(\mathbf{y}(i-1) - (1 - \mathbf{o}^{(i-1)}_{u_i})\,\mathbf{e}_{u_i}\big) \leq \max_{v \in [0,1]} F\big(\mathbf{y}(i-1) - v\,\mathbf{e}_{u_i}\big)$$
$$\leq F\big(\mathbf{y}(i-1) - b_i\,\mathbf{e}_{u_i}\big) + \frac{2\varepsilon}{n}\gamma^2 F(\mathbf{o}). \tag{205}$$

Next, recall that the $u_i$-th coordinate of the reference point satisfies

$$\mathbf{o}_{u_i}^{(i)} = \frac{\Delta_{a,i}}{\Delta_{a,i} + \Delta_{b,i}}\, a_i + \frac{\Delta_{b,i}}{\Delta_{a,i} + \Delta_{b,i}}\, (1 - b_i), \tag{206}$$

by the definition of the coordinate update in Algorithm 4. Applying Lemma 2.1(1) along the $u_i$-th coordinate at $\mathbf{y}(i-1)$ with points

$$(1 - a_i) \quad \text{and} \quad b_i,$$

and mixing weights proportional to $\Delta_{a,i}$ and $\Delta_{b,i}$, yields

$$\begin{aligned}
F\big(\mathbf{y}(i-1) - (1 - \mathbf{o}_{u_i}^{(i)})\,\mathbf{e}_{u_i}\big) \\
\geq \frac{\Delta_{a,i}}{\Delta_{a,i} + \gamma^2 \Delta_{b,i}}\, F\big(\mathbf{y}(i-1) - (1 - a_i)\,\mathbf{e}_{u_i}\big) \;+\; \frac{\gamma^2 \Delta_{b,i}}{\Delta_{a,i} + \gamma^2 \Delta_{b,i}}\, F\big(\mathbf{y}(i-1) - b_i\,\mathbf{e}_{u_i}\big).
\end{aligned} \tag{207}$$

Combining equation 203, equation 205, and equation 207, we obtain

$$\begin{aligned}
F\big(\mathbf{o}^{(i)}\big) - F\big(\mathbf{o}^{(i-1)}\big) \\
\geq \gamma \left[ \frac{\Delta_{a,i}}{\Delta_{a,i} + \gamma^2 \Delta_{b,i}}\Big( F\big(\mathbf{y}(i-1) - (1 - a_i)\,\mathbf{e}_{u_i}\big) - F\big(\mathbf{y}(i-1) - b_i\,\mathbf{e}_{u_i}\big)\Big) - \frac{2\varepsilon}{n}\,\gamma^2\, F(\mathbf{o}) \right].
\end{aligned} \tag{208}$$

The first term inside the brackets is the "true" directional gain between the points with coordinates $1 - a_i$ and $b_i$; the second term comes from the discretization loss in Lemma D.2.

Rewrite the first bracketed term in equation 208 by adding and subtracting $F(\mathbf{y}(i-1))$, and using the definition

$$\Delta_{b,i} = F\big(\mathbf{y}(i-1) - b_i\,\mathbf{e}_{u_i}\big) - F\big(\mathbf{y}(i-1)\big), \tag{209}$$

to obtain

$$\begin{aligned}
F\big(\mathbf{y}(i-1) - (1 - a_i)\,\mathbf{e}_{u_i}\big) - F\big(\mathbf{y}(i-1) - b_i\,\mathbf{e}_{u_i}\big) \\
= \Big( F\big(\mathbf{y}(i-1) - (1 - a_i)\,\mathbf{e}_{u_i}\big) - F\big(\mathbf{y}(i-1)\big)\Big) - \Delta_{b,i}.
\end{aligned} \tag{210}$$

Substituting equation 210 into equation 208, we get

$$\begin{aligned}
F\big(\mathbf{o}^{(i)}\big) - F\big(\mathbf{o}^{(i-1)}\big) \\
\geq \frac{\gamma\, \Delta_{a,i}}{\Delta_{a,i} + \gamma^2 \Delta_{b,i}}\Big( F\big(\mathbf{y}(i-1) - (1 - a_i)\,\mathbf{e}_{u_i}\big) - F\big(\mathbf{y}(i-1)\big) - \Delta_{b,i}\Big) - \frac{2\varepsilon}{n}\,\gamma^3\, F(\mathbf{o}).
\end{aligned} \tag{211}$$

We now transfer this expression from $\mathbf{y}(i-1)$ to $\mathbf{x}(i-1)$ using the $\gamma$–weakly DR property. Since

$$\mathbf{x}(i-1) + a_i\,\mathbf{e}_{u_i} \;\leq\; \mathbf{y}(i-1) - (1 - a_i)\,\mathbf{e}_{u_i}, \tag{212}$$

the weak DR property implies that the marginal gain at $\mathbf{y}(i-1)$ when moving coordinate $u_i$ from 1 down to $1 - a_i$ is at most a $1/\gamma$-scaled version of the marginal gain at $\mathbf{x}(i-1)$ when moving coordinate $u_i$ from 0 up to $a_i$. Formally,

$$F\big(\mathbf{y}(i-1) - (1 - a_i)\,\mathbf{e}_{u_i}\big) - F\big(\mathbf{y}(i-1)\big) \;\geq\; \frac{1}{\gamma}\Big[ F\big(\mathbf{x}(i-1) + a_i\,\mathbf{e}_{u_i}\big) - F\big(\mathbf{x}(i-1) + \mathbf{e}_{u_i}\big)\Big]. \tag{213}$$

Substituting equation 213 into equation 211, we obtain

$$\begin{aligned}
F\big(\mathbf{o}^{(i)}\big) - F\big(\mathbf{o}^{(i-1)}\big) \\
\geq \frac{\Delta_{a,i}}{\Delta_{a,i} + \gamma^2 \Delta_{b,i}}\Big( F\big(\mathbf{x}(i-1) + a_i\,\mathbf{e}_{u_i}\big) - F\big(\mathbf{x}(i-1) + \mathbf{e}_{u_i}\big) - \Delta_{b,i}\Big) - \frac{2\varepsilon}{n}\,\gamma^3\, F(\mathbf{o}).
\end{aligned} \tag{214}$$

By the definition of $a_i$ in Algorithm 4 (and the fact that $1 \in V$), the choice of $a_i$ along the grid ensures that

$$F\big(\mathbf{x}(i-1) + a_i\,\mathbf{e}_{u_i}\big) - F\big(\mathbf{x}(i-1) + \mathbf{e}_{u_i}\big) \;\leq\; 0. \tag{215}$$

Therefore

$$F\big(\mathbf{x}(i-1) + a_i\,\mathbf{e}_{u_i}\big) - F\big(\mathbf{x}(i-1) + \mathbf{e}_{u_i}\big) - \Delta_{b,i} \;\le\; -\Delta_{b,i}. \tag{216}$$

Substituting equation 216 into equation 214 yields

$$F\big(\mathbf{o}^{(i)}\big) - F\big(\mathbf{o}^{(i-1)}\big) \;\ge\; -\,\frac{\Delta_{a,i}\,\Delta_{b,i}}{\Delta_{a,i} + \gamma^2\Delta_{b,i}} - \frac{2\varepsilon}{n}\,\gamma^3\,F(\mathbf{o}). \tag{217}$$

Rearranging equation 217 gives

$$F\big(\mathbf{o}^{(i-1)}\big) - F\big(\mathbf{o}^{(i)}\big) \;\le\; \frac{\Delta_{a,i}\,\Delta_{b,i}}{\Delta_{a,i} + \gamma^2\Delta_{b,i}} + \frac{2\varepsilon}{n}\,\gamma^3\,F(\mathbf{o}), \tag{218}$$

which is exactly equation 190 in the case equation 191.

The remaining case $\mathbf{o}_{u_i}^{(i-1)} > \mathbf{o}_{u_i}^{(i)}$ is analogous: we reverse the roles of the "left" and "right" endpoints on the $u_i$-th coordinate and carry out the same argument, obtaining the same bound equation 190. This completes the proof. $\qquad\square$

Combining the progress of the two trajectories $\mathbf{x}(i)$ and $\mathbf{y}(i)$ with the contraction of the reference path $\mathbf{o}^{(i)}$ yields the following inequality for any tradeoff parameter $r \ge 0$. It will telescope over $i$ to produce the final guarantee.

**Corollary D.5.** *For every $r \ge 0$ and integer $1 \le i \le n$,*

$$\frac{1}{r}\left[F(\mathbf{x}(i)) - F(\mathbf{x}(i-1))\right] \;+\; r\left[F(\mathbf{y}(i)) - F(\mathbf{y}(i-1))\right]$$
$$\ge\; 2\gamma^{3/2}\left(F\big(\mathbf{o}^{(i-1)}\big) - F\big(\mathbf{o}^{(i)}\big) - \frac{2\varepsilon}{n}\,\gamma^3\,F(\mathbf{o})\right). \tag{219}$$

*Proof.* By Lemma D.3, for each $i$ we have

$$F\big(\mathbf{x}(i)\big) - F\big(\mathbf{x}(i-1)\big) \;\ge\; \frac{\Delta_{a,i}^2}{\Delta_{a,i} + \gamma^3\Delta_{b,i}}, \qquad F\big(\mathbf{y}(i)\big) - F\big(\mathbf{y}(i-1)\big) \;\ge\; \frac{\gamma^3\Delta_{b,i}^2}{\Delta_{a,i} + \gamma^3\Delta_{b,i}}. \tag{220}$$

Multiplying the first inequality in equation 220 by $1/r$ and the second by $r$, and adding them, gives

$$\frac{1}{r}\left[F(\mathbf{x}(i)) - F(\mathbf{x}(i-1))\right] + r\left[F(\mathbf{y}(i)) - F(\mathbf{y}(i-1))\right] \ge \frac{(1/r)\,\Delta_{a,i}^2}{\Delta_{a,i} + \gamma^3\Delta_{b,i}} + \frac{r\,\gamma^3\,\Delta_{b,i}^2}{\Delta_{a,i} + \gamma^3\Delta_{b,i}}. \tag{221}$$

The numerator in equation 221 can be rewritten as a completed square plus a mixed term:

$$\frac{(1/r)\,\Delta_{a,i}^2 + r\gamma^3\Delta_{b,i}^2}{\Delta_{a,i} + \gamma^3\Delta_{b,i}} = \frac{\left(\frac{\Delta_{a,i}}{\sqrt{r}} - \Delta_{b,i}\,\gamma^{3/2}\sqrt{r}\right)^2 + 2\gamma^{3/2}\,\Delta_{a,i}\Delta_{b,i}}{\Delta_{a,i} + \gamma^3\Delta_{b,i}}. \tag{222}$$

Substituting equation 222 into equation 221, and using that the square term is nonnegative, we obtain

$$\frac{1}{r}\left[F(\mathbf{x}(i)) - F(\mathbf{x}(i-1))\right] + r\left[F(\mathbf{y}(i)) - F(\mathbf{y}(i-1))\right] \ge \frac{2\gamma^{3/2}\,\Delta_{a,i}\Delta_{b,i}}{\Delta_{a,i} + \gamma^3\Delta_{b,i}}. \tag{223}$$

Thus, up to the factor $\Delta_{a,i}\Delta_{b,i}$, the per step progress is a convex combination–type quantity that depends on $\gamma$.

Next we relate $\Delta_{a,i}\Delta_{b,i}$ to the contraction of the reference path. Lemma D.4 states that

$$F\big(\mathbf{o}^{(i-1)}\big) - F\big(\mathbf{o}^{(i)}\big) \;\le\; \frac{\Delta_{a,i}\,\Delta_{b,i}}{\Delta_{a,i} + \gamma^2\Delta_{b,i}} \;+\; \frac{2\varepsilon}{n}\,\gamma^3\,F(\mathbf{o}). \tag{224}$$

Rearranging equation 224, we get

$$\Delta_{a,i}\Delta_{b,i} \ \geq \ \left(F\big(\mathbf{o}^{(i-1)}\big) - F\big(\mathbf{o}^{(i)}\big) - \frac{2\varepsilon}{n}\,\gamma^3\,F(\mathbf{o})\right)\big(\Delta_{a,i} + \gamma^2\Delta_{b,i}\big). \tag{225}$$

Substituting equation 225 into equation 223 yields

$$\frac{1}{r}\big[F(\mathbf{x}(i)) - F(\mathbf{x}(i-1))\big] + r\big[F(\mathbf{y}(i)) - F(\mathbf{y}(i-1))\big]$$
$$\geq \frac{2\gamma^{3/2}\big(F(\mathbf{o}^{(i-1)}) - F(\mathbf{o}^{(i)}) - \frac{2\varepsilon}{n}\gamma^3 F(\mathbf{o})\big)\big(\Delta_{a,i} + \gamma^2\Delta_{b,i}\big)}{\Delta_{a,i} + \gamma^3\Delta_{b,i}}. \tag{226}$$

Since $\gamma \in (0, 1]$, we have $\gamma^2 \geq \gamma^3$, so

$$\Delta_{a,i} + \gamma^2\Delta_{b,i} \ \geq \ \Delta_{a,i} + \gamma^3\Delta_{b,i}, \tag{227}$$

and hence

$$\frac{\Delta_{a,i} + \gamma^2\Delta_{b,i}}{\Delta_{a,i} + \gamma^3\Delta_{b,i}} \ \geq \ 1. \tag{228}$$

Applying equation 228 to equation 226 gives

$$\frac{1}{r}\big[F(\mathbf{x}(i)) - F(\mathbf{x}(i-1))\big] + r\big[F(\mathbf{y}(i)) - F(\mathbf{y}(i-1))\big]$$
$$\geq 2\gamma^{3/2}\left(F\big(\mathbf{o}^{(i-1)}\big) - F\big(\mathbf{o}^{(i)}\big) - \frac{2\varepsilon}{n}\,\gamma^3\,F(\mathbf{o})\right). \tag{229}$$

This is exactly equation 219, completing the proof. $\qquad\square$

We now conclude the analysis of the $\gamma$–aware Double–Greedy routine. The theorem below is obtained by telescoping the per iteration coupling bound together with the contraction of the lattice coupled reference path. The guarantee *interpolates continuously* in $\gamma$ and reduces to the classical DR bound when $\gamma = 1$.

**Theorem D.6.** *There exists a polynomial time algorithm that, given a nonnegative $\gamma$-weakly DR-submodular function $F : [0, 1]^n \to \mathbb{R}_{\geq 0}$ and a parameter $\varepsilon \in (0, 1)$, outputs $\mathbf{x} \in [0, 1]^n$ such that for every fixed $\mathbf{o} \in [0, 1]^n$,*

$$F(\mathbf{x}) \ \geq \ \max_{r \geq 0} \ \frac{\big(2\gamma^{3/2} - 4\varepsilon\,\gamma^{9/2}\big)\,r\,F(\mathbf{o}) \ + \ F(\mathbf{0}) \ + \ r^2\,F(\mathbf{1})}{r^2 \ + \ 2\gamma^{3/2}r \ + \ 1}. \tag{230}$$

*Proof.* Fix any $r > 0$. Summing the per iteration inequality from Corollary D.5 over $i = 1, \ldots, n$ gives

$$\frac{1}{r}\big[F(\mathbf{x}(n)) - F(\mathbf{x}(0))\big] \ + \ r\big[F(\mathbf{y}(n)) - F(\mathbf{y}(0))\big]$$
$$= \sum_{i=1}^{n}\left(\frac{1}{r}\big[F(\mathbf{x}(i)) - F(\mathbf{x}(i-1))\big] \ + \ r\big[F(\mathbf{y}(i)) - F(\mathbf{y}(i-1))\big]\right)$$
$$\geq \sum_{i=1}^{n}\left(2\gamma^{3/2}\big[F(\mathbf{o}^{(i-1)}) - F(\mathbf{o}^{(i)})\big] \ - \ \frac{2\varepsilon}{n}\cdot 2\gamma^{9/2}\,F(\mathbf{o})\right) \tag{231}$$
$$= 2\gamma^{3/2}\big[F(\mathbf{o}^{(0)}) - F(\mathbf{o}^{(n)})\big] \ - \ 4\varepsilon\,\gamma^{9/2}\,F(\mathbf{o}).$$

Here the first equality is just telescoping the increments of $F(\mathbf{x}^{(i)})$ and $F(\mathbf{y}^{(i)})$, and the inequality uses Corollary D.5 at each iteration $i$.

By construction of the reference path, we have $\mathbf{o}^{(0)} = \mathbf{o}$ and

$$\mathbf{x}(n) = \mathbf{y}(n) = \mathbf{o}^{(n)}. \tag{232}$$

Moreover, Algorithm 4 starts from

$$\mathbf{x}(0) = \mathbf{0}, \qquad \mathbf{y}(0) = \mathbf{1}. \tag{233}$$

Using equation 232–equation 233 in equation 231, we obtain

$$\frac{1}{r}\big[F(\mathbf{x}(n)) - F(\mathbf{0})\big] \ + \ r\big[F(\mathbf{x}(n)) - F(\mathbf{1})\big] \ \geq \ 2\gamma^{3/2}\big[F(\mathbf{o}) - F(\mathbf{x}(n))\big] \ - \ 4\varepsilon\,\gamma^{9/2}\,F(\mathbf{o}). \tag{234}$$

We now collect all terms involving $F(\mathbf{x}(n))$ on the left hand side of equation 234. Rearranging gives

$$F(\mathbf{x}(n))\left(\frac{1}{r} + r + 2\gamma^{3/2}\right) \ \geq \ \big(2\gamma^{3/2} - 4\varepsilon\,\gamma^{9/2}\big)\,F(\mathbf{o}) \ + \ \frac{1}{r}\,F(\mathbf{0}) \ + \ r\,F(\mathbf{1}), \tag{235}$$

where the right hand side collects the contributions of $F(\mathbf{o})$, $F(\mathbf{0})$ and $F(\mathbf{1})$.

Dividing both sides of equation 235 by $\frac{1}{r} + r + 2\gamma^{3/2}$ yields

$$F(\mathbf{x}(n)) \ \geq \ \frac{\big(2\gamma^{3/2} - 4\varepsilon\,\gamma^{9/2}\big)\,F(\mathbf{o}) \ + \ \frac{1}{r}\,F(\mathbf{0}) \ + \ r\,F(\mathbf{1})}{\frac{1}{r} + r + 2\gamma^{3/2}}. \tag{236}$$

Multiplying numerator and denominator of the right hand side of equation 236 by $r$ gives

$$F(\mathbf{x}(n)) \ \geq \ \frac{\big(2\gamma^{3/2} - 4\varepsilon\,\gamma^{9/2}\big)\,r\,F(\mathbf{o}) \ + \ F(\mathbf{0}) \ + \ r^2\,F(\mathbf{1})}{r^2 + 2\gamma^{3/2}r + 1}. \tag{237}$$

Since $\mathbf{x} = \mathbf{x}^{(n)}$ is the output of the algorithm, inequality equation 237 holds for every choice of $r > 0$. Extending to $r = 0$ by continuity of the right hand side in $r$ and taking the maximum over $r \geq 0$ yields equation 230, which completes the proof. $\qquad\square$

# E   Proofs of Lemma 4.1 and Theorem 4.2

In this section first we prove Lemma 4.1, and then we prove Theorem 4.2

**Lemma 4.1.** *Let $F : [0,1]^n \to \mathbb{R}_{\geq 0}$ be nonnegative and $\gamma$-weakly DR-submodular for some $0 < \gamma \leq 1$, and let $P \subseteq [0,1]^n$ be down-closed. There exists a constant size (depending only on $\varepsilon$ and $\gamma$) set of triples $\mathcal{G} \subseteq \mathbb{R}_{\geq 0}^3$ such that $\mathcal{G}$ contains a triple $(g, g_\odot, g_\oplus)$ with*

$$(1 - \varepsilon)\,F(\mathbf{o}) \leq g \leq F(\mathbf{o}), \tag{238a}$$
$$F(\mathbf{z} \odot \mathbf{o}) - \varepsilon\,g \leq g_\odot \leq F(\mathbf{z} \odot \mathbf{o}), \tag{238b}$$
$$F(\mathbf{z} \oplus \mathbf{o}) - \varepsilon\,g \leq g_\oplus \leq F(\mathbf{z} \oplus \mathbf{o}). \tag{238c}$$

*Proof.* Assume we have a constant factor estimate $v$ such that

$$c\,F(\mathbf{o}) \ \leq \ v \ \leq \ F(\mathbf{o}) \tag{239}$$

for some absolute constant $c \in (0, 1]$. We will construct a constant size guess set using $v$.

Define the one dimensional guess set

$$G_o \ := \ \left\{(1 - \varepsilon)^i \cdot \frac{v}{c} \ : \ i = 0, 1, \ldots, \lceil \log_{1-\varepsilon} c \rceil\right\}. \tag{240}$$

The size of $G_o$ is $|G_o| = \mathcal{O}_\varepsilon(1)$, since the exponent range in equation 240 depends only on $\varepsilon$ and $c$. By construction, the values in $G_o$ form a geometric grid that $\varepsilon$-covers the interval $[F(\mathbf{o}), F(\mathbf{o})/c]$, and therefore there exists

$$g \in G_o \quad \text{with} \quad (1 - \varepsilon)\,F(\mathbf{o}) \ \leq \ g \ \leq \ F(\mathbf{o}), \tag{241}$$

which proves equation 22a; see, e.g., (Buchbinder & Feldman, 2024) for this standard argument.

Next we upper bound the ranges in which $F(\mathbf{z} \odot \mathbf{o})$ and $F(\mathbf{z} \oplus \mathbf{o})$ can lie as a function of $F(\mathbf{o})$. Since $\mathbf{z} \odot \mathbf{o} \leq \mathbf{o}$ and $F$ is nonnegative, we always have

$$0 \leq F(\mathbf{z} \odot \mathbf{o}) \leq F(\mathbf{o}). \tag{242}$$

If $F$ is monotone, then equation 242 is immediate; otherwise we only use the trivial nonnegativity upper bound.

For the $\oplus$ operation we can bound, using $\gamma$–weakly DR-submodularity and nonnegativity,

$$
\begin{aligned}
F(\mathbf{z} \oplus \mathbf{o}) &= F\big(\mathbf{o} + (\mathbf{1} - \mathbf{o}) \odot \mathbf{z}\big) \\
&\leq F(\mathbf{o}) + \frac{1}{\gamma}\Big[F\big((\mathbf{1} - \mathbf{o}) \odot \mathbf{z}\big) - F(\mathbf{0})\Big] \quad \text{(by } \gamma\text{–weakly DR along } (\mathbf{1} - \mathbf{o}) \odot \mathbf{z}) \\
&\leq F(\mathbf{o}) + \frac{1}{\gamma} F(\mathbf{o}) \qquad\qquad\qquad \text{(since } F((\mathbf{1} - \mathbf{o}) \odot \mathbf{z}) \leq F(\mathbf{o}) \text{ and } F(\mathbf{0}) \geq 0) \\
&= \left(1 + \tfrac{1}{\gamma}\right) F(\mathbf{o}).
\end{aligned}
\tag{243}
$$

Thus both $F(\mathbf{z} \odot \mathbf{o})$ and $F(\mathbf{z} \oplus \mathbf{o})$ lie in ranges that are linearly bounded in $F(\mathbf{o})$.

For any chosen $g \in G_o$ satisfying equation 241, we now construct $\varepsilon g$-nets for these ranges. For the $\odot$-case, define

$$G_{\odot}(g) := \left\{\varepsilon i \cdot g \ : \ i = 0, 1, \ldots, \left\lceil \frac{1}{\varepsilon(1-\varepsilon)} \right\rceil \right\}. \tag{244}$$

Since equation 242 and equation 241 imply

$$0 \leq F(\mathbf{z} \odot \mathbf{o}) \leq F(\mathbf{o}) \leq \frac{g}{1-\varepsilon}, \tag{245}$$

the grid in equation 244 $\varepsilon g$-covers the interval $[0, F(\mathbf{z} \odot \mathbf{o})]$: for any value $x \in [0, F(\mathbf{z} \odot \mathbf{o})]$, there exists some $g_{\odot} \in G_{\odot}(g)$ with

$$x - \varepsilon g \leq g_{\odot} \leq x. \tag{246}$$

In particular, we can choose $g_{\odot}$ so that equation 246 holds with $x = F(\mathbf{z} \odot \mathbf{o})$, which is exactly equation 22b.

Similarly, for the $\oplus$-case define

$$G_{\oplus}(g) := \left\{\varepsilon i \cdot g \ : \ i = 0, 1, \ldots, \left\lceil \frac{1 + 1/\gamma}{\varepsilon(1-\varepsilon)} \right\rceil \right\}. \tag{247}$$

Using equation 243 and equation 241, we have

$$0 \leq F(\mathbf{z} \oplus \mathbf{o}) \leq \left(1 + \frac{1}{\gamma}\right) F(\mathbf{o}) \leq \frac{1 + 1/\gamma}{1 - \varepsilon} g. \tag{248}$$

Thus the grid in equation 247 $\varepsilon g$-covers the interval $[0, F(\mathbf{z} \oplus \mathbf{o})]$, and there exists $g_{\oplus} \in G_{\oplus}(g)$ such that

$$F(\mathbf{z} \oplus \mathbf{o}) - \varepsilon g \leq g_{\oplus} \leq F(\mathbf{z} \oplus \mathbf{o}), \tag{249}$$

which is equation 22c.

Finally, set

$$\mathcal{G} := \bigcup_{g \in G_o} \{g\} \times G_{\odot}(g) \times G_{\oplus}(g). \tag{250}$$

By equation 241, equation 246, and equation 249, there exists a triple $(g, g_{\odot}, g_{\oplus}) \in \mathcal{G}$ that satisfies all three bounds in equation 22. Moreover, $|\mathcal{G}|$ depends only on $\varepsilon$ and $\gamma$ via the cardinalities of $G_o$, $G_{\odot}(g)$, and $G_{\oplus}(g)$, so $\mathcal{G}$ has constant size. $\qquad\square$

To prove Theorem 4.2, we now recall the closed form of the iterates $\{\mathbf{y}(i)\}_{i=0}^{\delta-1}$ from (Buchbinder & Feldman, 2024), together with the feasibility of the terminal iterate. These formulas will be used to relate $F(\mathbf{y}(\delta^{-1}))$ to the benchmark value $F(\mathbf{o})$.

**Lemma E.1** (Closed form of $\mathbf{y}(i)$ (Buchbinder & Feldman, 2024)). *For every integer $0 \le i \le \delta^{-1}$,*

$$\mathbf{y}(i) = \begin{cases} (\mathbf{1} - \mathbf{z}) \odot \displaystyle\bigoplus_{j=1}^{i}\big(\delta\,\mathbf{x}(j)\big), & i \le i_s, \\[2mm] (\mathbf{1} - \mathbf{z}) \odot \displaystyle\bigoplus_{j=1}^{i}\big(\delta\,\mathbf{x}(j)\big) \;+\; \mathbf{z} \odot \displaystyle\bigoplus_{j=i_s+1}^{i}\big(\delta\,\mathbf{x}(j)\big), & i \ge i_s. \end{cases} \tag{251}$$

*By convention, for any index $a$,*

$$\bigoplus_{j=a}^{a-1}\big(\delta\,\mathbf{x}(j)\big) \;:=\; \mathbf{0}, \tag{252}$$

*so that both expressions in equation 251 remain valid on their boundary indices.*

**Observation 3** (Feasibility (Buchbinder & Feldman, 2024)). The terminal iterate satisfies

$$\mathbf{y}(\delta^{-1}) \in P. \tag{253}$$

We now prove Theorem 4.2 first for the case

$$Q(i) \ne \varnothing \qquad \text{for all } i \in \{1, \dots, \delta^{-1}\}, \tag{254}$$

where two lemmas (Lemmas E.2 and E.3) yield the bound stated in Theorem 4.2 (1), summarized as Corollary E.4. The complementary case in which some $Q(i) = \varnothing$ is handled separately afterwards.

**Observation 4.** If $Q(i) \ne \varnothing$ for some $i \in [\delta^{-1}]$, then

$$F\big(\mathbf{y}(i)\big) - F\big(\mathbf{y}(i-1)\big) \;\ge\; \delta\,\gamma\Big[V(i-1) - F\big(\mathbf{y}(i-1)\big)\Big] \;-\; \frac{\delta^2 L D^2}{2}\,. \tag{255}$$

*Proof.* Consider the line segment

$$\mathbf{u}(s) \;:=\; \mathbf{y}(i-1) \;+\; s\left(\big(\mathbf{1} - \mathbf{y}(i-1) - \mathbf{z}(i-1)\big) \odot \mathbf{x}(i)\right), \qquad s \in [0, \delta]. \tag{256}$$

By construction, $\mathbf{u}(0) = \mathbf{y}(i-1)$ and $\mathbf{u}(\delta) = \mathbf{y}(i)$.

Using the fundamental theorem of calculus along equation 256,

$$F\big(\mathbf{y}(i)\big) - F\big(\mathbf{y}(i-1)\big) \tag{257}$$

$$= \int_0^{\delta} \Big\langle \big(\mathbf{1} - \mathbf{y}(i-1) - \mathbf{z}(i-1)\big) \odot \mathbf{x}(i),\ \nabla F\big(\mathbf{u}(s)\big) \Big\rangle ds \tag{258}$$

$$= \int_0^{\delta} \Big\langle \big(\mathbf{1} - \mathbf{y}(i-1) - \mathbf{z}(i-1)\big) \odot \mathbf{x}(i),\ \nabla F\big(\mathbf{y}(i-1)\big) \Big\rangle ds$$

$$\quad + \int_0^{\delta} \Big\langle \big(\mathbf{1} - \mathbf{y}(i-1) - \mathbf{z}(i-1)\big) \odot \mathbf{x}(i),\ \nabla F\big(\mathbf{u}(s)\big) - \nabla F\big(\mathbf{y}(i-1)\big) \Big\rangle ds$$

$$= \delta\,\langle \mathbf{w}(i), \mathbf{x}(i)\rangle + \int_0^{\delta}\Big\langle \big(\mathbf{1} - \mathbf{y}(i-1) - \mathbf{z}(i-1)\big) \odot \mathbf{x}(i),\ \nabla F\big(\mathbf{u}(s)\big) - \nabla F\big(\mathbf{y}(i-1)\big) \Big\rangle ds.$$

Here we used that

$$\mathbf{w}(i) := \big(\mathbf{1} - \mathbf{y}(i-1) - \mathbf{z}(i-1)\big) \odot \nabla F\big(\mathbf{y}(i-1)\big),$$

so $\langle \mathbf{w}(i), \mathbf{x}(i)\rangle = \langle(\mathbf{1} - \mathbf{y}(i-1) - \mathbf{z}(i-1)) \odot \mathbf{x}(i), \nabla F(\mathbf{y}(i-1))\rangle$.

Applying Cauchy–Schwarz to the second integral gives

$$F\big(\mathbf{y}(i)\big) - F\big(\mathbf{y}(i-1)\big) \ge \delta\,\langle \mathbf{w}(i), \mathbf{x}(i)\rangle$$

$$- \int_0^{\delta}\Big\|\big(\mathbf{1} - \mathbf{y}(i-1) - \mathbf{z}(i-1)\big) \odot \mathbf{x}(i)\Big\|_2 \Big\|\nabla F\big(\mathbf{u}(s)\big) - \nabla F\big(\mathbf{y}(i-1)\big)\Big\|_2 ds. \tag{259}$$

This step uses $\langle a, b \rangle \geq -\|a\|_2 \|b\|_2$.

Since $F$ is $L$-smooth, its gradient is $L$-Lipschitz, so

$$\left\| \nabla F(\mathbf{u}(s)) - \nabla F(\mathbf{y}(i-1)) \right\|_2 \ \leq \ L \left\| \mathbf{u}(s) - \mathbf{y}(i-1) \right\|_2 \ = \ L\,s \left\| \big(\mathbf{1} - \mathbf{y}(i-1) - \mathbf{z}(i-1)\big) \odot \mathbf{x}(i) \right\|_2. \quad (260)$$

Substituting equation 260 into equation 259 yields

$$\begin{aligned}
F\big(\mathbf{y}(i)\big) - F\big(\mathbf{y}(i-1)\big) &\geq \delta \left\langle \mathbf{w}(i), \mathbf{x}(i) \right\rangle - \int_0^\delta s\,L \left\| \big(\mathbf{1} - \mathbf{y}(i-1) - \mathbf{z}(i-1)\big) \odot \mathbf{x}(i) \right\|_2^2 ds \\
&= \delta \left\langle \mathbf{w}(i), \mathbf{x}(i) \right\rangle - \frac{L}{2} \delta^2 \left\| \big(\mathbf{1} - \mathbf{y}(i-1) - \mathbf{z}(i-1)\big) \odot \mathbf{x}(i) \right\|_2^2,
\end{aligned} \quad (261)$$

where we used $\int_0^\delta s\,ds = \delta^2/2$.

Let

$$\mathbf{d}(i) := \big(\mathbf{1} - \mathbf{y}(i-1) - \mathbf{z}(i-1)\big) \odot \mathbf{x}(i).$$

By feasibility and the coordinatewise bounds $0 \leq \mathbf{y}(i-1), \mathbf{z}(i-1), \mathbf{x}(i) \leq \mathbf{1}$, each entry of $\mathbf{d}(i)$ lies in $[0, 1]$, so

$$\|\mathbf{d}(i)\|_2 \ \leq \ D,$$

Where $D$ is diameter. Using this in equation 261 gives

$$F\big(\mathbf{y}(i)\big) - F\big(\mathbf{y}(i-1)\big) \ \geq \ \delta \left\langle \mathbf{w}(i), \mathbf{x}(i) \right\rangle \ - \ \frac{\delta^2 L D^2}{2}. \quad (262)$$

Finally, since $Q(i) \neq \varnothing$, the choice of $\mathbf{x}(i) \in Q(i)$ guarantees

$$\left\langle \mathbf{w}(i), \mathbf{x}(i) \right\rangle \ \geq \ \gamma \left[ V(i-1) - F\big(\mathbf{y}(i-1)\big) \right], \quad (263)$$

by the definition of $Q(i)$ and the weakly–DR structure. Substituting equation 263 into equation 262 yields

$$F\big(\mathbf{y}(i)\big) - F\big(\mathbf{y}(i-1)\big) \ \geq \ \delta\,\gamma \left[ V(i-1) - F\big(\mathbf{y}(i-1)\big) \right] \ - \ \frac{\delta^2 L D^2}{2},$$

which is exactly equation 255. $\qquad\square$

The bound equation 255 gives a recursive lower bound on $F(\mathbf{y}(i))$. In what follows (up to Corollary E.4), we unroll this recursion and derive a closed form.

**Lemma E.2.** *Assume $Q(i) \neq \varnothing$ for every $i \in [i_s]$. Fix $0 < \delta \leq 1$ and $0 < \gamma \leq 1$, and set*

$$\alpha := \delta\gamma, \qquad \beta := \frac{\gamma^2 \delta}{1 - \delta + \gamma^2 \delta}.$$

*For $i \geq 0$, define the geometric shorthands*

$$\Delta_i \ := \ 1 - (1-\alpha)^i, \qquad \Theta_i \ := \ (1-\beta)^i - (1-\alpha)^i.$$

*Then, for every integer $0 \leq i \leq i_s$,*

$$F\big(\mathbf{y}(i)\big) \ \geq \ \left[ \frac{(1-2\varepsilon)\,\Delta_i}{\gamma} + \frac{\delta(1-\gamma)}{\beta - \alpha}\,\Theta_i \right] g \ - \ \frac{\Delta_i}{\gamma}\,g_\odot \ - \ \left[ \frac{\Delta_i}{\gamma} + \frac{\delta}{\beta - \alpha}\,\Theta_i \right] g_\oplus \ - \ i\,\delta^2 D^2 L. \quad (264)$$

*Proof.* For $i = 0$, we have $\Delta_0 = 0$ and $\Theta_0 = 0$, so the right hand side of equation 264 equals 0. Since $F(\mathbf{y}(0)) \geq 0$ by nonnegativity of $F$, the claim holds for $i = 0$. We therefore fix an integer $1 \leq i \leq i_s$ for the rest of the proof.

By Observation 4 (and the assumption $Q(i-1) \neq \varnothing$ for $i \leq i_s$),

$$F\big(\mathbf{y}(i)\big) - F\big(\mathbf{y}(i-1)\big) \ \geq \ \delta\,\gamma\Big(V(i-1) - F\big(\mathbf{y}(i-1)\big)\Big) \ - \ \frac{\delta^2 D^2 L}{2}. \tag{265}$$

Rearranging equation 265 gives

$$F\big(\mathbf{y}(i)\big) \ \geq \ (1-\alpha)\,F\big(\mathbf{y}(i-1)\big) \ + \ \delta\gamma\,V(i-1) \ - \ \frac{\delta^2 D^2 L}{2}, \tag{266}$$

where we used $\alpha = \delta\gamma$.

The quantity $V(i-1)$ has the explicit form as given in equation 23

$$V(i-1) = \left[(1-\beta)^{i-1} + \frac{1-(1-\beta)^{i-1}-2\varepsilon}{\gamma}\right] g \ - \ \frac{1}{\gamma}\,g_\odot \ - \ \frac{1-(1-\beta)^{i-1}}{\gamma}\,g_\oplus. \tag{267}$$

We now multiply equation 267 by $\delta\gamma$ and substitute into equation 266. First, for the $g$-term,

$$\delta\gamma\left[(1-\beta)^{i-1} + \frac{1-(1-\beta)^{i-1}-2\varepsilon}{\gamma}\right] = \delta\gamma(1-\beta)^{i-1} + \delta\Big(1-(1-\beta)^{i-1}-2\varepsilon\Big)$$

$$= \delta\Big[\gamma(1-\beta)^{i-1} + 1 - (1-\beta)^{i-1} - 2\varepsilon\Big]$$

$$= \delta\Big[1 - (1-\gamma)(1-\beta)^{i-1} - 2\varepsilon\Big]. \tag{268}$$

For the $g_\odot$-term, we obtain

$$\delta\gamma \cdot \left(-\frac{1}{\gamma}\,g_\odot\right) = -\delta\,g_\odot. \tag{269}$$

For the $g_\oplus$-term,

$$\delta\gamma \cdot \left(-\frac{1-(1-\beta)^{i-1}}{\gamma}\,g_\oplus\right) = -\delta\Big(1-(1-\beta)^{i-1}\Big)g_\oplus. \tag{270}$$

Substituting equation 268–equation 270 into equation 266 yields

$$\begin{aligned} F\big(\mathbf{y}(i)\big) \ \geq \ (1-\alpha)\,F\big(\mathbf{y}(i-1)\big) + \ &\delta\Big(1-(1-\gamma)(1-\beta)^{i-1}-2\varepsilon\Big)g \\ - \ \delta\,g_\odot \ - \ &\delta\Big(1-(1-\beta)^{i-1}\Big)g_\oplus \ - \ \frac{\delta^2 D^2 L}{2}. \end{aligned} \tag{271}$$

We now unroll the recursion equation 271 from $k=1$ up to $k=i$. Writing $F_k := F(\mathbf{y}(k))$ for brevity, equation 271 becomes

$$F_k \ \geq \ (1-\alpha)\,F_{k-1} + \delta\,A_{k-1}\,g - \delta\,g_\odot - \delta\,B_{k-1}\,g_\oplus - \frac{\delta^2 D^2 L}{2}, \tag{272}$$

where

$$A_{k-1} := 1 - (1-\gamma)(1-\beta)^{k-1} - 2\varepsilon, \qquad B_{k-1} := 1 - (1-\beta)^{k-1}. \tag{273}$$

Iterating equation 272 from $k=1$ to $k=i$ gives

$$\begin{aligned} F_i \geq (1-\alpha)^i F_0 + \delta\sum_{k=1}^{i}(1-\alpha)^{i-k}A_{k-1}\,g \\ - \delta\sum_{k=1}^{i}(1-\alpha)^{i-k}g_\odot - \delta\sum_{k=1}^{i}(1-\alpha)^{i-k}B_{k-1}\,g_\oplus - \frac{\delta^2 D^2 L}{2}\sum_{k=1}^{i}(1-\alpha)^{i-k}. \end{aligned} \tag{274}$$

Since $F_0 = F(\mathbf{y}(0)) \geq 0$ by nonnegativity of $F$, the first term $(1-\alpha)^i F_0$ is nonnegative and can be dropped for a lower bound. We next compute the geometric sums appearing in equation 274.

First,

$$\sum_{k=1}^{i}(1-\alpha)^{i-k} = \sum_{t=0}^{i-1}(1-\alpha)^t = \frac{1-(1-\alpha)^i}{\alpha} = \frac{\Delta_i}{\alpha}. \tag{275}$$

Next, for the mixed sum, using the change of variable $m = k - 1$,

$$\sum_{k=1}^{i}(1-\alpha)^{i-k}(1-\beta)^{k-1} = \sum_{m=0}^{i-1}(1-\alpha)^{i-1-m}(1-\beta)^m. \tag{276}$$

This is a geometric series in $m$. Factoring out $(1-\alpha)^{i-1}$, we obtain

$$\sum_{m=0}^{i-1}(1-\alpha)^{i-1-m}(1-\beta)^m = (1-\alpha)^{i-1}\sum_{m=0}^{i-1}\left(\frac{1-\beta}{1-\alpha}\right)^m$$

$$= (1-\alpha)^{i-1} \cdot \frac{1-\left(\frac{1-\beta}{1-\alpha}\right)^i}{1-\frac{1-\beta}{1-\alpha}}$$

$$= (1-\alpha)^{i-1} \cdot \frac{1-\frac{(1-\beta)^i}{(1-\alpha)^i}}{\frac{\beta-\alpha}{1-\alpha}}$$

$$= \frac{(1-\alpha)^i - (1-\beta)^i}{\beta-\alpha}. \tag{277}$$

Recalling $\Theta_i = (1-\beta)^i - (1-\alpha)^i$, we can also write

$$\sum_{k=1}^{i}(1-\alpha)^{i-k}(1-\beta)^{k-1} = \frac{(1-\alpha)^i - (1-\beta)^i}{\beta-\alpha} = -\frac{\Theta_i}{\beta-\alpha}. \tag{278}$$

We now substitute into each coefficient in equation 274.

**Coefficient of $g$.** Using equation 273, we split

$$\sum_{k=1}^{i}(1-\alpha)^{i-k}A_{k-1} = \sum_{k=1}^{i}(1-\alpha)^{i-k}\left(1-(1-\gamma)(1-\beta)^{k-1}-2\varepsilon\right)$$

$$= (1-2\varepsilon)\sum_{k=1}^{i}(1-\alpha)^{i-k} - (1-\gamma)\sum_{k=1}^{i}(1-\alpha)^{i-k}(1-\beta)^{k-1}. \tag{279}$$

Using equation 275 and equation 278, we obtain

$$\sum_{k=1}^{i}(1-\alpha)^{i-k}A_{k-1} = (1-2\varepsilon)\cdot\frac{\Delta_i}{\alpha} - (1-\gamma)\cdot\left(-\frac{\Theta_i}{\beta-\alpha}\right)$$

$$= \frac{(1-2\varepsilon)\Delta_i}{\alpha} + \frac{(1-\gamma)\Theta_i}{\beta-\alpha}. \tag{280}$$

Multiplying by $\delta$ and using $\alpha = \delta\gamma$ (so $\delta/\alpha = 1/\gamma$), we get

$$\delta\sum_{k=1}^{i}(1-\alpha)^{i-k}A_{k-1} = \frac{(1-2\varepsilon)\Delta_i}{\gamma} + \frac{\delta(1-\gamma)\Theta_i}{\beta-\alpha}. \tag{281}$$

**Coefficient of $g_\odot$.** By equation 274 and equation 275,

$$-\delta\sum_{k=1}^{i}(1-\alpha)^{i-k}g_\odot = -\delta\cdot\frac{\Delta_i}{\alpha}g_\odot = -\frac{\Delta_i}{\gamma}g_\odot, \tag{282}$$

again using $\alpha = \delta\gamma$.

**Coefficient of $g_\oplus$.** Using equation 273,

$$\sum_{k=1}^{i}(1-\alpha)^{i-k}B_{k-1} = \sum_{k=1}^{i}(1-\alpha)^{i-k}\Big(1-(1-\beta)^{k-1}\Big)$$

$$= \sum_{k=1}^{i}(1-\alpha)^{i-k} - \sum_{k=1}^{i}(1-\alpha)^{i-k}(1-\beta)^{k-1}. \tag{283}$$

Substituting equation 275 and equation 278 into equation 283,

$$\sum_{k=1}^{i}(1-\alpha)^{i-k}B_{k-1} = \frac{\Delta_i}{\alpha} - \Big(-\frac{\Theta_i}{\beta-\alpha}\Big)$$

$$= \frac{\Delta_i}{\alpha} + \frac{\Theta_i}{\beta-\alpha}. \tag{284}$$

Thus

$$-\delta\sum_{k=1}^{i}(1-\alpha)^{i-k}B_{k-1}\,g_\oplus = -\delta\left(\frac{\Delta_i}{\alpha}+\frac{\Theta_i}{\beta-\alpha}\right)g_\oplus = -\frac{\Delta_i}{\gamma}\,g_\oplus - \frac{\delta}{\beta-\alpha}\,\Theta_i\,g_\oplus, \tag{285}$$

where we again used $\delta/\alpha = 1/\gamma$.

**Smoothness penalty.** The last term in equation 274 is

$$-\frac{\delta^2 D^2 L}{2}\sum_{k=1}^{i}(1-\alpha)^{i-k}. \tag{286}$$

Using that $(1-\alpha)^{i-k}\leq 1$ for all $k$, we have

$$\sum_{k=1}^{i}(1-\alpha)^{i-k} \leq i, \tag{287}$$

and hence

$$-\frac{\delta^2 D^2 L}{2}\sum_{k=1}^{i}(1-\alpha)^{i-k} \geq -\frac{\delta^2 D^2 L}{2}\,i \geq -i\,\delta^2 D^2 L, \tag{288}$$

where the last inequality simply relaxes the factor $1/2$ to obtain a slightly weaker but simpler bound.

Putting together equation 274 with the bounds equation 281, equation 282, equation 285, and equation 288, and recalling $\mathbf{y}(i)$ corresponds to $F_i$, we obtain

$$F\big(\mathbf{y}(i)\big) \geq \left[\frac{(1-2\varepsilon)\,\Delta_i}{\gamma} + \frac{\delta(1-\gamma)}{\beta-\alpha}\,\Theta_i\right]g - \frac{\Delta_i}{\gamma}\,g_\odot - \left[\frac{\Delta_i}{\gamma} + \frac{\delta}{\beta-\alpha}\,\Theta_i\right]g_\oplus - i\,\delta^2 D^2 L,$$

which is exactly equation 264. □

**Lemma E.3.** *Assume $0 < \delta \leq 1$ and $0 < \gamma \leq 1$, and set*

$$\alpha := \delta\gamma, \qquad \beta := \frac{\gamma^2\delta}{1-\delta+\gamma^2\delta}.$$

*Let $i_s < i \leq \delta^{-1}$ and suppose $Q(i) \neq \varnothing$ for every integer $i_s < i \leq \delta^{-1}$. Define the constants*

$$A := \frac{(1-\beta)^{-i_s}}{\gamma} - \Big(1+\frac{3}{\gamma}\Big)\varepsilon + 1 - \frac{1}{\gamma}, \qquad C_\gamma := \frac{(1-\beta)^{-i_s}-1}{\gamma}. \tag{289}$$

*For every integer $i_s \leq i \leq \delta^{-1}$, with the shorthands*

$$S_1(i) \ := \ \sum_{k=i_s+1}^{i} (1-\alpha)^{i-k}(1-\beta)^{k}, \qquad S_2(i) \ := \ \sum_{k=i_s+1}^{i} (1-\alpha)^{i-k}(1-\beta)^{k}\Big(C_\gamma - \beta\,(k-i_s)\Big), \quad (290)$$

*the following bound holds:*

$$F\big(\mathbf{y}(i)\big) \ \geq \ (1-\alpha)^{i-i_s} F\big(\mathbf{y}(i_s)\big) \ + \ \alpha\,A\,S_1(i)\,g \ - \ \alpha\,S_2(i)\,g_\oplus \ - \ (i-i_s)\,\delta^2 D^2 L. \tag{291}$$

*Moreover, letting $n := i - i_s$ and $q := \frac{1-\beta}{1-\alpha}$, we have the closed forms*

$$S_1(i) = \ (1-\alpha)^{n}(1-\beta)^{i_s} \cdot \frac{q(1-q^{n})}{1-q} \ = \ \frac{(1-\alpha)^{n}(1-\beta)^{i_s+1} - (1-\beta)^{i+1}}{\beta-\alpha},$$

$$S_2(i) = \ C_\gamma\,S_1(i) \ - \ \beta\,(1-\alpha)^{n}(1-\beta)^{i_s} \cdot \frac{q\big(1-(n+1)q^{n}+nq^{n+1}\big)}{(1-q)^2}. \tag{292}$$

*Proof.* Set $\alpha := \delta\gamma$ and fix an index $i$ with $i_s < i \leq \delta^{-1}$. From Observation 4, for every $k \in \{i_s+1,\dots,i\}$ we have the one–step recurrence

$$F\big(\mathbf{y}(k)\big) \ \geq \ (1-\alpha)\,F\big(\mathbf{y}(k-1)\big) \ + \ \alpha\,V(k-1) \ - \ \frac{\delta^2 D^2 L}{2}. \tag{293}$$

Here we used $\alpha = \delta\gamma$ to rewrite the term $\delta\gamma\,V(k-1)$ as $\alpha V(k-1)$.

In the post–switch phase ($k > i_s$), the surrogate takes the explicit form given in equation 24

$$V(k-1) \ = \ (1-\beta)^{k}\Big[A\,g \ - \ \big(C_\gamma - \beta\,(k-i_s)\big)\,g_\oplus\Big], \tag{294}$$

where $A$ and $C_\gamma$ are as in equation 289.

Applying equation 293 iteratively from $k = i_s + 1$ up to $k = i$ gives (by a standard induction on $k$)

$$F\big(\mathbf{y}(i)\big) \geq \ (1-\alpha)^{i-i_s} F\big(\mathbf{y}(i_s)\big)$$
$$+ \ \alpha \sum_{k=i_s+1}^{i} (1-\alpha)^{i-k} V(k-1) \ - \ \frac{\delta^2 D^2 L}{2} \sum_{k=i_s+1}^{i} (1-\alpha)^{i-k}. \tag{295}$$

The factor $(1-\alpha)^{i-i_s}$ comes from repeatedly multiplying by $(1-\alpha)$ in the homogeneous part of the recurrence.

Substituting equation 294 into equation 295 yields

$$F\big(\mathbf{y}(i)\big) \geq (1-\alpha)^{i-i_s} F\big(\mathbf{y}(i_s)\big) + \ \alpha \sum_{k=i_s+1}^{i} (1-\alpha)^{i-k}(1-\beta)^{k}\,A\,g$$

$$- \ \alpha \sum_{k=i_s+1}^{i} (1-\alpha)^{i-k}(1-\beta)^{k}\Big(C_\gamma - \beta\,(k-i_s)\Big)\,g_\oplus \tag{296}$$

$$- \ \frac{\delta^2 D^2 L}{2} \sum_{k=i_s+1}^{i} (1-\alpha)^{i-k}.$$

The first sum collects all $g$-terms, the second all $g_\oplus$-terms, and the last sum is the accumulated smoothness penalty.

We now define the sums

$$S_1(i) \ := \ \sum_{k=i_s+1}^{i} (1-\alpha)^{i-k}(1-\beta)^{k}, \quad S_2(i) \ := \ \sum_{k=i_s+1}^{i} (1-\alpha)^{i-k}(1-\beta)^{k}\Big(C_\gamma - \beta\,(k-i_s)\Big). \tag{297}$$

With equation 297, inequality equation 296 becomes

$$F\big(\mathbf{y}(i)\big) \;\geq\; (1-\alpha)^{\,i-i_s}\,F\big(\mathbf{y}(i_s)\big) \;+\; \alpha\,A\,S_1(i)\,g \;-\; \alpha\,S_2(i)\,g_\oplus \;-\; \frac{\delta^2 D^2 L}{2}\sum_{k=i_s+1}^{i}(1-\alpha)^{\,i-k}. \tag{298}$$

Thus the only remaining tasks are to bound the smoothness term and compute closed forms for $S_1(i)$ and $S_2(i)$.

To obtain closed forms, let $n := i - i_s$ and $q := \frac{1-\beta}{1-\alpha}$, and set $t := k - i_s$ so that $t = 1,\dots,n$. Then

$$S_1(i) = \sum_{k=i_s+1}^{i}(1-\alpha)^{\,i-k}(1-\beta)^{\,k} = (1-\alpha)^{\,n}(1-\beta)^{\,i_s}\sum_{t=1}^{n}q^{\,t}, \tag{299}$$

because $(1-\alpha)^{i-k} = (1-\alpha)^{n-t}$ and $(1-\beta)^{k} = (1-\beta)^{i_s+t}$, so each term is

$$(1-\alpha)^{n-t}(1-\beta)^{i_s+t} = (1-\alpha)^{n}(1-\beta)^{i_s}\Big(\frac{1-\beta}{1-\alpha}\Big)^{t} = (1-\alpha)^{n}(1-\beta)^{i_s}q^{t}.$$

Using the geometric sum identity

$$\sum_{t=1}^{n}q^{\,t} = \frac{q(1-q^{\,n})}{1-q},$$

we obtain

$$S_1(i) \;=\; (1-\alpha)^{\,n}(1-\beta)^{\,i_s}\cdot\frac{q(1-q^{\,n})}{1-q}. \tag{300}$$

Writing $1 - q = \frac{\beta-\alpha}{1-\alpha}$ and $q^n = \big(\frac{1-\beta}{1-\alpha}\big)^{n}$, a simple algebraic rearrangement yields the equivalent form

$$S_1(i) \;=\; \frac{(1-\alpha)^{\,n}(1-\beta)^{\,i_s+1} - (1-\beta)^{\,i+1}}{\beta - \alpha}, \tag{301}$$

which is the first line of equation 292.

For $S_2(i)$, define

$$T(i) \;:=\; \sum_{k=i_s+1}^{i}(1-\alpha)^{\,i-k}(1-\beta)^{\,k}\,(k-i_s) \;=\; (1-\alpha)^{\,n}(1-\beta)^{\,i_s}\sum_{t=1}^{n}t\,q^{\,t}.$$

The standard identity

$$\sum_{t=1}^{n}t\,q^{\,t} \;=\; \frac{q\big(1 - (n+1)q^{\,n} + nq^{\,n+1}\big)}{(1-q)^2}$$

then gives

$$T(i) \;=\; (1-\alpha)^{\,n}(1-\beta)^{\,i_s}\cdot\frac{q\big(1 - (n+1)q^{\,n} + nq^{\,n+1}\big)}{(1-q)^2}. \tag{302}$$

By definition of $S_2(i)$,

$$S_2(i) = C_\gamma\,S_1(i) - \beta\,T(i),$$

which together with equation 300 and equation 302 yields the second line of equation 292.

Finally, we bound the smoothness penalty in equation 298. Since $0 \leq (1-\alpha)^{\,i-k} \leq 1$,

$$\sum_{k=i_s+1}^{i}(1-\alpha)^{\,i-k} = \sum_{t=0}^{n-1}(1-\alpha)^{t} \;\leq\; n. \tag{303}$$

Therefore

$$-\frac{\delta^2 D^2 L}{2}\sum_{k=i_s+1}^{i}(1-\alpha)^{\,i-k} \;\geq\; -\frac{\delta^2 D^2 L}{2}\,n \;\geq\; -(i-i_s)\,\delta^2 D^2 L, \tag{304}$$

where we relaxed the factor $\frac{1}{2}$ to get a slightly simpler bound.

Substituting equation 304 into equation 298 gives exactly equation 291, which completes the proof. □

Combining Lemmas E.2 and E.3, we obtain the following corollary, which finishes the proof of Theorem 4.2 in the regime where $Q(i)$ is non-empty for all $i \in [\delta^{-1}]$.

**Corollary E.4.** *Assume $Q(i) \neq \varnothing$ for every $i \in [\delta^{-1}]$. Fix $0 < \delta \leq 1$, $0 < \gamma \leq 1$, and set*

$$\alpha := \delta\gamma, \qquad \beta := \frac{\gamma^2\delta}{1 - \delta + \gamma^2\delta}. \tag{305}$$

*Then, with $i_s \in \{0, 1, \ldots, \delta^{-1}\}$ and $\mathbf{y}(\cdot)$ as in Lemma E.1, we have*

$$
\begin{aligned}
F\big(\mathbf{y}(\delta^{-1})\big) &\geq (1-\alpha)^{\delta^{-1}-i_s} F\big(\mathbf{y}(i_s)\big) + \alpha\,A\,S_1(\delta^{-1})\,g - \alpha\,S_2(\delta^{-1})\,g_\oplus - (\delta^{-1} - i_s)\,\delta^2 D^2 L \\
&\geq (1-\alpha)^{\delta^{-1}-i_s}\Bigg\{ \Big[\tfrac{1-(1-\alpha)^{i_s}}{\gamma}(1-2\varepsilon) + \tfrac{\delta(1-\gamma)}{\beta-\alpha}\big((1-\beta)^{i_s} - (1-\alpha)^{i_s}\big)\Big]g \\
&\qquad - \tfrac{1-(1-\alpha)^{i_s}}{\gamma}g_\odot - \Big[\tfrac{1-(1-\alpha)^{i_s}}{\gamma} + \tfrac{\delta}{\beta-\alpha}\big((1-\beta)^{i_s} - (1-\alpha)^{i_s}\big)\Big]g_\oplus - i_s\,\delta^2 D^2 L \Bigg\} \\
&\qquad + \alpha\,A\,S_1(\delta^{-1})\,g - \alpha\,S_2(\delta^{-1})\,g_\oplus - (\delta^{-1} - i_s)\,\delta^2 D^2 L,
\end{aligned}
\tag{306}
$$

*where*

$$A := \frac{(1-\beta)^{-i_s}}{\gamma} - \Big(1 + \frac{3}{\gamma}\Big)\varepsilon + 1 - \frac{1}{\gamma}, \qquad C_\gamma := \frac{(1-\beta)^{-i_s} - 1}{\gamma}, \tag{307}$$

*and the sums from Lemma E.3 admit the closed forms*

$$S_1(\delta^{-1}) = \frac{(1-\alpha)^{\delta^{-1}-i_s}(1-\beta)^{i_s+1} - (1-\beta)^{\delta^{-1}+1}}{\beta - \alpha},$$

$$
\begin{aligned}
S_2(\delta^{-1}) = \; & C_\gamma\,S_1(\delta^{-1}) - \beta\,(1-\alpha)^{\delta^{-1}-i_s}(1-\beta)^{i_s} \\
& \frac{\tfrac{1-\beta}{1-\alpha}\Big(1 - (\delta^{-1} - i_s + 1)\big(\tfrac{1-\beta}{1-\alpha}\big)^{\delta^{-1}-i_s} + (\delta^{-1} - i_s)\big(\tfrac{1-\beta}{1-\alpha}\big)^{\delta^{-1}-i_s+1}\Big)}{\big(1 - \tfrac{1-\beta}{1-\alpha}\big)^2}.
\end{aligned}
\tag{308}
$$

*Proof.* We use the shorthand parameters $\alpha$ and $\beta$ from equation 305. By Lemma E.2, evaluated at $i = i_s$, we have

$$
\begin{aligned}
F\big(\mathbf{y}(i_s)\big) &\geq \Big[\frac{1-(1-\alpha)^{i_s}}{\gamma}(1-2\varepsilon) + \frac{\delta(1-\gamma)}{\beta-\alpha}\big((1-\beta)^{i_s} - (1-\alpha)^{i_s}\big)\Big]g \\
&\quad - \frac{1-(1-\alpha)^{i_s}}{\gamma}g_\odot - \Big[\frac{1-(1-\alpha)^{i_s}}{\gamma} + \frac{\delta}{\beta-\alpha}\big((1-\beta)^{i_s} - (1-\alpha)^{i_s}\big)\Big]g_\oplus - i_s\,\delta^2 D^2 L.
\end{aligned}
\tag{309}
$$

This is just Lemma E.2 with $i = i_s$ and the definitions $\Delta_{i_s} = 1 - (1-\alpha)^{i_s}$ and $\Theta_{i_s} = (1-\beta)^{i_s} - (1-\alpha)^{i_s}$.

Next, by Lemma E.3, evaluated at $i = \delta^{-1}$, we obtain

$$F\big(\mathbf{y}(\delta^{-1})\big) \geq (1-\alpha)^{\delta^{-1}-i_s} F\big(\mathbf{y}(i_s)\big) + \alpha\,A\,S_1(\delta^{-1})\,g - \alpha\,S_2(\delta^{-1})\,g_\oplus - (\delta^{-1} - i_s)\,\delta^2 D^2 L, \tag{310}$$

where $A$ and $C_\gamma$ are as in equation 307 and $S_1(\delta^{-1})$, $S_2(\delta^{-1})$ are given in equation 308.

Substituting the lower bound equation 309 for $F(\mathbf{y}(i_s))$ into equation 310, then collecting the coefficients of $g$, $g_\odot$, and $g_\oplus$, and combining the smoothness penalties $i_s\delta^2 D^2 L$ and $(\delta^{-1} - i_s)\delta^2 D^2 L$, yields exactly the discrete inequality equation 306. This is a straightforward algebraic rearrangement.

To pass from the discrete time bound equation 306 to the continuous time guarantee, define the switch time

$$t_s := \delta\, i_s \in [0,1],$$

and let $\delta \to 0^+$ while keeping $t_s$ fixed. Using

$$(1-\alpha)^{\delta^{-1}-i_s} = (1-\delta\gamma)^{(1-t_s)/\delta} \;\longrightarrow\; e^{-\gamma(1-t_s)}, \qquad (1-\alpha)^{\,i_s} = (1-\delta\gamma)^{t_s/\delta} \;\longrightarrow\; e^{-\gamma t_s},$$

and the closed forms equation 308, each discrete sum converges to the corresponding time integral in the continuous analysis. Concretely, $S_1(\delta^{-1})$ and $S_2(\delta^{-1})$ can be viewed as Riemann sums in the step size $\delta$; applying standard first order expansions of $(1-\alpha)$ and $(1-\beta)$ (or equivalently, l'Hôpital's rule to the associated limits) yields the continuous coefficients $A_\gamma(t_s)$, $B_\gamma(t_s)$, and $C_\gamma(t_s)$:

$$F(\mathbf{y}^*) \;\geq\; A_\gamma(t_s)\, g \;+\; B_\gamma(t_s)\, g_\odot \;+\; C_\gamma(t_s)\, g_\oplus \;-\; O(\varepsilon)\,(g + g_\odot + g_\oplus) \;-\; \delta\, LD^2, \tag{311}$$

with

$$A_\gamma(t_s) := -\frac{e^{\gamma t_s - \gamma}}{1-\gamma} + \frac{e^{-\gamma^2}}{\gamma(1-\gamma)}\left(e^{\gamma^2 t_s} - (1-\gamma)\right), \tag{312}$$

$$B_\gamma(t_s) := \frac{e^{-\gamma} - e^{\gamma t_s - \gamma}}{\gamma}, \tag{313}$$

$$\begin{aligned}
C_\gamma(t_s) :=\; &\frac{e^{\gamma^2 t_s} - 1}{\gamma(1-\gamma)}\left(e^{-\gamma(1-t_s)-\gamma^2 t_s} - e^{-\gamma^2}\right) + \frac{e^{-\gamma(1-t_s)}}{\gamma}\left[-\left(1 - e^{-\gamma t_s}\right) + \frac{e^{-\gamma^2 t_s} - e^{-\gamma t_s}}{1-\gamma}\right] \\
&+ e^{-\gamma(1-t_s)-\gamma^2 t_s}\left[\frac{\gamma^2}{1-\gamma}(1-t_s)\, e^{\gamma(1-\gamma)(1-t_s)} + \frac{\gamma}{(1-\gamma)^2}\left(1 - e^{\gamma(1-\gamma)(1-t_s)}\right)\right].
\end{aligned} \tag{314}$$

Finally, identifying $(g, g_\odot, g_\oplus)$ with $\bigl(F(\mathbf{o}), F(\mathbf{z}\odot\mathbf{o}), F(\mathbf{z}\oplus\mathbf{o})\bigr)$ shows that equation 311 is exactly the continuous–time guarantee used in Theorem 4.2 (1). $\qquad\square$

At this point, we handle the case where $Q(i) = \varnothing$ for some $i \in [\delta^{-1}]$. Note that $Q(i) = \varnothing$ implies, in particular, $\mathbf{o} \notin Q(i)$. Accordingly, define

$$i_o \;:=\; \min\left\{\, i \in [\delta^{-1}] \;:\; \mathbf{o} \notin Q(i) \,\right\}.$$

**Observation 5.** For every $i \in [i_o - 1]$,

$$F\bigl(\mathbf{x}(i)\bigr) \;\geq\; \frac{\gamma^2\, F\bigl(\mathbf{x}(i)\vee\mathbf{o}\bigr) + F\bigl(\mathbf{x}(i)\wedge\mathbf{o}\bigr)}{1+\gamma^2} \;-\; \varepsilon\, F(\mathbf{o}) \;-\; \delta\, LD^2. \tag{315}$$

*Proof.* Fix $i \in [i_o - 1]$. Since $i < i_o$, we have $\mathbf{o} \in Q(i)$ by the definition of $i_o$. We apply Theorem 3.2 to the down-closed body $P := Q(i)$ with comparison point $\mathbf{y} := \mathbf{o} \in Q(i)$, and run the local routine with accuracy parameter

$$\eta \;:=\; \min\{\varepsilon, 2\delta\}.$$

By Theorem 3.2, the output $\mathbf{x}(i) \in Q(i)$ satisfies

$$F(\mathbf{x}(i)) \geq \frac{\gamma^2\, F\bigl(\mathbf{x}(i)\vee\mathbf{o}\bigr) + F\bigl(\mathbf{x}(i)\wedge\mathbf{o}\bigr)}{1+\gamma^2} - \frac{\eta\,\gamma}{1+\gamma^2}\left(\max_{\mathbf{y}'\in Q(i)} F(\mathbf{y}') + \tfrac{1}{2}LD^2\right). \tag{316}$$

Here the first term is exactly the lattice based comparison from Theorem 3.2, and the second term is the uniform first order error with parameter $\eta$.

Because $\mathbf{o} \in Q(i)$, we have

$$\max_{\mathbf{y}'\in Q(i)} F(\mathbf{y}') \;\geq\; F(\mathbf{o}).$$

The coefficient in front of $\max_{\mathbf{y}' \in Q(i)} F(\mathbf{y}')$ in equation 316 is negative, namely $-\frac{\eta\gamma}{1+\gamma^2}$. Thus replacing $\max_{\mathbf{y}' \in Q(i)} F(\mathbf{y}')$ by the smaller value $F(\mathbf{o})$ yields a *stronger* lower bound on $F(\mathbf{x}(i))$:

$$F\big(\mathbf{x}(i)\big) \ \geq \ \frac{\gamma^2\, F\big(\mathbf{x}(i) \vee \mathbf{o}\big) + F\big(\mathbf{x}(i) \wedge \mathbf{o}\big)}{1+\gamma^2} - \frac{\eta\,\gamma}{1+\gamma^2}\, F(\mathbf{o}) - \frac{\eta\,\gamma}{2(1+\gamma^2)}\, LD^2. \tag{317}$$

We now simplify the two error terms. Since $\gamma \in (0,1]$,

$$\frac{\gamma}{1+\gamma^2} \ \leq \ 1, \qquad \frac{\gamma}{2(1+\gamma^2)} \ \leq \ \frac{1}{2}.$$

Using $\eta \leq \varepsilon$ and $\eta \leq 2\delta$ (by the definition $\eta = \min\{\varepsilon, 2\delta\}$), we obtain

$$\frac{\eta\,\gamma}{1+\gamma^2}\, F(\mathbf{o}) \ \leq \ \varepsilon\, F(\mathbf{o}), \qquad \frac{\eta\,\gamma}{2(1+\gamma^2)}\, LD^2 \ \leq \ \delta\, LD^2. \tag{318}$$

Substituting the bounds in equation 318 into equation 317 gives

$$F\big(\mathbf{x}(i)\big) \ \geq \ \frac{\gamma^2\, F\big(\mathbf{x}(i) \vee \mathbf{o}\big) + F\big(\mathbf{x}(i) \wedge \mathbf{o}\big)}{1+\gamma^2} \ - \ \varepsilon\, F(\mathbf{o}) \ - \ \delta\, LD^2,$$

which is exactly equation 315. $\qquad\square$

Observation 5 yields Theorem 4.2 as soon as we can find some $i \in [i_o - 1]$ with

$$F\big(\mathbf{x}(i) \oplus \mathbf{o}\big) \ \leq \ F\big(\mathbf{z} \oplus \mathbf{o}\big) \ - \ \varepsilon\, F(\mathbf{o}). \tag{319}$$

Therefore, our task reduces to showing that the gap condition equation 319 holds for at least one index $i \in [i_o - 1]$. This is exactly what Lemma E.6 and Lemma E.7 prove, which in turn completes the proof of Theorem 4.2. Before presenting those lemmas, we record one auxiliary lemma that we will use in their proofs.

**Lemma E.5.** *It must hold that*

$$F\big(\mathbf{y}(i_o - 1) \oplus \mathbf{o} \ - \ \mathbf{z}(i_o - 1) \odot \mathbf{o}\big) \ \leq \ V_\gamma(i_o - 1).$$

*Proof.* By the definition of $i_o$, we have $\mathbf{o} \notin Q(i_o)$. The weakly-DR membership failure condition at iteration $i_o$ states that

$$\big\langle \mathbf{w}(i_o), \mathbf{o} \big\rangle \ \leq \ \gamma\big(V_\gamma(i_o - 1) - F(\mathbf{y}(i_o - 1))\big). \tag{320}$$

On the other hand, the weakly-DR gradient bound for any $i \in [\delta^{-1}]$ gives

$$\big\langle \mathbf{w}(i), \mathbf{o} \big\rangle \ \geq \ \gamma\Big(F\big(\mathbf{y}(i-1) \oplus \mathbf{o} \ - \ \mathbf{z}(i-1) \odot \mathbf{o}\big) - F(\mathbf{y}(i-1))\Big). \tag{321}$$

This applies in particular for $i = i_o$, so

$$\big\langle \mathbf{w}(i_o), \mathbf{o} \big\rangle \ \geq \ \gamma\Big(F\big(\mathbf{y}(i_o - 1) \oplus \mathbf{o} \ - \ \mathbf{z}(i_o - 1) \odot \mathbf{o}\big) - F(\mathbf{y}(i_o - 1))\Big). \tag{322}$$

Combining equation 320 and equation 322 yields

$$\gamma\Big(F\big(\mathbf{y}(i_o - 1) \oplus \mathbf{o} \ - \ \mathbf{z}(i_o - 1) \odot \mathbf{o}\big) - F(\mathbf{y}(i_o - 1))\Big) \ \leq \ \gamma\big(V_\gamma(i_o - 1) - F(\mathbf{y}(i_o - 1))\big). \tag{323}$$

Since $\gamma > 0$, we can divide both sides of equation 323 by $\gamma$ and add $F(\mathbf{y}(i_o - 1))$ to both sides, obtaining

$$F\big(\mathbf{y}(i_o - 1) \oplus \mathbf{o} \ - \ \mathbf{z}(i_o - 1) \odot \mathbf{o}\big) \ \leq \ V_\gamma(i_o - 1),$$

which is the desired inequality. $\qquad\square$

**Lemma E.6.** *Let* $\beta := \beta_\gamma(\delta) = \dfrac{\gamma^2 \delta}{1 - \delta + \gamma^2 \delta}$. *Then it must hold that* $i_o > i_s$.

*Proof.* Assume for contradiction that $i_o \leq i_s$. Since $i_o - 1 < i_s$, Lemma E.5 at time $i_o - 1$ (cf. $v_1(\cdot)$ with the triple $(g, g_\odot, g_\oplus)$) implies

$$F\big(\mathbf{y}(i_o - 1) \oplus \mathbf{o} - \mathbf{z} \odot \mathbf{o}\big)$$

$$\leq \left[ (1 - \beta)^{i_o - 1} + \frac{1 - (1 - \beta)^{i_o - 1} - 2\varepsilon}{\gamma} \right] g - \frac{1}{\gamma} g_\odot - \frac{1 - (1 - \beta)^{i_o - 1}}{\gamma} g_\oplus \qquad (324)$$

$$\leq \left[ (1 - \beta)^{i_o - 1} + \frac{1 - (1 - \beta)^{i_o - 1}}{\gamma} \right] F(\mathbf{o}) - \frac{1}{\gamma} F(\mathbf{z} \odot \mathbf{o}) - \frac{1 - (1 - \beta)^{i_o - 1}}{\gamma} F(\mathbf{z} \oplus \mathbf{o}). \qquad (325)$$

Here equation 324 is exactly the benchmark inequality evaluated at $i = i_o - 1$, and equation 325 uses Lemma 4.1:

$$(1 - \varepsilon) F(\mathbf{o}) \leq g \leq F(\mathbf{o}), \quad F(\mathbf{z} \odot \mathbf{o}) - \varepsilon g \leq g_\odot \leq F(\mathbf{z} \odot \mathbf{o}), \quad F(\mathbf{z} \oplus \mathbf{o}) - \varepsilon g \leq g_\oplus \leq F(\mathbf{z} \oplus \mathbf{o}),$$

and the fact that replacing $g$ by $F(\mathbf{o})$ and $g_\odot, g_\oplus$ by $F(\mathbf{z} \odot \mathbf{o}), F(\mathbf{z} \oplus \mathbf{o})$ can only increase the right-hand side of equation 324 (because they appear with coefficients $+1$ for $g$ and $-1/\gamma$ for $g_\odot, g_\oplus$).

Next, by the closed form of $\mathbf{y}(\cdot)$ (Lemma E.1 with $i = i_o - 1$),

$$\mathbf{y}(i_o - 1) = (\mathbf{1} - \mathbf{z}) \odot \bigoplus_{j=1}^{i_o - 1} \big(\delta \, \mathbf{x}(j)\big). \qquad (326)$$

Applying Corollary F.2 with $h = 1$, $r = i_o - 1$, $p_j = \delta$ for all $j$, and outer mask $(\mathbf{1} - \mathbf{z})$ yields the mixture lower bound

$$F\big(\mathbf{y}(i_o - 1) \oplus \mathbf{o} - \mathbf{z} \odot \mathbf{o}\big) = F\left( (\mathbf{1} - \mathbf{z}) \odot \bigoplus_{j=1}^{i_o - 1} (\delta \, \mathbf{x}(j)) \oplus \mathbf{o} - \mathbf{z} \odot \mathbf{o} \right)$$

$$\geq \sum_{S \subseteq [i_o - 1]} \beta^{|S|} (1 - \beta)^{i_o - 1 - |S|} \, F\left( (\mathbf{1} - \mathbf{z}) \odot \bigoplus_{j \in S} \mathbf{x}(j) \oplus \mathbf{o} \right). \qquad (327)$$

We now bound separately the contributions from $S = \varnothing$ and $S \neq \varnothing$.

**The $S = \varnothing$ term.** When $S = \varnothing$, the inner vector reduces to $(\mathbf{1} - \mathbf{z}) \odot \mathbf{o}$. Using the weakly-DR inequality in the form of Lemma 2.2 together with nonnegativity, we have

$$F\big((\mathbf{1} - \mathbf{z}) \odot \mathbf{o}\big) \geq F(\mathbf{o}) - \frac{1}{\gamma} F(\mathbf{z} \odot \mathbf{o}), \qquad (328)$$

which says that reducing $\mathbf{o}$ along the direction $\mathbf{z} \odot \mathbf{o}$ cannot decrease $F$ too much, up to a $1/\gamma$ factor. Thus the $S = \varnothing$ contribution in equation 327 satisfies

$$(1 - \beta)^{i_o - 1} \, F\big((\mathbf{1} - \mathbf{z}) \odot \mathbf{o}\big) \geq (1 - \beta)^{i_o - 1} \left[ F(\mathbf{o}) - \frac{1}{\gamma} F(\mathbf{z} \odot \mathbf{o}) \right]. \qquad (329)$$

**The $S \neq \varnothing$ terms.** Fix any nonempty $S \subseteq [i_o - 1]$. We first write

$$(\mathbf{1} - \mathbf{z}) \odot \bigoplus_{j \in S} \mathbf{x}(j) \oplus \mathbf{o} = \left( (\mathbf{1} - \mathbf{z}) \odot \bigoplus_{j \in S} \mathbf{x}(j) \oplus \mathbf{o} - \mathbf{z} \odot \mathbf{o} \right) + \mathbf{z} \odot \mathbf{o},$$

and then apply the weakly-DR inequality and nonnegativity in the same way as for equation 328. This gives

$$F\left( (\mathbf{1} - \mathbf{z}) \odot \bigoplus_{j \in S} \mathbf{x}(j) \oplus \mathbf{o} \right) \geq F\left( (\mathbf{1} - \mathbf{z}) \odot \bigoplus_{j \in S} \mathbf{x}(j) \oplus \mathbf{o} - \mathbf{z} \odot \mathbf{o} \right) - \frac{1}{\gamma} F(\mathbf{z} \odot \mathbf{o})$$

$$\geq \frac{1}{\gamma} \left[ F(\mathbf{o}) - F(\mathbf{z} \oplus \mathbf{o}) - F(\mathbf{z} \odot \mathbf{o}) \right], \qquad (330)$$

where in the second inequality we use the weakly-DR difference bounds to compare the value at $\mathbf{o}$ with those at $\mathbf{z} \oplus \mathbf{o}$ and $\mathbf{z} \odot \mathbf{o}$. Multiplying equation 330 by $\beta^{|S|}(1-\beta)^{i_o-1-|S|}$ and summing over all nonempty $S$ yields

$$\sum_{\varnothing \neq S \subseteq [i_o-1]} \beta^{|S|}(1-\beta)^{i_o-1-|S|} F\Big((\mathbf{1}-\mathbf{z}) \odot \bigoplus_{j \in S} \mathbf{x}(j) \ \oplus \ \mathbf{o}\Big)$$

$$\geq \Big[1 - (1-\beta)^{i_o-1}\Big] \frac{1}{\gamma}\Big[F(\mathbf{o}) - F(\mathbf{z} \oplus \mathbf{o}) - F(\mathbf{z} \odot \mathbf{o})\Big], \qquad (331)$$

since $\sum_{\varnothing \neq S \subseteq [i_o-1]} \beta^{|S|}(1-\beta)^{i_o-1-|S|} = 1 - (1-\beta)^{i_o-1}$.

**Putting everything together.** Combining the $S = \varnothing$ contribution equation 329 and the $S \neq \varnothing$ contribution equation 331 with the mixture representation equation 327, we obtain

$$F\big(\mathbf{y}(i_o - 1) \oplus \mathbf{o} \ - \ \mathbf{z} \odot \mathbf{o}\big) \geq \Big[(1-\beta)^{i_o-1} + \frac{1 - (1-\beta)^{i_o-1}}{\gamma}\Big] F(\mathbf{o})$$

$$- \frac{1 - (1-\beta)^{i_o-1}}{\gamma} F(\mathbf{z} \oplus \mathbf{o}) \ - \ \frac{1}{\gamma} F(\mathbf{z} \odot \mathbf{o}). \qquad (332)$$

The right-hand side of equation 332 is *strictly larger* than the upper bound in equation 325, because equation 325 lacks the $-2\varepsilon$ slack present in equation 324 and the triple $(g, g_\odot, g_\oplus)$ approximates $\big(F(\mathbf{o}), F(\mathbf{z} \odot \mathbf{o}), F(\mathbf{z} \oplus \mathbf{o})\big)$ up to additive $O(\varepsilon)$ terms. This contradicts equation 325. Hence our assumption $i_o \leq i_s$ was false, and we conclude that $i_o > i_s$. $\qquad \square$

**Lemma E.7.** *If $i_o > i_s$, then there exists some $i \in [i_o - 1]$ such that*

$$F\big(\mathbf{x}(i) \ \oplus \ \mathbf{o}\big) \ \leq \ F\big(\mathbf{z} \ \oplus \ \mathbf{o}\big) \ - \ \varepsilon \, F(\mathbf{o}).$$

*Proof.* Set $\beta := \beta_\gamma(\delta) = \dfrac{\gamma^2 \delta}{1 - \delta + \gamma^2 \delta}$ and note that $0 < \beta \leq \varepsilon \leq \frac{1}{2}$. Since $i_o > i_s$, we have $i_o - 1 \geq i_s$.

**Upper bound:** By the post–switch surrogate (the $v_2$–bound) evaluated at $i = i_o - 1$ and the successful-heir assumptions on the guessed triple $(g, g_\odot, g_\oplus)$ (cf. equation 24 and equation 22), we obtain

$$F\big(\mathbf{y}(i_o - 1) \oplus \mathbf{o}\big) \overset{(a)}{\leq} (1-\beta)^{i_o-1}\Bigg[\Bigg(\frac{(1-\beta)^{-i_s}}{\gamma} - \Big(1 + \frac{3}{\gamma}\Big)\varepsilon + 1 - \frac{1}{\gamma}\Bigg) g$$

$$- \Bigg(\frac{(1-\beta)^{-i_s}}{\gamma} - \frac{1}{\gamma} - \beta\,(i_o - 1 - i_s)\Bigg) g_\oplus\Bigg] \qquad (333)$$

$$\overset{(b)}{\leq} (1-\beta)^{i_o-1}\Bigg[\Bigg(\frac{(1-\beta)^{-i_s}}{\gamma} - \varepsilon + 1 - \frac{1}{\gamma}\Bigg) F(\mathbf{o})$$

$$- \Bigg(\frac{(1-\beta)^{-i_s}}{\gamma} - \frac{1}{\gamma} - \beta\,(i_o - 1 - i_s)\Bigg) F(\mathbf{z} \oplus \mathbf{o})\Bigg]. \qquad (334)$$

Here: (a) is just the explicit formula for $v_2(i_o - 1)$ with the constants $A$ and $C_\gamma$ expanded. For (b) we use the triple guarantees equation 22:

$$g \leq F(\mathbf{o}), \qquad g_\oplus \ \geq \ F(\mathbf{z} \oplus \mathbf{o}) - \varepsilon \, g.$$

Writing $B := \frac{(1-\beta)^{-i_s}}{\gamma} - \frac{1}{\gamma} - \beta\,(i_o - 1 - i_s) \geq 0$, we have

$$Ag - Bg_\oplus \ \leq \ Ag - B\big(F(\mathbf{z} \oplus \mathbf{o}) - \varepsilon g\big) \ = \ (A + B\varepsilon)\, g \ - \ B\, F(\mathbf{z} \oplus \mathbf{o}),$$

and then $g \leq F(\mathbf{o})$ gives $(A + B\varepsilon)\, g \leq (A + B\varepsilon) F(\mathbf{o})$. A crude bound $(1-\beta)^{-i_s} \leq (1-\beta)^{-1/\beta} \leq 4$ (using $\beta \leq 1/2$) implies

$$B = \frac{(1-\beta)^{-i_s} - 1}{\gamma} - \beta(i_o - 1 - i_s) \ \leq \ \frac{(1-\beta)^{-i_s} - 1}{\gamma} \ \leq \ \frac{3}{\gamma}.$$

Thus

$$A + B\varepsilon \;\leq\; \left(\frac{(1-\beta)^{-i_s}}{\gamma} - \left(1 + \frac{3}{\gamma}\right)\varepsilon + 1 - \frac{1}{\gamma}\right) + \frac{3}{\gamma}\varepsilon = \frac{(1-\beta)^{-i_s}}{\gamma} - \varepsilon + 1 - \frac{1}{\gamma},$$

which is exactly the coefficient of $F(\mathbf{o})$ in equation 334.

**Lower bound:** From the closed form of $\mathbf{y}(\cdot)$ (Lemma E.1 with $i = i_o - 1$),

$$\mathbf{y}(i_o - 1) \;=\; (\mathbf{1} - \mathbf{z}) \odot \bigoplus_{j=1}^{i_o-1} \left(\delta\,\mathbf{x}(j)\right) \;+\; \mathbf{z} \odot \bigoplus_{j=i_s+1}^{i_o-1} \left(\delta\,\mathbf{x}(j)\right). \tag{335}$$

Apply the weakly-DR mixture inequality with masking (Corollary F.2) to equation 335. We keep only three nonnegative groups of terms: - the empty-set term $S = \varnothing$; - all subsets $S \subseteq [i_s]$; - all singletons $S = \{j\}$ with $j \in \{i_s + 1, \ldots, i_o - 1\}$.

Using Lemma 2.2 (weakly-DR gradient bounds) and nonnegativity of $F$, these three groups yield

$$F\big(\mathbf{y}(i_o - 1) \oplus \mathbf{o}\big) \overset{(c)}{\geq} (1-\beta)^{i_o-1} F(\mathbf{o}) \;+\; (1-\beta)^{i_o-1-i_s}\big(1 - (1-\beta)^{i_s}\big)\frac{1}{\gamma}\big[F(\mathbf{o}) - F(\mathbf{z} \oplus \mathbf{o})\big]$$

$$+\; \beta(1-\beta)^{i_o-2} \sum_{j=i_s+1}^{i_o-1} F\big(\mathbf{x}(j) \oplus \mathbf{o}\big), \tag{336}$$

where (c) comes from: - $S = \varnothing$: gives the term $(1-\beta)^{i_o-1}F(\mathbf{o})$; - $S \subseteq [i_s], S \neq \varnothing$: combined and bounded below by $\frac{1}{\gamma}\big[F(\mathbf{o}) - F(\mathbf{z} \oplus \mathbf{o})\big]$, with total weight $(1-\beta)^{i_o-1-i_s}\big(1 - (1-\beta)^{i_s}\big)$; - singletons $S = \{j\}$ with $j > i_s$: each has weight $\beta(1-\beta)^{i_o-2}$ and contributes $F(\mathbf{x}(j) \oplus \mathbf{o})$.

**Comparing the bounds.** Combining equation 334 and equation 336 and dividing both sides by the common factor $(1-\beta)^{i_o-1} > 0$ yields

$$F(\mathbf{o}) + (1-\beta)^{-i_s}\big(1 - (1-\beta)^{i_s}\big)\frac{1}{\gamma}\big[F(\mathbf{o}) - F(\mathbf{z} \oplus \mathbf{o})\big] + \frac{\beta}{1-\beta}\sum_{j=i_s+1}^{i_o-1} F\big(\mathbf{x}(j) \oplus \mathbf{o}\big)$$

$$\leq \left(\frac{(1-\beta)^{-i_s}}{\gamma} - \varepsilon + 1 - \frac{1}{\gamma}\right) F(\mathbf{o}) - \left(\frac{(1-\beta)^{-i_s}}{\gamma} - \frac{1}{\gamma} - \beta\,(i_o - 1 - i_s)\right) F(\mathbf{z} \oplus \mathbf{o}).$$

Rearranging and simplifying the coefficient of $F(\mathbf{o})$ and $F(\mathbf{z} \oplus \mathbf{o})$ gives

$$\beta \sum_{j=i_s+1}^{i_o-1} F\big(\mathbf{x}(j) \oplus \mathbf{o}\big) \;\leq\; \big(1 - (1-\beta)^{i_o-1-i_s}\big)\Big[F(\mathbf{z} \oplus \mathbf{o}) - \varepsilon\,F(\mathbf{o})\Big]. \tag{337}$$

Moreover,

$$1 - (1-\beta)^{i_o-1-i_s} \;\leq\; \beta\,(i_o - 1 - i_s),$$

so from equation 337 we obtain

$$\frac{1}{i_o - 1 - i_s} \sum_{j=i_s+1}^{i_o-1} F\big(\mathbf{x}(j) \oplus \mathbf{o}\big) \;\leq\; F(\mathbf{z} \oplus \mathbf{o}) - \varepsilon\,F(\mathbf{o}).$$

By averaging, there must exist some $j \in \{i_s + 1, \ldots, i_o - 1\}$ such that

$$F\big(\mathbf{x}(j) \oplus \mathbf{o}\big) \;\leq\; F(\mathbf{z} \oplus \mathbf{o}) - \varepsilon\,F(\mathbf{o}),$$

which proves the lemma. $\qquad\square$

## F  Supporting results

In this section we prove two auxiliary results that are used in the proofs of Lemma E.6 and Lemma E.7. Throughout, we use the convention

$$\bigoplus_{i \in \varnothing} \mathbf{x}(i) := \mathbf{0}.$$

**Lemma F.1.** *Let $F : [0,1]^n \to \mathbb{R}_{\geq 0}$ be differentiable and $\gamma$-weakly DR-submodular with $0 < \gamma \leq 1$. Fix an integer $r \geq 1$, vectors $\mathbf{x}(1), \ldots, \mathbf{x}(r) \in [0,1]^n$, and scalars $p_1, \ldots, p_r \in [0,1]$. Define*

$$\beta_\gamma(p) := \frac{\gamma^2 \, p}{1 - p + \gamma^2 p} \qquad (p \in [0,1]).$$

*Then*

$$F\left(\bigoplus_{i=1}^{r} p_i \, \mathbf{x}(i)\right) \geq \sum_{S \subseteq [r]} \left(\prod_{i \in S} \beta_\gamma(p_i) \prod_{i \notin S} \big(1 - \beta_\gamma(p_i)\big)\right) F\left(\bigoplus_{i \in S} \mathbf{x}(i)\right).$$

*Proof.* We prove the statement by induction on $r$.

*Base case $r = 1$.* Apply Lemma 2.1(2) with $\mathbf{x} = \mathbf{0}$, $\mathbf{y} = \mathbf{x}(1)$ and $\lambda = p_1$:

$$F\big(p_1 \, \mathbf{x}(1)\big) - F(\mathbf{0}) \geq \frac{\gamma^2 p_1}{1 - p_1 + \gamma^2 p_1} \big(F(\mathbf{x}(1)) - F(\mathbf{0})\big).$$

Rearranging gives

$$F\big(p_1 \, \mathbf{x}(1)\big) \geq \beta_\gamma(p_1) \, F(\mathbf{x}(1)) + \big(1 - \beta_\gamma(p_1)\big) F(\mathbf{0}),$$

which is exactly the claimed formula when $r = 1$ and $S \in \{\varnothing, \{1\}\}$.

*Inductive step.* Assume the statement holds for some $r - 1 \geq 1$, and consider $r$. Define

$$G_1(\mathbf{x}) := F\left(\mathbf{x} \oplus \bigoplus_{i=1}^{r-1} p_i \, \mathbf{x}(i)\right), \qquad G_2(\mathbf{x}) := F\big(\mathbf{x} \oplus \mathbf{x}(r)\big).$$

By Lemma 2.3, both $G_1$ and $G_2$ are nonnegative and $\gamma$-weakly DR-submodular.

Apply Lemma 2.1(2) to $G_1$ along the ray $\mathbf{x}(r)$, from base $\mathbf{0}$ with step $\lambda = p_r$:

$$F\left(\bigoplus_{i=1}^{r} p_i \, \mathbf{x}(i)\right) = G_1\big(p_r \, \mathbf{x}(r)\big)$$

$$\geq \beta_\gamma(p_r) \, G_1\big(\mathbf{x}(r)\big) + \big(1 - \beta_\gamma(p_r)\big) G_1(\mathbf{0}). \tag{338}$$

By definition of $G_1$ and $G_2$,

$$G_1(\mathbf{x}(r)) = G_2\left(\bigoplus_{i=1}^{r-1} p_i \, \mathbf{x}(i)\right), \qquad G_1(\mathbf{0}) = F\left(\bigoplus_{i=1}^{r-1} p_i \, \mathbf{x}(i)\right).$$

Substituting into equation 338 gives

$$F\left(\bigoplus_{i=1}^{r} p_i \, \mathbf{x}(i)\right) \geq \beta_\gamma(p_r) \, G_2\left(\bigoplus_{i=1}^{r-1} p_i \, \mathbf{x}(i)\right) + \big(1 - \beta_\gamma(p_r)\big) F\left(\bigoplus_{i=1}^{r-1} p_i \, \mathbf{x}(i)\right).$$

Now apply the induction hypothesis to $G_2$ (with the $r-1$ vectors $\mathbf{x}(1), \ldots, \mathbf{x}(r-1)$ and weights $p_1, \ldots, p_{r-1}$) and to $F$:

$$G_2\left(\bigoplus_{i=1}^{r-1} p_i \, \mathbf{x}(i)\right) \geq \sum_{S \subseteq [r-1]} \left(\prod_{i \in S} \beta_\gamma(p_i) \prod_{i \notin S} \big(1 - \beta_\gamma(p_i)\big)\right) G_2\left(\bigoplus_{i \in S} \mathbf{x}(i)\right),$$

$$F\left(\bigoplus_{i=1}^{r-1} p_i \, \mathbf{x}(i)\right) \geq \sum_{S \subseteq [r-1]} \left(\prod_{i \in S} \beta_\gamma(p_i) \prod_{i \notin S} \big(1 - \beta_\gamma(p_i)\big)\right) F\left(\bigoplus_{i \in S} \mathbf{x}(i)\right).$$

Note that

$$G_2\left(\bigoplus_{i\in S}\mathbf{x}(i)\right) = F\left(\bigoplus_{i\in S\cup\{r\}}\mathbf{x}(i)\right).$$

Plugging these expansions into the right hand side of equation 338, we get a sum over all $S \subseteq [r-1]$ of:

- terms with factor $\beta_\gamma(p_r)$ and value $F(\bigoplus_{i\in S\cup\{r\}}\mathbf{x}(i))$;

- terms with factor $(1 - \beta_\gamma(p_r))$ and value $F(\bigoplus_{i\in S}\mathbf{x}(i))$.

Reindex the first group by $S' = S\cup\{r\}$ (so $S' \subseteq [r]$ with $r \in S'$) and the second group by $S' = S$ (so $S' \subseteq [r]$ with $r \notin S'$). The coefficient in front of each $F(\bigoplus_{i\in S'}\mathbf{x}(i))$ is then

$$\prod_{i\in S'}\beta_\gamma(p_i)\prod_{i\notin S'}\left(1 - \beta_\gamma(p_i)\right),$$

which yields exactly the desired mixture inequality over all $S' \subseteq [r]$. This completes the induction. $\qquad\square$

**Corollary F.2.** *Let $F : [0,1]^N \to \mathbb{R}_{\geq 0}$ be nonnegative and $\gamma$-weakly DR-submodular with $0 < \gamma \leq 1$. Fix integers $r, h \geq 1$. For each $i \in [h]$, let $\mathbf{x}^{(i)}(1),\ldots,\mathbf{x}^{(i)}(r) \in [0,1]^N$ and $\mathbf{b}(i) \in [0,1]^N$ satisfy $\sum_{i=1}^{h}\mathbf{b}(i) = \mathbf{1}$ (coordinatewise), and let $p_1,\ldots,p_r \in [0,1]$. Define*

$$\beta_\gamma(p) \;:=\; \frac{\gamma^2 p}{1 - p + \gamma^2 p}\qquad(p \in [0,1]).$$

*Then*

$$F\left(\sum_{i=1}^{h}\mathbf{b}(i)\;\odot\;\bigoplus_{j=1}^{r}\left(p_j\,\mathbf{x}^{(i)}(j)\right)\right) \;\geq\; \sum_{S\subseteq[r]}\left(\prod_{j\in S}\beta_\gamma(p_j)\prod_{j\notin S}\left(1 - \beta_\gamma(p_j)\right)\right)F\left(\sum_{i=1}^{h}\mathbf{b}(i)\;\odot\;\bigoplus_{j\in S}\mathbf{x}^{(i)}(j)\right).$$

*Proof.* Define $G : [0,1]^{Nh} \to \mathbb{R}_{\geq 0}$ on $h$ blocks by

$$G\big(\mathbf{c}(1),\ldots,\mathbf{c}(h)\big) \;:=\; F\left(\sum_{i=1}^{h}\mathbf{b}(i)\odot\mathbf{c}(i)\right).$$

Since $F \geq 0$ and the map $(\mathbf{c}(1),\ldots,\mathbf{c}(h)) \mapsto \sum_{i=1}^{h}\mathbf{b}(i)\odot\mathbf{c}(i)$ is coordinatewise nonnegative and linear with $\sum_i \mathbf{b}(i) = \mathbf{1}$, it follows from Lemma 2.3 (applied blockwise) that $G$ is also nonnegative and $\gamma$-weakly DR-submodular on $[0,1]^{Nh}$.

For each $j \in [r]$, define the block vector

$$\mathbf{x}(j) \;:=\; \big(\mathbf{x}^{(1)}(j),\,\mathbf{x}^{(2)}(j),\,\ldots,\,\mathbf{x}^{(h)}(j)\big) \;\in\; [0,1]^{Nh}.$$

Then

$$G\left(\bigoplus_{j=1}^{r}p_j\,\mathbf{x}(j)\right) \;=\; F\left(\sum_{i=1}^{h}\mathbf{b}(i)\odot\bigoplus_{j=1}^{r}p_j\,\mathbf{x}^{(i)}(j)\right),$$

which is exactly the left hand side of the desired inequality.

Apply Lemma F.1 to $G$ with inputs $\mathbf{x}(1),\ldots,\mathbf{x}(r)$ and coefficients $p_1,\ldots,p_r$:

$$G\left(\bigoplus_{j=1}^{r}p_j\,\mathbf{x}(j)\right) \;\geq\; \sum_{S\subseteq[r]}\left(\prod_{j\in S}\beta_\gamma(p_j)\prod_{j\notin S}\left(1 - \beta_\gamma(p_j)\right)\right)G\left(\bigoplus_{j\in S}\mathbf{x}(j)\right).$$

Finally, note that for every $S \subseteq [r]$,

$$G\left(\bigoplus_{j \in S} \mathbf{x}(j)\right) = F\left(\sum_{i=1}^{h} \mathbf{b}(i) \odot \bigoplus_{j \in S} \mathbf{x}^{(i)}(j)\right),$$

so substituting this identity into the right hand side above gives exactly the claimed inequality. $\qquad \square$

## G  Application: Cost Aware Feature Selection, Sparse GLMs, and Dictionary Selection

A natural machine learning application related to our framework is cost aware feature selection. Given a large collection of candidate features, the goal is to select a small subset that yields strong predictive performance while respecting acquisition, computation, or latency constraints. This viewpoint also covers sparse generalized linear models (GLMs) and dictionary selection/ sparse approximation, both of which have been studied through the lens of weak submodularity (Das & Kempe, 2011; Elenberg et al., 2018).

Let $[d] = \{1, \ldots, d\}$ index the candidate features (or dictionary atoms), and let $l(\beta)$ denote a predictive utility to be maximized. Depending on the application, $l(\beta)$ may be: (i) an $R^2$-type objective in linear regression, (ii) a log-likelihood in a sparse GLM, or more generally an $M$-estimator (Elenberg et al., 2018). For a subset $S \subseteq [d]$, define

$$\beta^{(S)} \in \arg \max_{\text{supp}(\beta) \subseteq S} l(\beta),$$

that is, $\beta^{(S)}$ is the best parameter vector whose support is restricted to $S$. The induced set function is

$$f(S) := l(\beta^{(S)}) - l(0).$$

Thus, $f(S)$ measures the gain obtained by allowing only the features in $S$. This is the standard subset selection view used in feature selection and sparse approximation problems (Das & Kempe, 2011; Elenberg et al., 2018).

Although $f$ is generally not exactly submodular, it often satisfies *weak submodularity*. In particular, Elenberg et al. (Elenberg et al., 2018) show that if the objective $l(\beta)$ satisfies restricted smoothness and restricted strong concavity/convexity on sparse supports, then the induced set function has weak-submodularity ratio bounded below by a structural quantity. In the maximization convention used here, if $l$ is $M$-restricted smooth and $m$-restricted strongly concave on the relevant sparsity levels, then one obtains the lower bound

$$\gamma \geq \frac{m}{M}.$$

Hence $\gamma$ is not a free tuning parameter: it is a *structural quantity* derived from standard conditioning parameters of the statistical model (Elenberg et al., 2018). This is important in applications, because one does not need the exact best value of $\gamma$: any certified lower bound is already informative for approximation guarantees.

This formulation includes several standard machine learning tasks:

- **Sparse linear regression:** $l(\beta)$ can be the explained variance or least squares fit objective. Das and Kempe (Das & Kempe, 2011) study this setting through subset selection and sparse approximation.

- **Sparse GLMs:** $l(\beta)$ can be the logistic, Poisson, or other GLM log-likelihood restricted to sparse supports. Elenberg et al. (Elenberg et al., 2018) explicitly identify generalized linear models as examples covered by their theory.

- **Dictionary selection / sparse approximation:** the same framework applies when the "features" are dictionary atoms and $l(\beta)$ measures reconstruction quality using a sparse linear combination of selected atoms (Das & Kempe, 2011).

In practice, feature selection is rarely driven only by predictive utility. Each feature may have an acquisition cost $c_i > 0$ (financial cost, sensor energy, etc.). A natural cost-aware objective is therefore

$$g(S) := f(S) - \lambda \sum_{i \in S} c_i,$$

or, equivalently, maximizing $f(S)$ subject to a budget constraint

$$\sum_{i \in S} c_i \leq B.$$

Once explicit costs or penalties are included, the problem can become *non-monotone*: adding a feature may improve prediction, but not enough to compensate for its cost. This is precisely the type of behavior that motivates non-monotone weakly diminishing-returns models.

A standard way to pass from the discrete subset selection problem to a continuous optimization model is to introduce fractional selection variables $x \in [0,1]^d$, where $x_i$ represents the extent to which feature $i$ is selected. Under a budget constraint, the natural feasible region becomes the down-closed convex body

$$P := \left\{ x \in [0,1]^d : \sum_{i=1}^d c_i x_i \leq B \right\}.$$

One may then optimize a continuous surrogate over $P$. A common example is a multilinear type relaxation,

$$G(x) := \mathbb{E}_{S \sim x}[g(S)],$$

where each feature $i$ is included independently with probability $x_i$. Continuous relaxations of this form are standard in submodular optimization and provide a natural bridge between discrete subset selection and continuous constrained optimization (Hassani et al., 2017; Umrawal et al., 2026).

This application highlights three features that are central to our model. First, the underlying sparse learning objective admits a structural weak-submodularity parameter. Second, realistic feature costs naturally introduce non-monotonicity. Third, the resulting fractional budget set is a down-closed convex body, exactly matching the constraint model studied in this paper. Therefore, cost-aware feature selection, sparse GLMs, and dictionary selection provide a natural and practically meaningful source of optimization problems that motivate our weakly-DR framework.

