# OpenReview forum: "Stronger Approximation Guarantees for Non-Monotone $\gamma$-Weakly DR-Submodular Maximization"
_TMLR — Decision pending for TMLR_

### Review · Reviewer_Kvzb · 2026-02-25

**Summary Of Contributions:**

The paper studies the problem of optimizing a gamma-weakly-DR-submodular function under down-closed convex bodies constraints. The paper observes that compared to the ordinary DR-submodular case, the weakly case has the imbalanced behavior, and therefore more efforts are required to achieve a better bound. The paper proposes a combination of the FWG algorithm and the local double greedy search algorithm to achieve a \theta \gamma bound with Poly(n, \delta^-1) running time.

**Audience:**

No

**Audience Explanation:**

The paper is theoretical, and the proposed algorithm is sort of "engineered" to achieve an improved guarantee, without quite significant insight into the problem itself. My major concern is that the application of machine learning is almost completely missing in the current paper.
1. The problem studied is a bit narrow/specific: optimizing a \gamma-weakly-DR-submodular function under convex down-closed constraints. The paper does not provide enough application examples for the target problem, e.g., which machine learning problems can be formalized as the target problem.

2. The designed algorithm assumes knowledge of the \gamma parameter. I am mostly concerned with this. Even if there are machine learning problems that can be naturally formulated as the target problem, it is very unlikely to know or estimate a precise \gamma value. I believe it is theoretically infeasible to estimate \gamma of an arbitrary function. This greatly limits the applicability of the proposed algorithm.

3. The paper does not have any empirical studies. I acknowledge that empirical results are not necessary in a pure theoretical paper, but they are a great bonus if the proposed method/problem can be used in real machine learning tasks.

**Claims And Evidence:**

Yes

**Claims Explanation:**

The paper solves a theoretical optimization problem with rigorous proofs. The design of the algorithm is clear and intuitive, given the arguments about the local optimality properties presented.

**Requested Changes:**

Please add examples of potential applications or some empirical evaluations, if possible.

---

> ### Author Response · Authors · 2026-04-18
>
> We sincerely thank the reviewer for the detailed and thoughtful feedback. We are encouraged that the reviewer found the technical claims convincing and the algorithmic design clear.
>
> **(1) On the concern that the paper is narrow/insufficiently motivated for ML**
>
> We thank the reviewer for raising this point. We agree that the previous version did not sufficiently highlight concrete ML/AI applications of this problem class. In the revised version, we have added a dedicated appendix section on applications (Appendix G). This shows that optimizing $\gamma$-weakly DR-submodular functions over down-closed convex sets is not merely a narrow abstract setting, but also arises naturally in cost aware feature selection and related subset selection problems in machine learning.
>
> **(2) On the assumption that the algorithm knows $\gamma$**
>
>
> In algorithm we used a paramenter $\gamma$, if we know then we get exact approximation guarantee based on the value of $\gamma$.
> But if we don't know the parameter $\gamma
> $, then the algorithm does not require the exact maximal weakly DR parameter. It is enough to provide any valid lower bound $\hat\gamma \le \gamma$. Indeed, if a function $F$ is $\gamma$-weakly DR-submodular, then it is automatically $\hat\gamma$-weakly DR-submodular for every $\hat\gamma\in(0,\gamma]$, since
> $$
> F(\mathbf{x}+c\mathbf{e}_i)-F(\mathbf{x})
> \ge
> \gamma\bigl(F(\mathbf{y}+c\mathbf{e}_i)-F(\mathbf{y})\bigr)
> $$
> immediately implies
> $$
> F(\mathbf{x}+c\mathbf{e}_i)-F(\mathbf{x})
> \ge
> \hat\gamma\bigl(F(\mathbf{y}+c\mathbf{e}_i)-F(\mathbf{y})\bigr)
> \qquad
> \text{for every } \hat\gamma \le \gamma.
> $$
>
> Therefore, exact knowledge of the sharp value of $\gamma$ is not required for correctness. A conservative lower bound is sufficient, and the resulting guarantee is then stated in terms of that supplied lower bound.
>
> This viewpoint is also consistent with how related structural parameters are used in the broader approximation literature. For example, in the weak-submodularity setting, approximation guarantees are often stated in terms of a lower bound on the relevant structural parameter, rather than its exact sharp value. In particular, Elenberg et al. [1] show that under restricted strong concavity and restricted smoothness, the weak-submodularity ratio admits a certified lower bound, and they give approximation guarantees from that lower bound. Although that work is in a different setting, it supports the same general principle that a structural lower bound is often sufficient for algorithmic guarantees.
>
>
> Accordingly, if an application admits a certified lower bound $\hat\gamma$, then our algorithm applies directly and returns a guarantee stated in terms of $\hat\gamma$.
>
> **(3) On empirical evaluation**
>
> We appreciate this suggestion. The current submission is primarily a theory paper, and its main contribution is a new worst case approximation guarantee with rigorous proofs. However, we agree that an empirical or illustrative component would improve accessibility for a broader audience. We therefore leave a careful empirical evaluation as an important direction for future work.
>
> [1] Ethan R Elenberg, Rajiv Khanna, Alexandros G Dimakis, and Sahand Negahban. Restricted strong convexity implies weak submodularity. The Annals of Statistics, 2018.

---

### Review · Reviewer_yMoB · 2026-03-11

**Summary Of Contributions:**

This paper studies approximation algorithms for optimizing non-monotone weakly DR-submodular maximization.

**Additional Comments:**

N/A

**Audience:**

Yes

**Audience Explanation:**

While approximation algorithms are more or less topics of theoretical computer science research, the study of optimization methods is of interest to machine learning researchers. If possible, the authors are encouraged to include some example applications of weakly DR optimization in machine learning or AI research.

**Claims And Evidence:**

Yes

**Claims Explanation:**

Results and proofs look reasonable

**Requested Changes:**

I only have two changes/comments for the authors to address.

1. Currently, the discussion of results of this paper, and comparison with existing works, are a bit confusing. By O(1/eps) or O(1/eps^3) complexity, does the author mean FPTAS? If this is the case, why would a 0.401 approximation matter? What is gamma -> is it the same with eps in FPTAS, or is it an algorithmic choice (envelope function), or is it an exogenous parameter characterizing difficulty of the problem (e.g. some functions are gamma-DR for smaller/larger gamma leading to better approximation, etc.)

I suggest including a table summarizing all results, including results of this paper. Each result should have rows like "assumptions", "time complexity", "approximation guarantee", "algorithmic ideas/frameworks", etc.

2. Like I have mentioned in the previous answer, it would be beneficial to include some discussion about applications to ML/AI problems, to make this paper more relevant for TMLR.

---

> ### Author Response · Authors · 2026-04-18
>
> We sincerely thank the reviewer for the positive assessment of the technical results and for the constructive suggestions regarding clarity and presentation.
>
> **(1) Meaning of $O(1/\epsilon)$, $O(1/\epsilon^3)$, and why the $0.401$ guarantee matters**
>
>
> We thank the reviewer for raising this point.  The parameters $\epsilon$ and $\delta$ in our paper are algorithmic accuracy parameters that control the error of a constant factor approximation framework, whereas $\gamma\in(0,1]$ is an exogenous structural parameter of the objective function that measures the strength of the weak DR property. Accordingly, our result is not an FPTAS: it does not provide a $(1-\epsilon)$-approximation. Instead, it gives a constant factor guarantee $\Phi_\gamma$ together with a multiplicative tolerance $O(\epsilon)$ and an additive discretization/smoothness error $O(\delta L D^2)$, as shown in (55). Their effect appears in the running time and in lower order error terms, but they do not turn the algorithm into a $(1-\epsilon)$-approximation scheme. This is exactly why improving the constant factor to $0.401$ at $\gamma=1$ is meaningful.
>
> Similarly, the quantities $O(1/\epsilon)$ and $O(1/\epsilon^3)$ in the cited prior works refer to the dependence of the running time (or oracle complexity) on the accuracy parameter $\epsilon$; they do not mean that those methods are FPTAS-type schemes. In those works as well, the approximation guarantee is centered around a fixed constant (such as $1/e$ or $0.401$), rather than around $1$.
>
> To clarify this confusion, we have added a detailed caption of Table 1.
>
> **(2) Clarifying what $\gamma$ means**
>
> In our paper, $\gamma\in(0,1]$ is an exogenous structural parameter that characterizes the extent to which the objective satisfies diminishing returns. It is not an algorithmic tuning parameter, and it is not the same as $\epsilon$. More precisely, $\gamma=1$ corresponds to the classical DR-submodular case, and the approximation guarantee depends on this structural parameter.
>
> To address this issue, we have added a few lines early in the introduction:
> "The parameter $\gamma\in(0,1]$ is a structural property of the objective. It quantifies how closely the function satisfies full diminishing returns: $\gamma=1$ recovers the classical DR-submodular setting, while smaller values of $\gamma$ allow weaker one sided marginal decay. "
>
> **(3) Suggested summary table**
>
> We appreciate this suggestion and agree that such a table improves readability.
> Accordingly, we have added Table 1 in the revised version.
>
> **(4) Applications to ML/AI**
>
> We agree with the reviewer that the paper would benefit from a clearer discussion of concrete AI/ML applications. In the revised version, we will add application oriented discussion in Appendix G, where we have added details on cost aware feature selection and related subset selection problems in machine learning.

---

### Review · Reviewer_TvdR · 2026-04-14

**Summary Of Contributions:**

This paper studies the problem of maximizing non-monotone $\gamma$-weakly DR-submodular functions over down-closed convex bodies. This setting generalizes continuous DR-submodular optimization, where diminishing returns hold only up to a multiplicative factor $\gamma \in (0,1]$. To address this problem, the paper proposes a unified $\gamma$-aware framework that combines a Frank–Wolfe guided measured continuous greedy method with a $\gamma$-weighted double-greedy algorithm, and optimizes a convex combination of their guarantees. In particular, the approach handles the asymmetry introduced by weak diminishing returns via $\gamma$-dependent progress certificates. The resulting algorithms are projection-free, rely only on first-order information, and run in polynomial time.

**Audience:**

Yes

**Audience Explanation:**

The paper addresses a theoretical problem in submodular optimization and provides improved approximation guarantees under a generalized setting. These results are likely to be of interest to researchers in theoretical machine learning, optimization, and algorithm design.

**Broader Impact Concerns:**

This work is primarily theoretical and focuses on optimization algorithms, with no direct or immediate ethical concerns. However, it would be beneficial for the authors to further discuss potential practical applications of the proposed methods and how their use may impact fairness and efficiency in real-world settings.

**Claims And Evidence:**

Yes

**Claims Explanation:**

The claims are supported by rigorous theoretical analysis. More specifically, the paper presents clear algorithms and establishes the approximation guarantee through a sequence of lemmas and theorems, with a well-structured argument that combines two methods. The treatment of the $\gamma$-weakly DR setting appears sound, although I did not verify every proof in detail. However, the evidence is purely theoretical, as no empirical validation is provided.

**Requested Changes:**

Please find the questions below:
1. Could you provide more intuition on $\Phi\gamma$, explain the optimization procedure more clearly, and add a concise explanation of how the two components interact and why their combination improves the guarantees?
3. Could you streamline the notation, or include a table summarizing the key definitions and symbols?
4. Could you further discuss the practical relevance of the proposed methods (e.g., what types of problems they can address) and include some simple empirical validation?

---

> ### Author Response · Authors · 2026-04-18
>
> We thank the reviewer for the careful reading of our paper and for the positive assessment of its technical soundness and relevance.
>
>
> **(1) Intuition on $\Phi_\gamma$, and why combination improves the guarantees**
>
>
> Intuitively, $\Phi_{\gamma}$ is the final approximation value obtained after combining the two parts of our analysis. For a fixed $\gamma$, it measures the best guarantee we can certify by optimally balancing the contribution of the Frank-Wolfe guided measured continuous greedy component and the $\gamma$-weighted double-greedy component. Thus, $\Phi_{\gamma}$ should be viewed as the solution to the optimization problem in (56).
>
>
> Regarding the optimization procedure, the main algorithm can be understood in two components. First, the Frank-Wolfe guided measured continuous greedy updates construct a feasible fractional trajectory inside the down-closed convex body using only linear optimization steps, thereby producing a global certificate driven by first order information. Second, the $\gamma$-weighted double-greedy argument is applied to the same solution structure to obtain a complementary local certificate that is robust to non-monotonicity. The final approximation guarantee $\Phi_{\gamma}$ is then obtained by optimizing the convex combination of these two bounds, rather than relying on either one alone. The first component is responsible for global exploration and accumulation of gain, the second controls the loss caused by adverse non-monotone directions, and their combination improves the guarantee because each component is stronger in a different regime, so the optimized mixture yields a strictly better worst case bound than either component in isolation. We have revised the paper to make this interaction more explicit.
>
>
>
> **(2) Table summarizing the key definitions and symbols**
>
> We thank the reviewer for this helpful suggestion. To improve readability and clarity, we have added a detailed notation summary and a list of key definitions in Appendix A.
>
>
>
>
> **(3) On the practical relevance of the proposed methods and empirical validation**
>
> We thank the reviewer for this suggestion. In the revised version, we have strengthened the discussion of practical relevance by adding concrete machine learning applications in Appendix G. These applications naturally exhibit the main features addressed by our framework.
>
> We agree that empirical validation would further improve accessibility; however, the current submission is primarily a theory paper whose main contribution is a new worst case approximation guarantee with rigorous proofs. We leave such an empirical investigation as an important direction for future work, while clarifying the practical relevance of the framework in the revised manuscript.

---

### Decision · Action_Editor_t8kY · 2026-07-05

**Recommendation:** Accept with minor revision

**Additional Comments:**

I agree with Reviewer Kvzb that this paper should discuss more and demonstrate some applications in machine learning, given that it is submitted to a machine learning journal. Specifically:

1) The authors have added a dedicated appendix section on machine learning applications (Appendix G). Please incorporate some discussion in the main body of the paper.

2) More importantly, please add some experimental results (at least one, ideally two or three) on machine learning to the paper, based on the discussion in Appendix G. The experimental results need not to be large-scale. Ideally, these experimental results should demonstrate the advantages of the proposed algorithms over the state-of-the-art algorithms. My understanding is that, even for mainly theoretical papers like this paper, some (even preliminary) experimental results can significantly strengthen them.

**Audience:**

Yes

**Audience Explanation:**

My understanding is that this paper studies a theoretical problem in submodular optimization and provides improved approximation guarantees under a generalized setting. These results should be of interest to researchers in theoretical machine learning and optimization.

Reviewer Kvzb has raised two concerns about this submission:

(1) the scope is a bit restricted: optimizing a $\gamma$-weakly-DR-submodular function under convex down-closed constraints

(2) even though the authors add some discussion about application in machine learning, but the application is still weak.

I think both points make sense. For (1), I agree that the scope of this paper is restricted. However, still some researchers in theoretical machine learning and optimization will be interested in this paper. For (2), I will request the authors to make some modifications to further strengthen this paper.

**Claims And Evidence:**

Yes

**Claims Explanation:**

All three reviewers have answered Yes to this question. After reading the paper, the reviews, and the author responses, I agree with the reviewers and also answer Yes to this question.